# WFR-MFM: One-Step Inference for Dynamic Unbalanced Optimal Transport

Xinyu Wang [* 1]  Ruoyu Wang [* 2]  Qiangwei Peng [* 1]  Peijie Zhou [2 3 4 5]  Tiejun Li [1 2 4 6]

## Abstract

Reconstructing dynamical evolution from limited observations is a fundamental challenge in single-cell biology, where dynamic unbalanced optimal transport (OT) provides a principled framework for modeling coupled transport and mass variation. However, existing approaches rely on trajectory simulation at inference time, making inference a key bottleneck for scalable applications. In this work, we propose a mean-flow framework for unbalanced flow matching that summarizes both transport and mass-growth dynamics over arbitrary time intervals using mean velocity and mass-growth fields, enabling fast one-step generation without trajectory simulation. To solve dynamic unbalanced OT under the Wasserstein–Fisher–Rao geometry, we further build on this framework to develop **Wasserstein–Fisher–Rao Mean Flow Matching (WFR-MFM)**. Across synthetic and real single-cell RNA sequencing datasets, WFR-MFM achieves orders-of-magnitude faster inference than a range of existing baselines while maintaining high predictive accuracy, and enables efficient perturbation response prediction on large synthetic datasets with thousands of conditions.

## 1. Introduction

Reconstructing dynamical evolution from limited observations is a fundamental problem across many scientific domains (Chen et al., 2018). The central challenge lies in inferring a plausible evolution that connects these sparse

observations. This challenge is particularly evident in single-cell RNA sequencing (scRNA-seq), where destructive measurements yield independent snapshots at discrete times rather than cell trajectories (Zheng et al., 2017; Chen et al., 2022). Therefore, trajectory inference in scRNA-seq aims to recover the underlying cellular dynamics from such sparse snapshots (Schiebinger et al., 2019b; Tong et al., 2020). A common approach to modeling latent continuous-time dynamics is through neural ordinary differential equations (ODEs) (Chen et al., 2018), which parameterize system evolution via expressive, learnable vector fields capable of representing complex nonlinear dynamics. Despite their flexibility, these models rely on trajectory-level ODE simulation, leading to two major challenges: (i) high computational cost and potential instabilities during **training** (Yan et al., 2019), and (ii) substantial computational overhead at **inference** due to iterative trajectory integration (Song et al., 2023; Frans et al., 2025; Kim et al., 2024).

To alleviate the challenges posed by ODE simulation, recent studies have proposed alternative approaches. Flow Matching (FM) (Lipman et al., 2023; Liu et al., 2022; Pooladian et al., 2023; Albergo & Vanden-Eijnden, 2022) reformulates the learning objective to enable simulation-free training by directly learning the velocity field governing an ODE through supervised regression. While FM enables simulation-free training, inference remains trajectory-based and still requires ODE simulation. Some studies (Yang et al., 2025; Peng et al., 2025) introduce additional consistency constraints when learning instantaneous velocity fields to accelerate inference, while MeanFlow (Geng et al., 2025a) parameterizes the mean velocity field over time intervals to enable direct inference within the FM framework without trajectory simulation. Such approaches enable fast inference but are primarily formulated under mass-preserving assumptions.

However, in many real-world systems this assumption does not hold, as exemplified by single-cell dynamics involving proliferation and apoptosis. Several neural ODE-based models have been successfully applied to capture such unbalanced dynamics (Sha et al., 2024; Peng et al., 2026b; Zhang et al., 2025b; Sun et al., 2025). However, such unbalanced formulations are more computationally demanding than their mass-preserving counterparts, as neural ODE implementations must simulate both transport and mass-growth

[*]Equal contribution [1]LMAM and School of Mathematical Sciences, Peking University, Beijing, China [2]Center for Machine Learning Research, Peking University, Beijing, China [3]Center for Quantitative Biology, Peking University, Beijing, China [4]National Engineering Laboratory for Big Data Analysis and Applications, Beijing, China [5]AI for Science Institute, Beijing, China [6]Center for Data Science, Peking University, Beijing, China. Correspondence to: Peijie Zhou <pjzhou@pku.edu.cn>, Tiejun Li <tieli@pku.edu.cn>.

*Proceedings of the 43rd International Conference on Machine Learning*, Seoul, South Korea. PMLR 306, 2026. Copyright 2026 by the author(s).

dynamics during both training and inference. This has motivated recent work on unbalanced FM, most of which focus on regressing transport velocity fields while neglecting explicit modeling of growth dynamics (Eyring et al., 2024; Corso et al., 2025; Cao et al., 2025). A principled way to jointly model transport and mass variation is provided by dynamic unbalanced optimal transport (OT), with the Wasserstein–Fisher–Rao (WFR) geometry coupling transport with mass creation and annihilation (Liero et al., 2016; Chizat et al., 2018a; Kondratyev et al., 2016). Building on this framework, WFR-FM (Peng et al., 2026a) integrates unbalanced OT and FM under the WFR geometry, jointly learning both a transport vector field and a growth-rate function with simulation-free training. Nevertheless, inference in existing unbalanced FM methods, including WFR-FM, remains trajectory-based and still requires numerical ODE simulation, leaving the inference challenge for unbalanced dynamics largely unresolved.

Efficient inference for unbalanced dynamics has become a central bottleneck in emerging applications such as large-scale single-cell perturbation prediction. Recent studies have increasingly focused on predicting cellular responses to perturbations from single-cell data at scale (Roohani et al., 2024; Lotfollahi et al., 2019; Zinati et al., 2024; Klein et al., 2025). Single-cell perturbations inherently induce unbalanced dynamics, with large changes in cell abundance before and after perturbation, motivating dynamic unbalanced modeling. Crucially, the space of perturbation conditions expands combinatorially across modality, dosage, timing, and experimental context such as cell line, tissue, and batch effects (Klein et al., 2025). Together, these factors make ODE-based inference impractical for perturbation prediction, making fast inference a necessity.

To address this remaining challenge, we introduce a mean-flow framework for unbalanced FM that enables fast one-step generation for mass-varying dynamics by directly advancing system states over finite time intervals, without relying on trajectory-level ODE simulation. To solve dynamic unbalanced OT under the WFR geometry, we further develop **Wasserstein–Fisher–Rao Mean Flow Matching (WFR-MFM)** based on the proposed mean-flow framework, yielding a mean-flow matching algorithm with simulation-free training and inference. We validate the effectiveness of WFR-MFM on a range of synthetic and real scRNA-seq datasets, including a large-scale synthetic perturbation dataset, with systematic comparisons to representative existing methods (Sha et al., 2024; Zhang et al., 2025b; Huguet et al., 2022; Sun et al., 2025; Tong et al., 2024b; Kapusniak et al., 2024; Rohbeck et al., 2025; Eyring et al., 2024; Wang et al., 2025; Peng et al., 2026a). Our contributions can be summarized as follows:

- We formulate a mean-flow framework for unbalanced

FM, which summarizes transport and mass-growth dynamics over arbitrary time intervals through mean velocity and mass-growth fields. This formulation replaces trajectory-based inference with direct state-to-state updates, enabling rapid one-step generation without simulating continuous-time dynamics.

- We propose **WFR-MFM**, a mean flow matching algorithm to approximate dynamic unbalanced OT under the WFR geometry. By learning mean velocity and mass-growth fields, the algorithm enables direct one-step generation along WFR geodesics.

- We evaluate WFR-MFM using synthetic and real scRNA-seq datasets, comparing its inference efficiency and accuracy with existing methods. Furthermore, we apply WFR-MFM to a large-scale synthetic perturbation dataset to reveal its potential for inference in perturbation response prediction in single-cell biology.

## 2. Related Work

**Flow Matching.** Flow matching (Lipman et al., 2023; Liu et al., 2022; Pooladian et al., 2023; Albergo & Vanden-Eijnden, 2022) provides a simulation-free framework for learning continuous velocity fields through conditional probability paths in generative modeling, avoiding ODE-based path simulation during training. The framework has been extended to encompass OT couplings (Tong et al., 2024a; Klein et al., 2024), conditional generation (Rohbeck et al., 2025), multi-marginal formulations (Rohbeck et al., 2025; Lee et al., 2025), stochastic dynamics (Tong et al., 2024b; Lee et al., 2025), and other generalizations (Kapusniak et al., 2024; Zhang et al., 2024; Atanackovic et al., 2025; Petrović et al., 2025). However, most methods assume normalized distributions and thus mass conservation. In biological systems with proliferation and apoptosis, dynamics are inherently mass-varying, motivating recent work on unbalanced FM (Eyring et al., 2024; Cao et al., 2025; Corso et al., 2025; Wang et al., 2025; Peng et al., 2026a; Song et al., 2025). Among these, WFR-FM (Peng et al., 2026a) uniquely provides an OT-consistent formulation that strictly recovers dynamic unbalanced OT under the WFR geometry.

**Few-Step Flow Models.** Few-step generation has been widely explored in diffusion models via consistency-based objectives (Song et al., 2023), and similar efficiency concerns have recently driven few-step flow models in FM. To reduce the inference overhead caused by ODE-based trajectory simulation in FM, few-step or one-step flow models have been proposed for fast generation. The first class of methods (Yang et al., 2025; Peng et al., 2025; Frans et al., 2025) still learns the instantaneous velocity field and accelerates sampling via additional consistency constraints. For example, Consistency-FM (Yang et al., 2025) enforces

self-consistency of the velocity field with a multi-segment training strategy. More recently, MeanFlow (Geng et al., 2025a) departs from instantaneous dynamics by parameterizing velocity fields averaged over finite time intervals, enabling direct one-step inference within the FM framework. Subsequent works propose various refinements to the MeanFlow framework (Guo et al., 2025; Zhang et al., 2025a; Geng et al., 2025b). Beyond averaged velocity parameterizations, closely related solution-based formulations target one-step generation from scratch by learning the solution function of the velocity ODE, as exemplified by SoFlow (Luo et al., 2025). However, these approaches are fundamentally built on mass-preserving assumptions and are not applicable to unbalanced dynamics.

**Unbalanced Optimal Transport and Dynamic Models of scRNA-seq Data.** Unbalanced optimal transport generalizes classical OT (Kantorovich, 1942; Benamou & Brenier, 2000) by allowing mass variation, which is essential for modeling biological systems involving proliferation and apoptosis. It relaxes the hard marginal constraints of classical OT (Benamou, 2003; Figalli, 2010; Caffarelli & McCann, 2010) by penalizing deviations of transported mass through suitable divergence terms. The WFR metric (Liero et al., 2016; Chizat et al., 2018a; Kondratyev et al., 2016) provides a dynamic formulation of unbalanced OT that jointly models transport and mass variation through coupled velocity and mass-growth fields. Recently, deep learning solvers for dynamic WFR have been developed to characterize continuous-time single-cell dynamics (Neklyudov et al., 2023; 2024; Sha et al., 2024; Peng et al., 2026b; Zhang et al., 2025b; Sun et al., 2025; Peng et al., 2026a). Despite providing OT-consistent descriptions of unbalanced dynamics, existing methods typically rely on trajectory-based simulation at inference time rather than direct one-step generation, leading to substantial computational overhead at scale. Motivated by this limitation, we develop **WFR-MFM** under the WFR geometry as an efficient method for inference in dynamic unbalanced optimal transport.

## 3. Mathematical Background

**Setup.** We consider two endpoint measures $\mu_0$ and $\mu_1$ supported on a domain $\mathcal{X} \subset \mathbb{R}^d$, with density functions $\mu_0(\boldsymbol{x})$ at $t = 0$ and $\mu_1(\boldsymbol{x})$ at $t = 1$. Let $\mathcal{M}_+(\mathcal{X})$ denote the space of finite, absolutely continuous nonnegative measures on $\mathcal{X}$. The total masses of $\mu_0$ and $\mu_1$ are not required to match.

**Flow Matching for Probability Flow.** FM (Lipman et al., 2023; Liu et al., 2022; Pooladian et al., 2023; Albergo & Vanden-Eijnden, 2022) provides a general framework for learning continuous-time dynamics of measures by regressing parameterized vector fields to analytically specified tar-

gets. In the mass-preserving setting, FM models a time-dependent probability path $\{\rho_t\}_{t \in [0,1]}$ with velocity field $\boldsymbol{u}_t$ satisfying the continuity equation

$$\partial_t \rho_t + \nabla \cdot (\rho_t \boldsymbol{u}_t) = 0,$$

which characterizes deterministic measure flows induced by transport. If a kinetic energy minimization over admissible transport paths is imposed, the formulation reduces to the dynamic OT problem (Benamou & Brenier, 2000). FM learns such dynamics by regressing a parameterized vector field $\boldsymbol{u}_\theta(\boldsymbol{x}, t)$ to the true field, avoiding backpropagation through ODE solvers:

$$\mathcal{L}_{\text{FM}}(\theta) = \mathbb{E}_{t, \boldsymbol{x} \sim \rho_t} \left[ \|\boldsymbol{u}_\theta(\boldsymbol{x}, t) - \boldsymbol{u}_t(\boldsymbol{x})\|_2^2 \right].$$

To obtain a tractable training objective, FM constructs conditional paths $\rho_t(\cdot \mid \boldsymbol{z})$ admitting closed-form vector fields $\boldsymbol{u}_t(\cdot \mid \boldsymbol{z})$, and defines the regression objective

$$\mathcal{L}_{\text{CFM}}(\theta) = \mathbb{E}_{t, \boldsymbol{z}, \boldsymbol{x} \sim \rho_t(\cdot \mid \boldsymbol{z})} \left[ \|\boldsymbol{u}_\theta(\boldsymbol{x}, t) - \boldsymbol{u}_t(\boldsymbol{x} \mid \boldsymbol{z})\|_2^2 \right].$$

When the conditional paths and coupling distribution are chosen to match the classical OT formulation, the learned dynamics recover the geodesics of the dynamic OT problem (Tong et al., 2024a).

**Variable mass flow.** Consider a measure path $: [0, 1] \times \mathbb{R}^d \to \mathbb{R}_+$, together with a velocity field $\boldsymbol{u}_t : [0, 1] \times \mathbb{R}^d \to \mathbb{R}^d$ and a growth-rate function $g_t : [0, 1] \times \mathbb{R}^d \to \mathbb{R}$. Assuming sufficient regularity, $(\boldsymbol{u}_t, g_t)$ induces a unique time-dependent flow $\boldsymbol{\phi}_t : [0, 1] \times \mathbb{R}^d \to \mathbb{R}^d$, $m_t : [0, 1] \times \mathbb{R}^d \to \mathbb{R}$ defined via an ODE:

$$\frac{d}{dt} \begin{pmatrix} \boldsymbol{\phi}_t(\boldsymbol{x}) \\ \ln m_t(\boldsymbol{x}) \end{pmatrix} = \begin{pmatrix} \boldsymbol{u}_t(\boldsymbol{\phi}_t(\boldsymbol{x})) \\ g_t(\boldsymbol{\phi}_t(\boldsymbol{x})) \end{pmatrix},$$

$$\boldsymbol{\phi}_0(\boldsymbol{x}) = \boldsymbol{x}, \quad m_0(\boldsymbol{x}) \text{ given,}$$

which defines a push-forward mapping with changing mass $(\boldsymbol{\phi}_t, m_t)_\# : \mathcal{M}_+(\mathbb{R}^d) \to \mathcal{M}_+(\mathbb{R}^d)$ for each $t \in [0, 1]$ such that

$$\rho_t(\boldsymbol{x}) = (\boldsymbol{\phi}_t, m_t)_\# \rho_0(\boldsymbol{x})$$
$$:= \rho_0(\boldsymbol{\phi}_t^{-1}(\boldsymbol{x})) \left| \det \nabla_{\boldsymbol{x}} \boldsymbol{\phi}_t^{-1}(\boldsymbol{x}) \right| \frac{m_t(\boldsymbol{x})}{m_0(\boldsymbol{\phi}_t^{-1}(\boldsymbol{x}))}$$

and satisfies the continuity equation with source term

$$\partial_t \rho_t(\boldsymbol{x}) + \nabla_{\boldsymbol{x}} \cdot (\boldsymbol{u}_t(\boldsymbol{x}) \rho_t(\boldsymbol{x})) = g_t(\boldsymbol{x}) \rho_t(\boldsymbol{x}).$$

Furthermore, imposing a variational principle penalizing transport and mass variation yields dynamic unbalanced OT problems. A prominent example is the WFR metric (Chizat et al., 2018a;b; Liero et al., 2018), which provides a principled formulation for the dynamic evolution $\{\rho_t\}_{t \in [0,1]}$

from $\mu_0$ to $\mu_1$ while jointly modeling transport and mass variation:

$$
\begin{aligned}
\mathrm{WFR}_\delta^2(\mu_0, \mu_1) = \inf_{\rho, \boldsymbol{u}, g} \int_0^1 & \int_{\mathcal{X}} \tfrac{1}{2} \big( \|\boldsymbol{u}(\boldsymbol{x}, t)\|_2^2 \\
& + \delta^2 \|g(\boldsymbol{x}, t)\|_2^2 \big) \rho_t(\boldsymbol{x}) \, \mathrm{d}\boldsymbol{x} \mathrm{d}t,
\end{aligned}
\tag{1}
$$

$$
\text{s.t. } \partial_t \rho + \nabla_{\boldsymbol{x}} \cdot (\rho \boldsymbol{u}) = \rho g, \quad \rho_0 = \mu_0, \; \rho_1 = \mu_1.
$$

**Unbalanced Flow Matching.** Unbalanced FM extends FM to variable-mass dynamics by jointly learning a velocity field $\boldsymbol{u}_t$ and a growth-rate function $g_t$ via neural networks $\boldsymbol{u}_{\boldsymbol{\theta}}(\boldsymbol{x}, t)$ and $g_{\boldsymbol{\phi}}(\boldsymbol{x}, t)$ trained under the unbalanced FM objective

$$
\begin{aligned}
\mathcal{L}_{\mathrm{UFM}}(\boldsymbol{\theta}, \boldsymbol{\phi}) = \mathbb{E}_{t, \boldsymbol{x} \sim \rho_t} \Big( & \|\boldsymbol{u}_{\boldsymbol{\theta}}(\boldsymbol{x}, t) - \boldsymbol{u}_t(\boldsymbol{x})\|_2^2 \\
& + \lambda \|g_{\boldsymbol{\phi}}(\boldsymbol{x}, t) - g_t(\boldsymbol{x})\|_2^2 \Big).
\end{aligned}
$$

To obtain a tractable training objective, unbalanced FM adopts a latent-variable formulation with $\boldsymbol{z} = ((\boldsymbol{x}_0, m_0), (\boldsymbol{x}_1, m_1))$ sampled from a coupling distribution $q(\boldsymbol{z})$, and constructs conditional measure paths $\rho_t(\boldsymbol{x} \mid \boldsymbol{z})$ together with corresponding conditional target fields, leading to the conditional unbalanced flow matching (CUFM) objective

$$
\begin{aligned}
\mathcal{L}_{\mathrm{CUFM}}(\boldsymbol{\theta}, \boldsymbol{\phi}) = \mathbb{E}_{t, \boldsymbol{z}, \boldsymbol{x} \sim \rho_t(\boldsymbol{x}|\boldsymbol{z})} \Big( & \|\boldsymbol{u}_{\boldsymbol{\theta}}(\boldsymbol{x}, t) - \boldsymbol{u}_t(\boldsymbol{x}|\boldsymbol{z})\|_2^2 \\
& + \lambda \|g_{\boldsymbol{\phi}}(\boldsymbol{x}, t) - g_t(\boldsymbol{x}|\boldsymbol{z})\|_2^2 \Big).
\end{aligned}
$$

With appropriate conditional paths and couplings, unbalanced FM recovers the geodesics of dynamic unbalanced OT; for example, under the WFR geometry, it leads to the WFR-FM formulation (Peng et al., 2026a). Nevertheless, existing unbalanced FM methods require simulating the full state trajectory $(x_t, m_t)_{t \in [0,1]}$ during inference in order to obtain the terminal state $(x_1, m_1)$, which incurs substantial computational cost at scale. We next introduce a mean-flow framework for unbalanced FM that enables direct one-step inference.

## 4. Mean-Flow for Unbalanced Dynamic OT

In this section, we introduce the concept of *mean flow* and formulate a general framework for one-step generation in unbalanced FM. We then specialize this framework to solve dynamic unbalanced OT under the WFR geometry, resulting in WFR-MFM.

### 4.1. Mean-flow Variables and Derivative Identities

**Mean-flow Variables.** To this end, we introduce time-averaged dynamics through the *mean velocity* $\boldsymbol{v}(\boldsymbol{x}, t, T)$ and *mean mass-growth rate* $h(\boldsymbol{x}, t, T)$. For any trajectory

$\boldsymbol{x}_\tau$ over a time interval $(t, T)$ with $t < T$, we define

$$
\begin{aligned}
\boldsymbol{v}(\boldsymbol{x}, t, T) &= \frac{1}{T - t} \int_t^T \boldsymbol{u}_\tau(\boldsymbol{x}_\tau) \, d\tau, \\
h(\boldsymbol{x}, t, T) &= \frac{1}{T - t} \int_t^T g_\tau(\boldsymbol{x}_\tau) \, d\tau,
\end{aligned}
\tag{2}
$$

where $\boldsymbol{u}_\tau$ and $g_\tau$ denote the transport velocity and mass-growth rate, respectively. These mean-flow variables represent the mean velocity and mass-growth rate required to evolve the system from $(\boldsymbol{x}_t, m_t)$ at time $t$ to $(\boldsymbol{x}_T, m_T)$ at time $T$ under the flow

$$
\frac{d\boldsymbol{x}_\tau}{d\tau} = \boldsymbol{u}_\tau(\boldsymbol{x}_\tau), \quad \boldsymbol{x}_t = \boldsymbol{x}.
$$

By definition, the mean-flow variables $\boldsymbol{v}$ and $h$ satisfy appropriate boundary conditions and inherent consistency constraints. As the time interval shrinks, they recover the instantaneous quantities:

$$
\lim_{T \to t} \boldsymbol{v}(\boldsymbol{x}, t, T) = \boldsymbol{u}(\boldsymbol{x}, t), \quad \lim_{T \to t} h(\boldsymbol{x}, t, T) = g(\boldsymbol{x}, t).
$$

Equivalently, the definitions of $\boldsymbol{v}$ and $h$ can be rewritten as

$$
\begin{aligned}
(T - t) \, \boldsymbol{v}(\boldsymbol{x}, t, T) &= \int_t^T \boldsymbol{u}_\tau(\boldsymbol{x}_\tau) \, d\tau, \\
(T - t) \, h(\boldsymbol{x}, t, T) &= \int_t^T g_\tau(\boldsymbol{x}_\tau) \, d\tau.
\end{aligned}
\tag{3}
$$

**Additive consistency.** The additivity of the integral implies, for any intermediate $s$ with $t < s < T$,

$$
(T - t) \, \boldsymbol{v}(\boldsymbol{x}, t, T) = (s - t) \, \boldsymbol{v}(\boldsymbol{x}, t, s) + (T - s) \, \boldsymbol{v}(\boldsymbol{x}_s, s, T).
$$

Therefore, $\boldsymbol{v}$ satisfies multi-step consistency over any partition $t_0 = t < t_1 < \cdots < t_K = T$:

$$
(T - t) \, \boldsymbol{v}(\boldsymbol{x}, t, T) = \sum_{k=0}^{K-1} (t_{k+1} - t_k) \, \boldsymbol{v}(\boldsymbol{x}_{t_k}, t_k, t_{k+1}).
$$

The mean mass-growth rate field $h(\boldsymbol{x}, t, T)$ satisfies the same additive and multi-step consistency.

**Derivative identities.** Differentiating Eq. (3) with respect to $t$ yields

$$
\begin{aligned}
\boldsymbol{v}(\boldsymbol{x}, t, T) &= \boldsymbol{u}_t(\boldsymbol{x}) + (T - t) \frac{d}{dt} \boldsymbol{v}(\boldsymbol{x}, t, T), \\
h(\boldsymbol{x}, t, T) &= g_t(\boldsymbol{x}) + (T - t) \frac{d}{dt} h(\boldsymbol{x}, t, T),
\end{aligned}
$$

where

$$
\begin{aligned}
\frac{d}{dt} \boldsymbol{v}(\boldsymbol{x}, t, T) &= \partial_t \boldsymbol{v}(\boldsymbol{x}, t, T) + (\nabla_{\boldsymbol{x}} \boldsymbol{v}(\boldsymbol{x}, t, T)) \, \boldsymbol{u}_t(\boldsymbol{x}), \\
\frac{d}{dt} h(\boldsymbol{x}, t, T) &= \partial_t h(\boldsymbol{x}, t, T) + \nabla_{\boldsymbol{x}} h(\boldsymbol{x}, t, T) \cdot \boldsymbol{u}_t(\boldsymbol{x}).
\end{aligned}
$$

These derivative identities link the mean-flow variables $(\boldsymbol{v}, h)$ to the instantaneous fields $(\boldsymbol{u}_t, g_t)$.

**Algorithm 1** Mean-Flow Inference for Unbalanced FM

**Input:** mean-flow variables $v$ and $h$; initial state $(\boldsymbol{x}_0, m_0)$; time partition $0 = t_0 < \cdots < t_K = 1$ (optional).
**Output:** Terminal state $(\boldsymbol{x}_1, m_1)$.
$(\boldsymbol{x}_{t_0}, m_{t_0}) \leftarrow (\boldsymbol{x}_0, m_0)$
**for** $k = 0$ **to** $K - 1$ **do**
$\quad \boldsymbol{x}_{t_{k+1}} \leftarrow \boldsymbol{x}_{t_k} + (t_{k+1} - t_k)\, \boldsymbol{v}(\boldsymbol{x}_{t_k}, t_k, t_{k+1})$
$\quad m_{t_{k+1}} \leftarrow m_{t_k} \exp\big((t_{k+1} - t_k)\, h(\boldsymbol{x}_{t_k}, t_k, t_{k+1})\big)$
**end for**
**return** $(\boldsymbol{x}_1, m_1)$

## 4.2. Inference and Training

**Inference.** Given the mean-flow variables $(\boldsymbol{v}, h)$, the mean-flow framework provides a direct terminal mapping for unbalanced dynamics. The **one-step update**

$$\boldsymbol{x}_1 = \boldsymbol{x}_0 + \boldsymbol{v}(\boldsymbol{x}_0, 0, 1), \quad m_1 = m_0 \exp\big(h(\boldsymbol{x}_0, 0, 1)\big)$$

computes the terminal state in a single evaluation. More generally, inference can be performed via a **multi-step update** based on additive consistency. For a partition $0 = t_0 < t_1 < \cdots < t_K = 1$:

$$\boldsymbol{x}_{t_{k+1}} = \boldsymbol{x}_{t_k} + (t_{k+1} - t_k)\, \boldsymbol{v}(\boldsymbol{x}_{t_k}, t_k, t_{k+1}),$$
$$m_{t_{k+1}} = m_{t_k} \exp\big((t_{k+1} - t_k)\, h(\boldsymbol{x}_{t_k}, t_k, t_{k+1})\big).$$

When intermediate time points are available, the partition follows the data; otherwise it can be chosen arbitrarily. One-step inference corresponds to $K = 1$, while larger $K$ provides a finer approximation to the underlying continuous dynamics. The complete inference procedure is summarized in Algorithm 1.

**Training.** We train the mean-flow variables $(\boldsymbol{v}, h)$ within the mean-flow framework by parameterizing them with neural networks and enforcing the derivative identities in Sec. 4.1 via the **unbalanced mean flow matching loss**:

$$\mathcal{L}(\boldsymbol{\theta}, \boldsymbol{\phi}) = \mathbb{E}_{t < T,\ \boldsymbol{x} \sim \rho_t(\boldsymbol{x})} \Big[ \|\boldsymbol{v}_{\boldsymbol{\theta}}(\boldsymbol{x}, t, T) - \mathrm{sg}(\boldsymbol{v}(\boldsymbol{x}, t, T))\|_2^2$$
$$+ \lambda \|h_{\boldsymbol{\phi}}(\boldsymbol{x}, t, T) - \mathrm{sg}(h(\boldsymbol{x}, t, T))\|_2^2 \Big],$$

where $\lambda > 0$ balances the supervision of the velocity and mass-growth fields, and $\mathrm{sg}(\cdot)$ is the stop-gradient operator. The regression targets $(\boldsymbol{v}, h)$ are computed via the derivative identities

$$\boldsymbol{v}(\boldsymbol{x}, t, T) = \boldsymbol{u}_t(\boldsymbol{x}) + (T - t)\Big[\partial_t \boldsymbol{v}_{\boldsymbol{\theta}}(\boldsymbol{x}, t, T)$$
$$+ (\nabla_{\boldsymbol{x}} \boldsymbol{v}_{\boldsymbol{\theta}}(\boldsymbol{x}, t, T))\, \boldsymbol{u}_t(\boldsymbol{x})\Big],$$

$$h(\boldsymbol{x}, t, T) = g_t(\boldsymbol{x}) + (T - t)\Big[\partial_t h_{\boldsymbol{\phi}}(\boldsymbol{x}, t, T)$$
$$+ \nabla_{\boldsymbol{x}} h_{\boldsymbol{\phi}}(\boldsymbol{x}, t, T) \cdot \boldsymbol{u}_t(\boldsymbol{x})\Big].$$

Minimizing the regression loss enforces $(\boldsymbol{v}_{\boldsymbol{\theta}}, h_{\boldsymbol{\phi}})$ to recover the true mean-flow variables. To obtain a tractable training objective, we introduce conditional mean-field targets $\boldsymbol{v}(\boldsymbol{x}, t, T \mid \boldsymbol{z})$ and $h(\boldsymbol{x}, t, T \mid \boldsymbol{z})$, leading to the **conditional unbalanced mean flow matching loss**:

$$\mathcal{L}_{\mathrm{c}}(\boldsymbol{\theta}, \boldsymbol{\phi}) = \mathbb{E}_{(t,T)\,:\,t<T, \boldsymbol{z}, \boldsymbol{x} \sim \rho_t(\boldsymbol{x}|\boldsymbol{z})} \Big[ \|\boldsymbol{v}_{\boldsymbol{\theta}}(\boldsymbol{x}, t, T)$$
$$- \mathrm{sg}(\boldsymbol{v}(\boldsymbol{x}, t, T \mid \boldsymbol{z}))\|_2^2 + \lambda \|h_{\boldsymbol{\phi}}(\boldsymbol{x}, t, T)$$
$$- \mathrm{sg}(h(\boldsymbol{x}, t, T \mid \boldsymbol{z}))\|_2^2 \Big],$$

where the conditional regression targets are defined as

$$\boldsymbol{v}(\boldsymbol{x}, t, T \mid \boldsymbol{z}) = \boldsymbol{u}_t(\boldsymbol{x} \mid \boldsymbol{z}) + (T - t)\Big[\partial_t \boldsymbol{v}_{\boldsymbol{\theta}}(\boldsymbol{x}, t, T)$$
$$+ (\nabla_{\boldsymbol{x}} \boldsymbol{v}_{\boldsymbol{\theta}}(\boldsymbol{x}, t, T))\, \boldsymbol{u}_t(\boldsymbol{x} \mid \boldsymbol{z})\Big],$$

$$h(\boldsymbol{x}, t, T \mid \boldsymbol{z}) = g_t(\boldsymbol{x} \mid \boldsymbol{z}) + (T - t)\Big[\partial_t h_{\boldsymbol{\phi}}(\boldsymbol{x}, t, T)$$
$$+ \nabla_{\boldsymbol{x}} h_{\boldsymbol{\phi}}(\boldsymbol{x}, t, T) \cdot \boldsymbol{u}_t(\boldsymbol{x} \mid \boldsymbol{z})\Big].$$

Here $\boldsymbol{u}_t(\boldsymbol{x} \mid \boldsymbol{z})$ and $g_t(\boldsymbol{x} \mid \boldsymbol{z})$ represent the instantaneous transport velocity and the mass-growth rate associated with the conditional measure path.

The following theorem shows that the unconditional objective differs from its conditional counterpart only by a constant, thereby justifying the optimization of the tractable conditional loss $\mathcal{L}_{\mathrm{c}}$. The proof is provided in Appendix B.1.

**Theorem 4.1.** *If $\rho_t(\boldsymbol{x}) > 0$ for all $\boldsymbol{x} \in \mathcal{X}$ and $q(\boldsymbol{z})$ is independent of $(\boldsymbol{x}, T)$, then $\mathcal{L}(\boldsymbol{\theta}, \boldsymbol{\phi}) = \mathcal{L}_{\mathrm{c}}(\boldsymbol{\theta}, \boldsymbol{\phi}) + C$, for a constant $C$ that does not depend on $(\boldsymbol{\theta}, \boldsymbol{\phi})$. Consequently,*

$$\nabla_{(\boldsymbol{\theta}, \boldsymbol{\phi})} \mathcal{L}(\boldsymbol{\theta}, \boldsymbol{\phi}) = \nabla_{(\boldsymbol{\theta}, \boldsymbol{\phi})} \mathcal{L}_{\mathrm{c}}(\boldsymbol{\theta}, \boldsymbol{\phi}).$$

Thus, the conditional and unconditional objectives yield identical optimal estimators of the mean-flow variables, and the method naturally extends from two to multiple time points.

*Remark* 4.2. The choice of the mean-flow targets and their conditional counterparts in the unbalanced setting is *not trivial*. Their specific construction via the mean-flow derivative identities is crucial for the validity of Theorem 4.1. A seemingly natural alternative is to define the learning objective directly according to the mean-flow definition in Eq. (2). The corresponding conditional mean-flow variables are given by

$$\tilde{\boldsymbol{v}}(\boldsymbol{x}, t, T \mid \boldsymbol{z}) = \frac{1}{T - t} \int_t^T \boldsymbol{u}_\tau(\boldsymbol{x}_\tau \mid \boldsymbol{z})\, d\tau,$$
$$\tilde{h}(\boldsymbol{x}, t, T \mid \boldsymbol{z}) = \frac{1}{T - t} \int_t^T g_\tau(\boldsymbol{x}_\tau \mid \boldsymbol{z})\, d\tau.$$

However, such constructions generally break the equivalence between the conditional and unconditional objectives, whose proof is provided in Appendix B.2.

**Algorithm 2** WFR-MFM Training

**Input:** Sample-able distributions $\mu_0, \mu_1$; bandwidth $\sigma$; OET batch size $B$; training batch size $b$; WFR penalty $\delta$; mean-flow variables $\boldsymbol{v_\theta}(\boldsymbol{x}, t, T)$ and $h_\phi(\boldsymbol{x}, t, T)$

**Output:** Trained mean fields $\boldsymbol{v_\theta}$ and $h_\phi$

**for** $k = 0 \to K - 1$ **do**

$\quad \gamma^{(k)} \leftarrow \text{OET}(\mu_{t_k}, \mu_{t_{k+1}})$ **or** mini-batch $\text{OET}(\mu_{t_k}, \mu_{t_{k+1}}; B)$

$\quad \gamma_0^{(k)}(\boldsymbol{x}, \boldsymbol{y}) \leftarrow \frac{\gamma^{(k)}(\boldsymbol{x}, \boldsymbol{y})}{\int_{\mathcal{X}} \gamma^{(k)}(\boldsymbol{x}, \boldsymbol{z}) \mathrm{d}\boldsymbol{z}} \mu_{t_k}(\boldsymbol{x})$

**end for**

**while** Training **do**

$\quad$ **for** $k = 0 \to K - 1$ **do**

$\qquad$ Sample $b$ pairs $(\boldsymbol{x}_{t_k}, \boldsymbol{x}_{t_{k+1}}) \sim \gamma_0$

$\qquad$ Compute travelling-Dirac constants $A, B, \boldsymbol{\omega}_0, \tau$

$\qquad$ Sample time pairs $(t, T) \in [t_k, t_{k+1}]$ with $t < T$

$\qquad \boldsymbol{\eta}_{t^{(k)}} \leftarrow \boldsymbol{x}_{t_k} + \boldsymbol{\omega}_0 \Lambda_t(\boldsymbol{x}_{t_k}, \boldsymbol{x}_{t_{k+1}})$

$\qquad$ Sample $\boldsymbol{x}^{(k)} \sim \mathcal{N}(\boldsymbol{\eta}_{t^{(k)}}, \sigma^2 \mathbf{I})$

$\qquad \boldsymbol{u}^{(k)} \leftarrow \boldsymbol{\omega}_0 / (m_t(\boldsymbol{x}_{t_k}, \boldsymbol{x}_{t_{k+1}})(t_{k+1} - t_k))$

$\qquad g^{(k)} \leftarrow \frac{\mathrm{d}}{\mathrm{d}t} \ln m_t(\boldsymbol{x}_{t_k}, \boldsymbol{x}_{t_{k+1}}) / (t_{k+1} - t_k)$

$\qquad m^{(k)} \leftarrow m_t(\boldsymbol{x}_{t_k}, \boldsymbol{x}_{t_{k+1}}) / \gamma_0^{(k)}(\boldsymbol{x}_{t_k}, \boldsymbol{x}_{t_{k+1}})$

$\qquad \boldsymbol{v}^{(k)} \leftarrow \boldsymbol{u}^{(k)} + \text{sg}((T - t)[\partial_t \boldsymbol{v_\theta}(\boldsymbol{x}, t, T) + (\nabla_{\boldsymbol{x}} \boldsymbol{v_\theta}(\boldsymbol{x}, t, T)) \boldsymbol{u}^{(k)}])$

$\qquad h^{(k)} \leftarrow g^{(k)} + \text{sg}((T - t)[\partial_t h_\phi(\boldsymbol{x}, t, T) + \nabla_{\boldsymbol{x}} h_\phi(\boldsymbol{x}, t, T) \cdot \boldsymbol{u}^{(k)}])$

$\quad$ **end for**

$\quad$ Concatenate $\{\boldsymbol{x}^{(i)}, t^{(i)}, T^{(i)}, \boldsymbol{v}^{(i)}, h^{(i)}, m^{(i)}\}_{i=1}^K$ into batched tensors $\{\boldsymbol{x}^b, t^b, T^b, \boldsymbol{v}^b, h^b, m^b\}$

$\quad \mathcal{L}_c(\boldsymbol{\theta}, \boldsymbol{\phi}) \leftarrow (\|\boldsymbol{v_\theta}(\boldsymbol{x}^b, t^b, T^b) - \boldsymbol{v}^b\|_2^2 + \lambda \|h_\phi(\boldsymbol{x}^b, t^b, T^b) - h^b\|_2^2) m^b$

$\quad (\boldsymbol{\theta}, \boldsymbol{\phi}) \leftarrow \text{Update}((\boldsymbol{\theta}, \boldsymbol{\phi}), (\nabla_{\boldsymbol{\theta}} \mathcal{L}_c, \nabla_{\boldsymbol{\phi}} \mathcal{L}_c))$

**end while**

**return** $\boldsymbol{v_\theta}$ and $h_\phi$.

**WFR-MFM for Dynamic Unbalanced OT.** Within the general mean-flow framework for unbalanced FM, the choice of the coupling distribution $q(z)$ determines the underlying transport geometry. When $q(z)$ is chosen as the unbalanced OT coupling induced by an optimal entropy transport problem under the WFR geometry, the resulting mean-flow dynamics follow WFR geodesics and admit direct state-to-state evolution. Theoretical details on the construction of $q(z)$ are provided in Appendix A.1.

This specialization gives rise to WFR-MFM, a mean-flow algorithm for dynamic unbalanced OT that enables one-step inference under the WFR geometry without trajectory-level ODE simulation. The training algorithm is summarized in Algorithm 2.

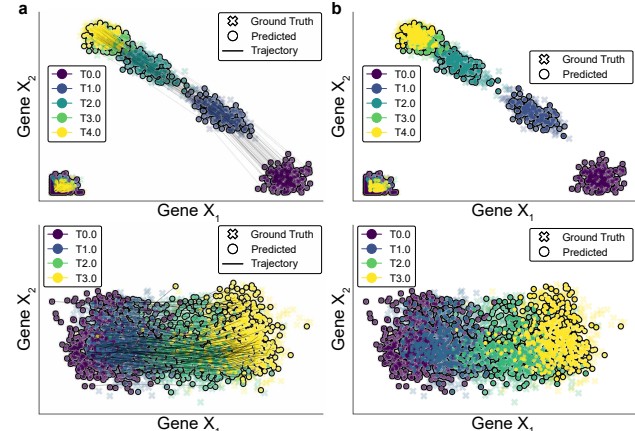

*Figure 1.* **Learned dynamics on the Gene and EMT datasets.** Top: Gene dataset; bottom: EMT dataset. (a) Individual cell trajectories with evolving weights. (b) Population-level dynamics obtained by resampling cells according to normalized weights.

# 5. Numerical Results

We evaluate WFR-MFM by addressing the following questions: **Q1**: Does WFR-MFM significantly accelerate inference? **Q2**: Does WFR-MFM maintain high predictive accuracy under fast inference? **Q3**: Is there a controllable trade-off between inference speed and accuracy? **Q4**: Is WFR-MFM scalable in both training and inference?

**Massive acceleration in inference speed (Q1).** We evaluate inference runtime on three synthetic datasets (Gene, Dyngen, Gaussian), the real-world embryoid bodies (EB) dataset (Moon et al., 2019), and three real multi-time scRNA-seq datasets (EMT (Cook & Vanderhyden, 2020), CITE-seq (Lance et al., 2022), and Mouse hematopoiesis (Weinreb et al., 2020)). Existing methods for dynamic unbalanced OT perform inference via ODE simulation and thus incur similar computational costs. As a representative baseline, we compared against WFR-FM, which follows the standard ODE-based inference paradigm. Its original implementation uses an adaptive Dormand–Prince RK5(4) solver (Dormand & Prince, 1980); we additionally report results with a fixed-step explicit Euler solver (100 steps). The runtime comparison is presented in Table 1. Across all datasets, WFR-MFM achieves substantial acceleration relative to these ODE-based baselines, providing inference *speedups of two to three orders of magnitude* compared to adaptive RK solvers and remaining approximately *two orders of magnitude faster* than 100-step Euler solvers. These results demonstrate the significant advantages of WFR-MFM in fast inference.

**Accuracy preserved under fast inference (Q2).** A natural concern is whether accuracy of distribution will decrease. We evaluate WFR-MFM on synthetic datasets (Table 2) and

*Table 1.* **Inference time comparison across datasets.** All values are reported as *mean ± sd* (seconds per inference). $N_0$ denotes the number of cells at the initial time point and $T$ is the number of future time points to infer. WFR-MFM is evaluated over 1000 runs, while WFR-FM with adaptive Dormand–Prince RK5(4) and with fixed-step Euler (100 steps) are each evaluated over 100 runs. *Spd-RK* and *Spd-Eu* denote the speedup of WFR-MFM relative to WFR-FM (RK) and WFR-FM (Euler), respectively.

| DATASET | DIM | $N_0$ | T | WFR-FM (RK) | WFR-FM (EULER) | WFR-MFM | SPD-RK | SPD-EU |
|---|---|---|---|---|---|---|---|---|
| GENE | 2 | 400 | 4 | 2.178 $\pm 0.148$ | 0.370 $\pm 0.021$ | 0.0038 $\pm 0.0009$ | 573× | 97× |
| DYNGEN | 5 | 156 | 4 | 0.488 $\pm 0.049$ | 0.409 $\pm 0.039$ | 0.0039 $\pm 0.0010$ | 125× | 105× |
| GAUSSIAN | 1000 | 500 | 1 | 0.794 $\pm 0.087$ | 0.143 $\pm 0.018$ | 0.0013 $\pm 0.0006$ | 611× | 110× |
| EMT | 10 | 577 | 3 | 0.225 $\pm 0.018$ | 0.311 $\pm 0.025$ | 0.0029 $\pm 0.0009$ | 78× | 107× |
| EB | 50 | 2381 | 4 | 9.191 $\pm 0.505$ | 0.384 $\pm 0.021$ | 0.0031 $\pm 0.0013$ | 2965× | 124× |
| CITE | 50 | 7476 | 3 | 0.107 $\pm 0.006$ | 0.319 $\pm 0.008$ | 0.0028 $\pm 0.0006$ | 38× | 114× |
| MOUSE | 50 | 4638 | 2 | 3.440 $\pm 0.194$ | 0.198 $\pm 0.008$ | 0.0021 $\pm 0.0007$ | 1638× | 94× |

real biological datasets (Table 3 in Appendix D.4) using $\mathcal{W}_1$ and RME. Across synthetic datasets (Gene, Dyngen, Gaussian), WFR-MFM consistently ranks first or second in both metrics. On real datasets (EMT, EB, CITE, Mouse), evaluations on held-out time points show a similar pattern, with WFR-MFM achieving the best score on CITE and top-2 performance on the remaining datasets. Overall, these results demonstrate that WFR-MFM achieves fast inference without compromising accuracy. Figure 1 visualizes Gene and EMT datasets from two complementary perspectives: (a) individual cell trajectories with a fixed cell count and evolving weights, and (b) changes in cell abundance obtained by normalizing weights into probabilities and resampling.

**Smooth speed–accuracy trade-off (Q3).** We assess whether WFR-MFM enables a controllable balance between inference speed and accuracy by varying only the number of inference steps $K$ in Algorithm 1, without retraining the model. As $K$ increases, $\mathcal{W}_1$ and RME may exhibit minor fluctuations at small $K$ but decrease and stabilize overall (Fig. 2a), while runtime grows approximately linearly (Fig. 2b). Together, these results demonstrate that WFR-MFM offers a continuous speed–accuracy trade-off: one-step inference maximizes efficiency, while a small number of steps achieves accuracy comparable to full dynamic WFR at a fraction of its computational cost. Moreover, to facilitate practical usage, we provide an explicit adaptive stopping criterion for selecting the number of inference steps in Appendix C.

**Scalability to Large-Scale Datasets (Q4).** We evaluate WFR-MFM in a high-dimensional setting by comparing predictive accuracy and computational efficiency with multiple baseline methods, and further assess its inference scalability and practical potential on a large-scale synthetic perturbation benchmark. On the 100D EB dataset, we compare predictive accuracy ($\mathcal{W}_1$ distance) and computational efficiency (training time, inference time, and memory usage) across multiple methods (Figure 3). WFR-MFM achieves competitive prediction accuracy while significantly outper-

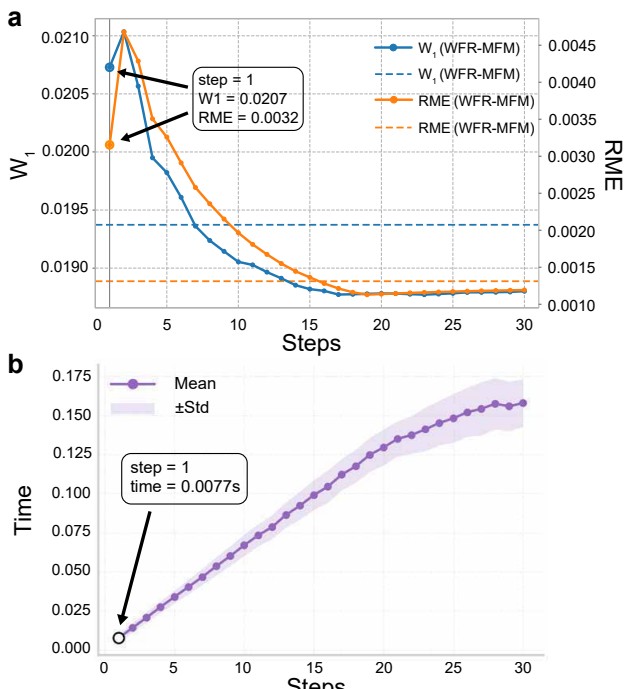

*Figure 2.* **Speed–accuracy trade-off (Q3) on the Simulation dataset.** (a) $\mathcal{W}_1$ and RME versus the number of inference steps. (b) Corresponding inference runtime versus the number of steps. For each step number, we perform 1000 independent runs.

forming existing methods in inference speed, and also ranks among the top in terms of training cost and memory usage. Additional visualizations and detailed quantitative results are provided in Appendix D.9.

In single-cell perturbation experiments, perturbation conditions grow combinatorially across modalities and experimental contexts (Klein et al., 2025), making scalable and rapid inference essential. To enable such large-scale perturbation studies within a single model, we condition the mean velocity field $v$ and rate field $h$ on a perturbation embedding $c$, i.e., $v(x, c, t, T)$ and $h(x, c, t, T)$. Details are provided in Appendix A.2, following strategies shown effective in prior work (Klein et al., 2025; Rohbeck et al., 2025). We evaluate

*Table 2.* Synthetic dataset performance under mean $\mathcal{W}_1$ and RME. RME is reported only for unbalanced methods. For stochastic inference methods, we report mean $\pm$ sd across five independent runs.

| METHOD | GENE (2D) | | DYNGEN (5D) | | GAUSSIAN (1000D) | |
|---|---|---|---|---|---|---|
| | $\mathcal{W}_1$ ($\downarrow$) | RME ($\downarrow$) | $\mathcal{W}_1$ ($\downarrow$) | RME ($\downarrow$) | $\mathcal{W}_1$ ($\downarrow$) | RME ($\downarrow$) |
| MMFM (ROHBECK ET AL., 2025) | 0.298 | — | 1.371 | — | 2.833 | — |
| METRIC FM (KAPUSNIAK ET AL., 2024) | 0.311 | — | 1.767 | — | 3.794 | — |
| SF2M (TONG ET AL., 2024B) | $0.224_{\pm0.007}$ | — | $1.277_{\pm0.017}$ | — | $3.543_{\pm0.002}$ | — |
| MIOFLOW (HUGUET ET AL., 2022) | 0.148 | — | 0.965 | — | 2.858 | — |
| TIGON (SHA ET AL., 2024) | 0.045 | 0.014 | 0.512 | 0.047 | 2.263 | 0.127 |
| DEEPRUOT (ZHANG ET AL., 2025B) | $0.043_{\pm0.002}$ | $0.017_{\pm0.001}$ | $0.623_{\pm0.032}$ | $0.065_{\pm0.011}$ | $3.785_{\pm0.009}$ | $0.303_{\pm0.070}$ |
| VAR-RUOT (SUN ET AL., 2025) | $0.079_{\pm0.003}$ | $0.008_{\pm0.002}$ | $0.522_{\pm0.008}$ | $0.177_{\pm0.007}$ | $2.813_{\pm0.004}$ | $0.041_{\pm0.006}$ |
| UOT-FM (EYRING ET AL., 2024) | 0.093 | 0.010 | 1.204 | 0.097 | 2.771 | 0.033 |
| VGFM (WANG ET AL., 2025) | 0.046 | 0.006 | 0.598 | 0.037 | 3.010 | 0.037 |
| WFR-FM (PENG ET AL., 2026A) | **0.019** | **0.001** | **0.135** | **0.005** | **2.233** | 0.044 |
| **WFR-MFM** | 0.021 | 0.003 | 0.176 | 0.023 | **2.233** | **0.001** |

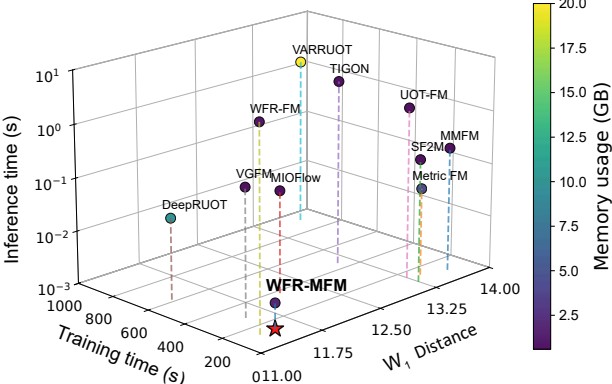

*Figure 3.* **Efficiency on the 100D EB dataset.** Methods are compared by $\mathcal{W}_1$ distance, training time, and inference time (log scale), with colors indicating GPU memory usage.

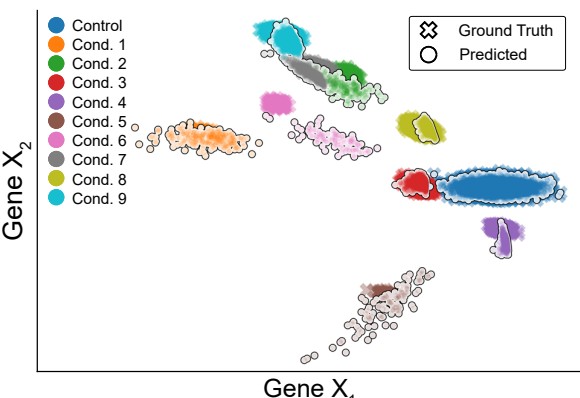

*Figure 4.* **Predictions on unseen perturbation conditions.** Gene expression distributions before and after perturbation for nine unseen conditions, visualized by resampling cells using predicted weights (control denotes the unperturbed baseline).

large-scale perturbation inference on a synthetic benchmark with 5100 perturbation conditions (Appendix D.13), using 100 for training and 5000 for testing. WFR-MFM completes inference for all 5000 unseen conditions on 10,000 cells in 6.55 seconds, whereas WFR-FM requires about 0.40 seconds for a representative perturbation condition, implying roughly two thousand seconds for full-scale inference. Despite this speedup, WFR-MFM maintains strong accuracy ($\mathcal{W}_1 = 0.115$, RME= 0.069), with predicted distributions closely matching ground truth (Figure 4). These results show that WFR-MFM generalizes from limited observed perturbations to thousands of unseen conditions, enabling large-scale perturbation response prediction.

## 6. Conclusion

We formulate a mean-flow framework for unbalanced FM that summarizes transport and mass-growth dynamics over arbitrary time intervals via mean velocity and mass-growth fields, enabling fast one-step generation without trajectory simulation. We then develop **WFR-MFM**, a mean flow

matching algorithm for dynamic unbalanced OT under the WFR geometry with simulation-free inference. Experiments on synthetic and real scRNA-seq datasets demonstrate orders-of-magnitude inference speedups while maintaining strong predictive accuracy, as well as a controllable speed–accuracy trade-off through multi-step consistency.

While WFR-MFM already achieves strong inference efficiency and accuracy, recent refinements to the mean-flow framework that improve stability and inference quality (Guo et al., 2025; Zhang et al., 2025a; Geng et al., 2025b) are not incorporated here and may further enhance its performance. Extending WFR-MFM to large-scale experimental perturbation datasets and validating its practical utility in real-world settings remain important directions for future work. Beyond the WFR geometry, the proposed mean-flow perspective may further extend to other unbalanced transport formulations with well-defined Dirac-to-Dirac flows.

## Acknowledgements

This work was supported by the National Key R&D Program of China (No. 2021YFA1003301 to T.L.) and the National Natural Science Foundation of China (NSFC Nos. 12288101 to T.L. & P.Z., 8206100646 and T2321001 to P.Z.). We acknowledge support from the High-performance Computing Platform of Peking University for computation.

## Impact Statement

This paper presents work whose goal is to advance the field of machine learning. There are many potential societal consequences of our work, none of which we feel must be specifically highlighted here.

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

# A. Additional Theoretical and Method Details

## A.1. Details of WFR-MFM

Most constructions in this subsection follow WFR-FM (Peng et al., 2026a); the mean-flow training objective is the main departure.

**WFR optimal transport.** The dynamic formulation of unbalanced OT endowed with the WFR metric (Chizat et al., 2018a;b; Liero et al., 2018) is

$$\text{WFR}_\delta^2(\mu_0, \mu_1) = \inf_{\rho, g, \boldsymbol{u}} \int_0^1 \int_{\mathcal{X}} \frac{1}{2}\big(\|\boldsymbol{u}(\boldsymbol{x}, t)\|_2^2 + \delta^2 \|g(\boldsymbol{x}, t)\|_2^2\big) \rho_t(\boldsymbol{x}) \, \mathrm{d}\boldsymbol{x} \, \mathrm{d}t \tag{4}$$
$$\text{s.t.} \quad \partial_t \rho + \nabla_{\boldsymbol{x}} \cdot (\rho \boldsymbol{u}) = \rho g, \quad \rho_0 = \mu_0, \ \rho_1 = \mu_1.$$

The closed-form WFR distance between two Dirac measures $m_0 \delta_{\boldsymbol{x}_0}$ and $m_1 \delta_{\boldsymbol{x}_1}$ is given by (Chizat et al., 2018a; Liero et al., 2018):

$$\text{WFR-DD}_\delta^2(m_0 \delta_{\boldsymbol{x}_0}, m_1 \delta_{\boldsymbol{x}_1}) = 2\delta^2 \Big(m_0 + m_1 - 2\sqrt{m_0 m_1} \, \overline{\cos}\big(\tfrac{\|\boldsymbol{x}_0 - \boldsymbol{x}_1\|_2}{2\delta}\big)\Big),$$

where $\overline{\cos}(\boldsymbol{x}) = \cos(\min\{\boldsymbol{x}, \frac{\pi}{2}\})$. When $\|\boldsymbol{x}_0 - \boldsymbol{x}_1\|_2 < \pi\delta$, the corresponding geodesic curve—i.e., the OT path, known as the traveling Dirac—is described by

$$m(t) = At^2 - 2Bt + m_0, \qquad \boldsymbol{u}(\boldsymbol{x}, t) \, m(t) = \boldsymbol{\omega}_0, \tag{5}$$

where $A, B, \boldsymbol{\omega}_0$ are

$$\begin{cases} A = m_0 + m_1 - 2\sqrt{\dfrac{m_0 m_1}{1 + \tau^2}}, \qquad B = m_0 - \sqrt{\dfrac{m_0 m_1}{1 + \tau^2}}, \\ \boldsymbol{\omega}_0 = 2\delta\tau \sqrt{\dfrac{m_0 m_1}{1 + \tau^2}} \, \boldsymbol{l}, \qquad \tau = \tan\big(\tfrac{\|\boldsymbol{x}_1 - \boldsymbol{x}_0\|_2}{2\delta}\big), \end{cases} \tag{6}$$

and $\boldsymbol{l}$ is the unit vector pointing from $\boldsymbol{x}_0$ to $\boldsymbol{x}_1$.

**Static semi-coupling form.** An equivalent static (Kantorovich) formulation of the WFR problem (4) is (Chizat et al., 2018b; Liero et al., 2018)

$$\text{WFR}_\delta^2(\mu_0, \mu_1) = \inf_{(\gamma_0, \gamma_1) \in (\mathcal{M}_+(\mathcal{X}^2))^2} \int_{\mathcal{X}^2} \text{WFR-DD}_\delta^2\big(\gamma_0(\boldsymbol{x}, \boldsymbol{y})\delta_{\boldsymbol{x}}, \ \gamma_1(\boldsymbol{x}, \boldsymbol{y})\delta_{\boldsymbol{y}}\big) \, \mathrm{d}\boldsymbol{x} \, \mathrm{d}\boldsymbol{y}, \tag{7}$$

subject to

$$\Gamma(\mu_0, \mu_1) \overset{\text{def.}}{=} \left\{ (\gamma_0, \gamma_1) \in (\mathcal{M}_+(\mathcal{X}^2))^2 : \int_{\mathcal{X}} \gamma_0(\boldsymbol{x}, \boldsymbol{y}) \, \mathrm{d}\boldsymbol{y} = \mu_0(\boldsymbol{x}), \ \int_{\mathcal{X}} \gamma_1(\boldsymbol{x}, \boldsymbol{y}) \, \mathrm{d}\boldsymbol{x} = \mu_1(\boldsymbol{y}) \right\}.$$

**Optimal Entropy Transport.** Equivalently, WFR can be cast as an Optimal Entropy Transport (OET) problem (Chizat et al., 2018b; Liero et al., 2018):

$$\text{WFR}_\delta^2(\mu_0, \mu_1) = 2\delta^2 \inf_{\gamma \in \mathcal{M}_+(\mathcal{X}^2)} \left\{ \int_{\mathcal{X}^2} \big[ -2 \ln \cos_+ \tfrac{\|\boldsymbol{x} - \boldsymbol{y}\|_2}{2\delta} \big] \gamma(\boldsymbol{x}, \boldsymbol{y}) \, \mathrm{d}\boldsymbol{x} \, \mathrm{d}\boldsymbol{y} \right.$$
$$\left. + \text{KL}\left( \int \gamma(\boldsymbol{x}, \boldsymbol{y}) \, \mathrm{d}\boldsymbol{y} \, \Big\| \, \mu_0(\boldsymbol{x}) \right) + \text{KL}\left( \int \gamma(\boldsymbol{x}, \boldsymbol{y}) \, \mathrm{d}\boldsymbol{x} \, \Big\| \, \mu_1(\boldsymbol{y}) \right) \right\}. \tag{8}$$

When $\gamma$ is the optimal coupling of Eq. (8), the semi-coupling

$$\gamma_0(\boldsymbol{x}, \boldsymbol{y}) = \frac{\gamma(\boldsymbol{x}, \boldsymbol{y})}{\int_{\mathcal{X}} \gamma(\boldsymbol{x}, \boldsymbol{z}) \, \mathrm{d}\boldsymbol{z}} \, \mu_0(\boldsymbol{x}), \qquad \gamma_1(\boldsymbol{x}, \boldsymbol{y}) = \frac{\gamma(\boldsymbol{x}, \boldsymbol{y})}{\int_{\mathcal{X}} \gamma(\boldsymbol{z}, \boldsymbol{y}) \, \mathrm{d}\boldsymbol{z}} \, \mu_1(\boldsymbol{y})$$

solves the static WFR problem in Eq. (7).

**Conditional measure path.**    Given $\boldsymbol{z} \sim q(\boldsymbol{z})$, we consider the decoupled conditional path

$$\rho_t(\boldsymbol{x} \mid \boldsymbol{z}) \;=\; m_t(\boldsymbol{z})\, \tilde{\rho}_t(\boldsymbol{x} \mid \boldsymbol{z}),$$

with conditional dynamics

$$\partial_t \rho_t(\boldsymbol{x} \mid \boldsymbol{z}) + \nabla_{\boldsymbol{x}} \cdot \big(\rho_t(\boldsymbol{x} \mid \boldsymbol{z})\, \boldsymbol{u}_t(\boldsymbol{x} \mid \boldsymbol{z})\big) = g_t(\boldsymbol{x} \mid \boldsymbol{z})\, \rho_t(\boldsymbol{x} \mid \boldsymbol{z}). \tag{9}$$

**Marginalization.**    Define the marginals by

$$\rho_t(\boldsymbol{x}) = \int \rho_t(\boldsymbol{x} \mid \boldsymbol{z})\, q(\boldsymbol{z})\, \mathrm{d}\boldsymbol{z}, \quad \boldsymbol{u}_t(\boldsymbol{x}) = \int \boldsymbol{u}_t(\boldsymbol{x} \mid \boldsymbol{z}) \tfrac{\rho_t(\boldsymbol{x}|\boldsymbol{z})q(\boldsymbol{z})}{\rho_t(\boldsymbol{x})}\, \mathrm{d}\boldsymbol{z}, \quad g_t(\boldsymbol{x}) = \int g_t(\boldsymbol{x} \mid \boldsymbol{z}) \tfrac{\rho_t(\boldsymbol{x}|\boldsymbol{z})q(\boldsymbol{z})}{\rho_t(\boldsymbol{x})}\, \mathrm{d}\boldsymbol{z}. \tag{10}$$

If $q(\boldsymbol{z})$ is independent of $(\boldsymbol{x}, t)$ and Eq. (9) holds for each $\boldsymbol{z}$, then Eq. (10) satisfies the continuity equation with source term

$$\partial_t \rho_t(\boldsymbol{x}) + \nabla_{\boldsymbol{x}} \cdot (\rho_t(\boldsymbol{x})\boldsymbol{u}_t(\boldsymbol{x})) = g_t(\boldsymbol{x})\, \rho_t(\boldsymbol{x}). \tag{11}$$

**Conditional Gaussian measure path.**    We take the conditional Gaussian measure path (CGMP)

$$\tilde{\rho}_t(\boldsymbol{x} \mid \boldsymbol{z}) = \mathcal{N}\big(\boldsymbol{x} \,\big|\, \boldsymbol{\eta}_t(\boldsymbol{z}),\, \boldsymbol{\sigma}_t^2(\boldsymbol{z})\big), \qquad \rho_t(\boldsymbol{x} \mid \boldsymbol{z}) = m_t(\boldsymbol{z})\, \tilde{\rho}_t(\boldsymbol{x} \mid \boldsymbol{z}), \tag{12}$$

with explicit conditional field and rate

$$\boldsymbol{u}_t(\boldsymbol{x} \mid \boldsymbol{z}) = \frac{\boldsymbol{\sigma}_t'(\boldsymbol{z})}{\boldsymbol{\sigma}_t(\boldsymbol{z})}\big(\boldsymbol{x} - \boldsymbol{\eta}_t(\boldsymbol{z})\big) + \boldsymbol{\eta}_t'(\boldsymbol{z}), \qquad g_t(\boldsymbol{x} \mid \boldsymbol{z}) = \partial_t \ln m_t(\boldsymbol{z}). \tag{13}$$

**WFR coupling solves the dynamic WFR problem.**    Consider two measures $\mu_0(\boldsymbol{x})$ and $\mu_1(\boldsymbol{x})$ as the source and target of a WFR problem, and denote by $(\gamma_0, \gamma_1)$ a semi-coupling solving the static form. Without loss of generality, let $\mu_0(\boldsymbol{x})$ be a probability density on $\mathcal{X}$, so that $\gamma_0(\boldsymbol{x}_0, \boldsymbol{x}_1)$ is a probability density on $\mathcal{X}^2$. Let $\boldsymbol{z} = (\boldsymbol{x}_0, \boldsymbol{x}_1) \sim \gamma_0(\boldsymbol{x}_0, \boldsymbol{x}_1)$ represent the source and target locations. Motivated by the traveling Dirac solution of the dynamic WFR problem (Chizat et al., 2018a) and following the construction in WFR-FM (Peng et al., 2026a), we adopt a special CGMP, termed the traveling Gaussian.

$$\begin{aligned}
\tilde{\rho}_t(\boldsymbol{x} \mid \boldsymbol{x}_0, \boldsymbol{x}_1) &= \mathcal{N}\Big(\boldsymbol{x} \,\Big|\, \boldsymbol{x}_0 + \boldsymbol{\omega}_0 \int_0^t \frac{\mathrm{d}s}{m_s(\boldsymbol{x}_0, \boldsymbol{x}_1)},\, \sigma^2 \mathbf{I}\Big) := \mathcal{N}\big(\boldsymbol{x} \,\big|\, \boldsymbol{x}_0 + \boldsymbol{\omega}_0 \Lambda_t(\boldsymbol{x}_0, \boldsymbol{x}_1),\, \sigma^2 \mathbf{I}\big) \\
&= \mathcal{N}\Big(\boldsymbol{x} \,\Big|\, \boldsymbol{x}_0 + \frac{\boldsymbol{\omega}_0}{\sqrt{m_0 A - B^2}}\Big(\arctan \frac{At - B}{\sqrt{m_0 A - B^2}} - \arctan \frac{-B}{\sqrt{m_0 A - B^2}}\Big),\, \sigma^2 \mathbf{I}\Big),
\end{aligned} \tag{14}$$

where $\boldsymbol{\omega}_0$ and $m_t(\boldsymbol{x}_0, \boldsymbol{x}_1)$ are defined by Eqs. (5) and (6), with boundary condition $m_0(\boldsymbol{x}_0, \boldsymbol{x}_1) = 1$ and $m_1(\boldsymbol{x}_0, \boldsymbol{x}_1) = \frac{\gamma_1(\boldsymbol{x}_0, \boldsymbol{x}_1)}{\gamma_0(\boldsymbol{x}_0, \boldsymbol{x}_1)}$. Under this construction, the marginal boundary measures converge to $\mu_0$ and $\mu_1$ respectively as $\sigma \to 0$, and the induced measure path follows the WFR geodesic.

**Training of WFR-MFM.**    Under the WFR geometry, the mean-flow framework yields the following training losses for WFR-MFM. The mean velocity and mass-growth fields $(\boldsymbol{v}, h)$ are parameterized by neural networks $(\boldsymbol{v}_{\boldsymbol{\theta}}, h_{\boldsymbol{\phi}})$. The unbalanced mean flow matching objective is given by

$$\mathcal{L}(\boldsymbol{\theta}, \boldsymbol{\phi}) = \mathbb{E}_{t<T,\, \boldsymbol{x} \sim \rho_t(\boldsymbol{x})}\Big[\|\boldsymbol{v}_{\boldsymbol{\theta}}(\boldsymbol{x}, t, T) - \mathrm{sg}(\boldsymbol{v}(\boldsymbol{x}, t, T))\|_2^2 + \lambda\, \|h_{\boldsymbol{\phi}}(\boldsymbol{x}, t, T) - \mathrm{sg}(h(\boldsymbol{x}, t, T))\|_2^2\Big],$$

where $\lambda > 0$ balances the supervision of the velocity and mass-growth fields, and $\mathrm{sg}(\cdot)$ is the stop-gradient operator. The regression targets $(\boldsymbol{v}, h)$ are computed from the mean-field derivative identities:

$$\begin{aligned}
\boldsymbol{v}(\boldsymbol{x}, t, T) &= \boldsymbol{u}_t(\boldsymbol{x}) + (T - t)\Big[\partial_t \boldsymbol{v}_{\boldsymbol{\theta}}(\boldsymbol{x}, t, T) + (\nabla_{\boldsymbol{x}} \boldsymbol{v}_{\boldsymbol{\theta}}(\boldsymbol{x}, t, T))\, \boldsymbol{u}_t(\boldsymbol{x})\Big], \\
h(\boldsymbol{x}, t, T) &= g_t(\boldsymbol{x}) + (T - t)\Big[\partial_t h_{\boldsymbol{\phi}}(\boldsymbol{x}, t, T) + \nabla_{\boldsymbol{x}} h_{\boldsymbol{\phi}}(\boldsymbol{x}, t, T) \cdot \boldsymbol{u}_t(\boldsymbol{x})\Big].
\end{aligned} \tag{15}$$

For conditional training, the corresponding objective takes the form

$$\mathcal{L}_{\mathrm{c}}(\boldsymbol{\theta}, \boldsymbol{\phi}) = \mathbb{E}_{t<T, \boldsymbol{z} \sim q(\boldsymbol{z}), \boldsymbol{x} \sim \tilde{\rho}_t(\boldsymbol{x}|\boldsymbol{z})}\Big[\|\boldsymbol{v}_{\boldsymbol{\theta}}(\boldsymbol{x}, t, T) - \mathrm{sg}(\boldsymbol{v}(\boldsymbol{x}, t, T \mid \boldsymbol{z}))\|_2^2 + \lambda\, \|h_{\boldsymbol{\phi}}(\boldsymbol{x}, t, T) - \mathrm{sg}(h(\boldsymbol{x}, t, T \mid \boldsymbol{z}))\|_2^2\Big] m_t(\boldsymbol{z}),$$

where the conditional regression targets are defined as

$$\boldsymbol{v}(\boldsymbol{x}, t, T \mid \boldsymbol{z}) = \boldsymbol{u}_t(\boldsymbol{x} \mid \boldsymbol{z}) + (T - t)\left[\partial_t \boldsymbol{v_\theta}(\boldsymbol{x}, t, T) + (\nabla_{\boldsymbol{x}} \boldsymbol{v_\theta}(\boldsymbol{x}, t, T))\,\boldsymbol{u}_t(\boldsymbol{x} \mid \boldsymbol{z})\right],$$

$$h(\boldsymbol{x}, t, T \mid \boldsymbol{z}) = g_t(\boldsymbol{x} \mid \boldsymbol{z}) + (T - t)\left[\partial_t h_{\boldsymbol{\phi}}(\boldsymbol{x}, t, T) + \nabla_{\boldsymbol{x}} h_{\boldsymbol{\phi}}(\boldsymbol{x}, t, T) \cdot \boldsymbol{u}_t(\boldsymbol{x} \mid \boldsymbol{z})\right].$$

The conditional targets $\boldsymbol{v}(\boldsymbol{x}, t, T \mid \boldsymbol{z})$ and $h(\boldsymbol{x}, t, T \mid \boldsymbol{z})$ are defined analogously using the conditional instantaneous fields $\boldsymbol{u}_t(\boldsymbol{x} \mid \boldsymbol{z})$ and $g_t(\boldsymbol{x} \mid \boldsymbol{z})$.

### A.2. Conditional Extension of WFR-MFM for Perturbations

Many biological datasets involve multiple perturbation conditions, where a common control distribution $\mu_0$ is mapped to a family of perturbed endpoints $\{\mu_1^c\}_{c \in \mathcal{C}}$, with $c$ indexing the perturbation identity. Under the WFR formula, this perturbation induces condition-dependent transport and mass-growth dynamics, resulting in conditional fields $\boldsymbol{u}_t(\boldsymbol{x}, \boldsymbol{c})$ and $g_t(\boldsymbol{x}, \boldsymbol{c})$.

We extend the mean-flow formulation to this perturbation setting by conditioning the mean velocity and mean mass-growth fields on a perturbation embedding $\boldsymbol{c}$, defined as

$$\boldsymbol{v}(\boldsymbol{x}, \boldsymbol{c}, t, T) \;=\; \frac{1}{T - t}\int_t^T \boldsymbol{u}_\tau(\boldsymbol{x}_\tau, \boldsymbol{c})\,\mathrm{d}\tau, \quad h(\boldsymbol{x}, \boldsymbol{c}, t, T) \;=\; \frac{1}{T - t}\int_t^T g_\tau(\boldsymbol{x}_\tau, \boldsymbol{c})\,\mathrm{d}\tau.$$

With this extension, we parameterize the conditional mean fields by neural networks $\boldsymbol{v_\theta}(\boldsymbol{x}, \boldsymbol{c}, t, T)$ and $h_{\boldsymbol{\phi}}(\boldsymbol{x}, \boldsymbol{c}, t, T)$, and define the conditional training objective for learning perturbation-dependent mean fields as

$$\mathcal{L}_{\mathrm{c}}(\boldsymbol{\theta}, \boldsymbol{\phi}) = \mathbb{E}_{t < T, \boldsymbol{c}, \boldsymbol{z}, \boldsymbol{x}}\left[\|\boldsymbol{v_\theta}(\boldsymbol{x}, \boldsymbol{c}, t, T) - \mathrm{sg}(\boldsymbol{v}(\boldsymbol{x}, \boldsymbol{c}, t, T \mid \boldsymbol{z}))\|_2^2 + \lambda\,\|h_{\boldsymbol{\phi}}(\boldsymbol{x}, \boldsymbol{c}, t, T) - \mathrm{sg}(h(\boldsymbol{x}, \boldsymbol{c}, t, T \mid \boldsymbol{z}))\|_2^2\right] m_t(\boldsymbol{z}),$$

where

$$\begin{aligned}
\boldsymbol{v}(\boldsymbol{x}, \boldsymbol{c}, t, T \mid \boldsymbol{z}) &= \boldsymbol{u}_t(\boldsymbol{x}, \boldsymbol{c} \mid \boldsymbol{z}) + (T - t)[\partial_t \boldsymbol{v_\theta}(\boldsymbol{x}, \boldsymbol{c}, t, T) + (\nabla_{\boldsymbol{x}} \boldsymbol{v_\theta}(\boldsymbol{x}, \boldsymbol{c}, t, T))\,\boldsymbol{u}_t(\boldsymbol{x}, \boldsymbol{c} \mid \boldsymbol{z})], \\
h(\boldsymbol{x}, \boldsymbol{c}, t, T \mid \boldsymbol{z}) &= g_t(\boldsymbol{x}, \boldsymbol{c} \mid \boldsymbol{z}) + (T - t)[\partial_t h_{\boldsymbol{\phi}}(\boldsymbol{x}, \boldsymbol{c}, t, T) + \nabla_{\boldsymbol{x}} h_{\boldsymbol{\phi}}(\boldsymbol{x}, \boldsymbol{c}, t, T) \cdot \boldsymbol{u}_t(\boldsymbol{x}, \boldsymbol{c} \mid \boldsymbol{z})].
\end{aligned} \tag{16}$$

Intuitively, after conditioning on the perturbation embedding $\boldsymbol{c}$, both one-step and multi-step inference follow the same update rule as in the base model; only the underlying mean fields are altered in a perturbation-specific manner. This enables WFR-MFM to support large perturbation collections within a single unified model, instead of training separate models for different perturbations. The complete training procedure is summarized in Algorithm 3.

## B. Proofs

### B.1. Proof of Theorem 4.1

**Theorem 4.1.** *If $\rho_t(\boldsymbol{x}) > 0$ for all $\boldsymbol{x} \in \mathcal{X}$ and $t \in [0, 1]$, and $q(\boldsymbol{z})$ is independent of $(\boldsymbol{x}, t, T)$, then*

$$\mathcal{L}(\boldsymbol{\theta}, \boldsymbol{\phi}) = \mathcal{L}_{\mathrm{c}}(\boldsymbol{\theta}, \boldsymbol{\phi}) + C, \tag{17}$$

*for some constant $C$ independent of $(\boldsymbol{\theta}, \boldsymbol{\phi})$. Consequently,*

$$\nabla_{\boldsymbol{\theta}, \boldsymbol{\phi}}\mathcal{L}(\boldsymbol{\theta}, \boldsymbol{\phi}) = \nabla_{\boldsymbol{\theta}, \boldsymbol{\phi}}\mathcal{L}_{\mathrm{c}}(\boldsymbol{\theta}, \boldsymbol{\phi}).$$

*Proof.* We first restate the unconditional and conditional mean flow matching losses. The **unconditional mean flow matching loss** is defined as

$$\mathcal{L}(\boldsymbol{\theta}, \boldsymbol{\phi}) = \mathbb{E}_{t < T, \boldsymbol{x} \sim \rho_t(\boldsymbol{x})}\left[\|\boldsymbol{v_\theta}(\boldsymbol{x}, t, T) - \mathrm{sg}(\boldsymbol{v}(\boldsymbol{x}, t, T))\|_2^2 + \lambda\,\|h_{\boldsymbol{\phi}}(\boldsymbol{x}, t, T) - \mathrm{sg}(h(\boldsymbol{x}, t, T))\|_2^2\right],$$

where

$$\begin{aligned}
\boldsymbol{v}(\boldsymbol{x}, t, T) &= \boldsymbol{u}_t(\boldsymbol{x}) + (T - t)[\partial_t \boldsymbol{v_\theta}(\boldsymbol{x}, t, T) + (\nabla_{\boldsymbol{x}} \boldsymbol{v_\theta}(\boldsymbol{x}, t, T))\,\boldsymbol{u}_t(\boldsymbol{x})], \\
h(\boldsymbol{x}, t, T) &= g_t(\boldsymbol{x}) + (T - t)[\partial_t h_{\boldsymbol{\phi}}(\boldsymbol{x}, t, T) + \nabla_{\boldsymbol{x}} h_{\boldsymbol{\phi}}(\boldsymbol{x}, t, T) \cdot \boldsymbol{u}_t(\boldsymbol{x})].
\end{aligned} \tag{18}$$

---

**Algorithm 3** Conditional WFR-MFM Training for Perturbations

---

**Input:** Perturbation set $\mathcal{C}$; Control distribution $\mu_0$; perturbed distributions $\{\mu_1^c\}_{c \in \mathcal{C}}$; bandwidth $\sigma$; OET batch size $B$; training batch size $b$; WFR penalty $\{\delta^c\}_{c \in \mathcal{C}}$; condition embeddings $\{c\}_{c \in \mathcal{C}}$; condition batch size $B_c$; mean fields $\boldsymbol{v}_{\boldsymbol{\theta}}(\boldsymbol{x}, \boldsymbol{c}, t, T)$ and $h_{\boldsymbol{\phi}}(\boldsymbol{x}, \boldsymbol{c}, t, T)$
**Output:** trained $\boldsymbol{v}_{\boldsymbol{\theta}}$ and $h_{\boldsymbol{\phi}}$
**Precompute couplings**
**for** $c \in \mathcal{C}$ **do**
   $\gamma^{(c)} \leftarrow \text{OET}(\mu_0, \mu_1^{(c)})$
   $\gamma_0^{(c)}(\boldsymbol{x}, \boldsymbol{y}) \leftarrow \dfrac{\gamma^{(c)}(\boldsymbol{x}, \boldsymbol{y})}{\int \gamma^{(c)}(\boldsymbol{x}, \boldsymbol{z})\,\mathrm{d}\boldsymbol{z}}\,\mu_0(\boldsymbol{x})$
   Store $\gamma_0^{(c)}$ in memory
**end for**
**while** Training **do**
   Sample perturbation minibatch $\mathcal{S} \subset \mathcal{C}$ with $|\mathcal{S}| = B_c$
   **for** $c \in \mathcal{S}$ **do**
      Retrieve $\gamma_0^{(c)}$ and embedding $\boldsymbol{c}$
      Sample $b$ pairs $(\boldsymbol{x}_0, \boldsymbol{x}_1) \sim \gamma_0^{(c)}$
      Compute travelling-WFR quantities $(A^{(c)}, B^{(c)}, \boldsymbol{\omega}_0^{(c)}, \tau^{(c)}, m_t^{(c)})$
      Sample $(t, T)$
      $\boldsymbol{\eta}_t \leftarrow \boldsymbol{x}_0 + \boldsymbol{\omega}_0^{(c)}\,\Lambda_t(\boldsymbol{x}_0, \boldsymbol{x}_1)$
      Sample $\boldsymbol{x} \sim \mathcal{N}(\boldsymbol{\eta}_t, \sigma^2 \mathbf{I})$
      $\boldsymbol{u} \leftarrow \boldsymbol{\omega}_0^{(c)}/m_t^{(c)}(\boldsymbol{x}_0, \boldsymbol{x}_1)$
      $g \leftarrow \frac{d}{dt} \ln m_t^{(c)}(\boldsymbol{x}_0, \boldsymbol{x}_1)$
      $m \leftarrow m_t^{(c)}(\boldsymbol{x}_0, \boldsymbol{x}_1)/\gamma_0^{(c)}(\boldsymbol{x}_0, \boldsymbol{x}_1)$
      $\boldsymbol{v}^{(c)} \leftarrow \boldsymbol{u} + \text{sg}((T{-}t)\,[\partial_t \boldsymbol{v}_{\boldsymbol{\theta}}(\boldsymbol{x}, \boldsymbol{c}, t, T) + (\nabla_{\boldsymbol{x}} \boldsymbol{v}_{\boldsymbol{\theta}}(\boldsymbol{x}, \boldsymbol{c}, t, T))\boldsymbol{u}])$
      $h^{(c)} \leftarrow g + \text{sg}((T{-}t)\,[\partial_t h_{\boldsymbol{\phi}}(\boldsymbol{x}, \boldsymbol{c}, t, T) + \nabla_{\boldsymbol{x}} h_{\boldsymbol{\phi}}(\boldsymbol{x}, \boldsymbol{c}, t, T)\cdot\boldsymbol{u}])$
      Collate samples into batched tensors $\{\boldsymbol{x}^b, t^b, T^b, \boldsymbol{v}^b, h^b, m^b\}$.
      $\mathcal{L}_c(\boldsymbol{\theta}, \boldsymbol{\phi}) \leftarrow \left( \|\boldsymbol{v}_{\boldsymbol{\theta}}(\boldsymbol{x}^b, \boldsymbol{c}, t^b, T^b) - \boldsymbol{v}^b\|_2^2 + \lambda\,\|h_{\boldsymbol{\phi}}(\boldsymbol{x}^b, \boldsymbol{c}, t^b, T^b) - h^b\|_2^2 \right) m^b$
   **end for**
   $\mathcal{L} \leftarrow \frac{1}{B_c} \sum_{c \in \mathcal{S}} \mathcal{L}_c$
   Update $(\boldsymbol{\theta}, \boldsymbol{\phi})$ using $\nabla \mathcal{L}$
**end while**
**return** $\boldsymbol{v}_{\boldsymbol{\theta}}, h_{\boldsymbol{\phi}}$

---

The corresponding **conditional mean flow matching loss** is

$$\mathcal{L}_{\text{c}}(\boldsymbol{\theta}, \boldsymbol{\phi}) = \mathbb{E}_{t < T, \boldsymbol{z} \sim q(\boldsymbol{z}), \boldsymbol{x} \sim \rho_t(\boldsymbol{x})} \left[ \|\boldsymbol{v}_{\boldsymbol{\theta}}(\boldsymbol{x}, t, T) - \text{sg}(\boldsymbol{v}(\boldsymbol{x}, t, T \mid \boldsymbol{z}))\|_2^2 + \lambda\,\|h_{\boldsymbol{\phi}}(\boldsymbol{x}, t, T) - \text{sg}(h(\boldsymbol{x}, t, T \mid \boldsymbol{z}))\|_2^2 \right] m_t(\boldsymbol{z}),$$

where $\text{sg}(\cdot)$ denotes the stop-gradient operator, preventing gradient flow through the target functions. The conditional targets are defined via derivative identities:

$$\boldsymbol{v}(\boldsymbol{x}, t, T \mid \boldsymbol{z}) = \boldsymbol{u}_t(\boldsymbol{x} \mid \boldsymbol{z}) + (T - t)[\partial_t \boldsymbol{v}_{\boldsymbol{\theta}}(\boldsymbol{x}, t, T) + (\nabla_{\boldsymbol{x}} \boldsymbol{v}_{\boldsymbol{\theta}}(\boldsymbol{x}, t, T))\,\boldsymbol{u}_t(\boldsymbol{x} \mid \boldsymbol{z})],$$

$$h(\boldsymbol{x}, t, T \mid \boldsymbol{z}) = g_t(\boldsymbol{x} \mid \boldsymbol{z}) + (T - t)[\partial_t h_{\boldsymbol{\phi}}(\boldsymbol{x}, t, T) + \nabla_{\boldsymbol{x}} h_{\boldsymbol{\phi}}(\boldsymbol{x}, t, T) \cdot \boldsymbol{u}_t(\boldsymbol{x} \mid \boldsymbol{z})].$$

We prove Eq. (17) in 5 steps.

**Step 1. Expansion of the unconditional $\boldsymbol{v}$-related loss.** The component of $\mathcal{L}(\boldsymbol{\theta}, \boldsymbol{\phi})$ associated with $\boldsymbol{v}$ can be expanded as

$$\mathbb{E}_{t < T, \boldsymbol{x} \sim \rho_t(\boldsymbol{x})} \big\| \boldsymbol{v}_{\boldsymbol{\theta}}(\boldsymbol{x}, t, T) - \text{sg}(\boldsymbol{v}(\boldsymbol{x}, t, T)) \big\|_2^2$$

$$= \mathbb{E}_{t < T, \boldsymbol{x} \sim \rho_t(\boldsymbol{x})} \left[ \|\boldsymbol{v}_{\boldsymbol{\theta}}(\boldsymbol{x}, t, T)\|_2^2 - 2\langle \boldsymbol{v}_{\boldsymbol{\theta}}(\boldsymbol{x}, t, T),\, \text{sg}(\boldsymbol{v}(\boldsymbol{x}, t, T)) \rangle + \|\text{sg}(\boldsymbol{v}(\boldsymbol{x}, t, T))\|_2^2 \right] \tag{19}$$

$$= \mathbb{E}_{t < T, \boldsymbol{x} \sim \rho_t(\boldsymbol{x})} \|\boldsymbol{v}_{\boldsymbol{\theta}}(\boldsymbol{x}, t, T)\|_2^2 - 2\,\mathbb{E}_{t < T, \boldsymbol{x} \sim \rho_t(\boldsymbol{x})} \langle \boldsymbol{v}_{\boldsymbol{\theta}}(\boldsymbol{x}, t, T),\, \text{sg}(\boldsymbol{v}(\boldsymbol{x}, t, T)) \rangle + \tilde{C}_1,$$

where $\tilde{C}_1 = \mathbb{E}_{t<T,\, \boldsymbol{x}\sim\rho_t(\boldsymbol{x})}\|\mathrm{sg}(\boldsymbol{v}(\boldsymbol{x},t,T))\|_2^2$ is constant with respect to $\boldsymbol{\theta}$.

**Step 2. Transformation of the cross term.** We next analyze the cross term in Eq. (19):

$$
\mathbb{E}_{t<T,\, \boldsymbol{x}\sim\rho_t(\boldsymbol{x})}\big\langle \boldsymbol{v_\theta}(\boldsymbol{x},t,T),\, \mathrm{sg}(\boldsymbol{v}(\boldsymbol{x},t,T))\big\rangle
$$

$$
\overset{(1)}{=} \mathbb{E}_{t<T,\, \boldsymbol{x}\sim\rho_t(\boldsymbol{x})}\Big\langle \boldsymbol{v_\theta}(\boldsymbol{x},t,T),\, \mathrm{sg}(\boldsymbol{u}_t(\boldsymbol{x}) + (T-t)[\partial_t \boldsymbol{v_\theta}(\boldsymbol{x},t,T) + (\nabla_{\boldsymbol{x}}\boldsymbol{v_\theta}(\boldsymbol{x},t,T))\,\boldsymbol{u}_t(\boldsymbol{x})])\Big\rangle
$$

$$
= \mathbb{E}_{t<T}\int_{\mathcal{X}}\Big\langle \boldsymbol{v_\theta}(\boldsymbol{x},t,T),\, \mathrm{sg}(\boldsymbol{u}_t(\boldsymbol{x})\rho_t(\boldsymbol{x}) + (T-t)[\partial_t \boldsymbol{v_\theta}(\boldsymbol{x},t,T)\rho_t(\boldsymbol{x}) + (\nabla_{\boldsymbol{x}}\boldsymbol{v_\theta}(\boldsymbol{x},t,T))\,\boldsymbol{u}_t(\boldsymbol{x})\rho_t(\boldsymbol{x})])\Big\rangle d\boldsymbol{x}
$$

$$
\overset{(2)}{=} \mathbb{E}_{t<T}\int_{\mathcal{X}}\int_{\mathcal{Z}}\Big\langle \boldsymbol{v_\theta}(\boldsymbol{x},t,T),\, \mathrm{sg}\big(\boldsymbol{u}_t(\boldsymbol{x}\,|\,\boldsymbol{z}) + (T-t)[\partial_t \boldsymbol{v_\theta}(\boldsymbol{x},t,T) + (\nabla_{\boldsymbol{x}}\boldsymbol{v_\theta}(\boldsymbol{x},t,T))\,\boldsymbol{u}_t(\boldsymbol{x}\,|\,\boldsymbol{z})]\big)\Big\rangle \rho_t(\boldsymbol{x}\,|\,\boldsymbol{z})q(\boldsymbol{z})\,d\boldsymbol{z}\,d\boldsymbol{x}
$$

$$
= \mathbb{E}_{t<T,\, \boldsymbol{z}\sim q(\boldsymbol{z}),\, \boldsymbol{x}\sim\rho_t(\boldsymbol{x}|\boldsymbol{z})}\big\langle \boldsymbol{v_\theta}(\boldsymbol{x},t,T),\, \mathrm{sg}(\boldsymbol{v}(\boldsymbol{x},t,T\,|\,\boldsymbol{z}))\big\rangle,
\tag{20}
$$

where (1) applies the definition of $\boldsymbol{v}(\boldsymbol{x},t,T)$ from Eq. (18), and (2) uses the marginal flow velocity

$$
\boldsymbol{u}_t(\boldsymbol{x}) = \frac{\displaystyle\int \boldsymbol{u}_t(\boldsymbol{x}\mid\boldsymbol{z})\,\rho_t(\boldsymbol{x}\mid\boldsymbol{z})\,q(\boldsymbol{z})\,d\boldsymbol{z}}{\rho_t(\boldsymbol{x})}
$$

and

$$
\rho_t(\boldsymbol{x}) = \int \rho_t(\boldsymbol{x}|\boldsymbol{z})q(\boldsymbol{z})\mathrm{d}\boldsymbol{z}.
$$

**Step 3. Conversion to the conditional expectation.** Substituting Eq. (20) into Eq. (19), we obtain

$$
\mathbb{E}_{t<T,\, \boldsymbol{x}\sim\rho_t(\boldsymbol{x})}\big\|\boldsymbol{v_\theta}(\boldsymbol{x},t,T) - \mathrm{sg}(\boldsymbol{v}(\boldsymbol{x},t,T))\big\|_2^2
$$

$$
= \mathbb{E}_{t<T,\, \boldsymbol{z}\sim q(\boldsymbol{z}),\, \boldsymbol{x}\sim\rho_t(\boldsymbol{x}|\boldsymbol{z})}\|\boldsymbol{v_\theta}(\boldsymbol{x},t,T)\|_2^2 - 2\,\mathbb{E}_{t<T,\, \boldsymbol{z}\sim q(\boldsymbol{z}),\, \boldsymbol{x}\sim\rho_t(\boldsymbol{x}|\boldsymbol{z})}\big\langle \boldsymbol{v_\theta}(\boldsymbol{x},t,T),\, \mathrm{sg}(\boldsymbol{v}(\boldsymbol{x},t,T\,|\,\boldsymbol{z}))\big\rangle + \tilde{C}_1
$$

$$
= \mathbb{E}_{t<T,\, \boldsymbol{z}\sim q(\boldsymbol{z}),\, \boldsymbol{x}\sim\rho_t(\boldsymbol{x}|\boldsymbol{z})}\Big[\|\boldsymbol{v_\theta}(\boldsymbol{x},t,T)\|_2^2 - 2\big\langle \boldsymbol{v_\theta}(\boldsymbol{x},t,T),\, \mathrm{sg}(\boldsymbol{v}(\boldsymbol{x},t,T\,|\,\boldsymbol{z}))\big\rangle\Big] + \tilde{C}_1
\tag{21}
$$

$$
= \mathbb{E}_{t<T,\, \boldsymbol{z}\sim q(\boldsymbol{z}),\, \boldsymbol{x}\sim\rho_t(\boldsymbol{x}|\boldsymbol{z})}\big\|\boldsymbol{v_\theta}(\boldsymbol{x},t,T) - \mathrm{sg}(\boldsymbol{v}(\boldsymbol{x},t,T\,|\,\boldsymbol{z}))\big\|_2^2 + C_1,
$$

where $C_1$ is a constant independent of $(\boldsymbol{\theta}, \boldsymbol{\phi})$.

**Step 4. Extension to the mass-growth rate field.** A similar derivation holds for the $h$-term:

$$
\mathbb{E}_{t<T,\, \boldsymbol{x}\sim\rho_t(\boldsymbol{x})}\lambda\,\|h_{\boldsymbol{\phi}}(\boldsymbol{x},t,T) - \mathrm{sg}(h(\boldsymbol{x},t,T))\|_2^2 = \mathbb{E}_{t<T,\, \boldsymbol{z}\sim q(\boldsymbol{z}),\, \boldsymbol{x}\sim\rho_t(\boldsymbol{x}|\boldsymbol{z})}\lambda\,\|h_{\boldsymbol{\phi}}(\boldsymbol{x},t,T) - \mathrm{sg}(h(\boldsymbol{x},t,T\,|\,\boldsymbol{z}))\|_2^2 + C_2.
\tag{22}
$$

**Step 5. Conclusion.** Combining Eqs. (21) and (22), we conclude that

$$
\mathcal{L}(\boldsymbol{\theta},\boldsymbol{\phi}) = \mathcal{L}_{\mathrm{c}}(\boldsymbol{\theta},\boldsymbol{\phi}) + C,
$$

for some constant $C$ independent of $(\boldsymbol{\theta},\boldsymbol{\phi})$. Hence,

$$
\nabla_{\boldsymbol{\theta},\boldsymbol{\phi}}\mathcal{L}(\boldsymbol{\theta},\boldsymbol{\phi}) = \nabla_{\boldsymbol{\theta},\boldsymbol{\phi}}\mathcal{L}_{\mathrm{c}}(\boldsymbol{\theta},\boldsymbol{\phi}),
$$

which completes the proof. $\square$

### B.2. Failure of Time-Averaged Conditional Mean Fields

This appendix shows that redefining the learning objective by direct time averaging along trajectories does *not* preserve the equivalence between the unconditional and conditional objectives established in Theorem 4.1. Concretely, consider the

following alternative definitions of the mean-field targets:

$$\boldsymbol{v}(\boldsymbol{x}, t, T) = \frac{1}{T-t} \int_t^T \boldsymbol{u}_\tau(\boldsymbol{x}_\tau) \, d\tau, \quad h(\boldsymbol{x}, t, T) = \frac{1}{T-t} \int_t^T g_\tau(\boldsymbol{x}_\tau) \, d\tau,$$

$$\boldsymbol{v}(\boldsymbol{x}, t, T \mid \boldsymbol{z}) = \frac{1}{T-t} \int_t^T \boldsymbol{u}_\tau(\boldsymbol{x}_\tau \mid \boldsymbol{z}) \, d\tau, \quad h(\boldsymbol{x}, t, T \mid \boldsymbol{z}) = \frac{1}{T-t} \int_t^T g_\tau(\boldsymbol{x}_\tau \mid \boldsymbol{z}) \, d\tau.$$
(23)

**Proposition B.1** (Failure of equivalence under time-averaged targets). *Consider the alternative (time-averaged) targets defined in Eq. (23). Then, in general, the equivalence in Theorem 4.1 fails: there exist a coupling distribution $q(\boldsymbol{z})$ and conditional measure paths $\rho_t(\cdot \mid \boldsymbol{z})$ such that there is no constant $C$ independent of $(\boldsymbol{\theta}, \boldsymbol{\phi})$ for which*

$$\mathcal{L}(\boldsymbol{\theta}, \boldsymbol{\phi}) = \mathcal{L}_{\mathrm{c}}(\boldsymbol{\theta}, \boldsymbol{\phi}) + C.$$

*Consequently, the minimization problems induced by $\mathcal{L}_{\mathrm{c}}$ and $\mathcal{L}$ are not equivalent in general.*

*Proof.* We begin by stating the definitions of the unconditional loss and the corresponding conditional loss.

$$\mathcal{L}(\boldsymbol{\theta}, \boldsymbol{\phi}) = \mathbb{E}_{t<T, \, \boldsymbol{x} \sim \rho_t(\boldsymbol{x})} \Big[ \|\boldsymbol{v}_{\boldsymbol{\theta}}(\boldsymbol{x}, t, T) - \boldsymbol{v}(\boldsymbol{x}, t, T)\|_2^2 + \lambda \|h_{\boldsymbol{\phi}}(\boldsymbol{x}, t, T) - h(\boldsymbol{x}, t, T)\|_2^2 \Big],$$

$$\mathcal{L}_{\mathrm{c}}(\boldsymbol{\theta}, \boldsymbol{\phi}) = \mathbb{E}_{t<T, \boldsymbol{z}, \boldsymbol{x} \sim \rho_t(\boldsymbol{x}|\boldsymbol{z})} \Big[ \|\boldsymbol{v}_{\boldsymbol{\theta}}(\boldsymbol{x}, t, T) - \boldsymbol{v}(\boldsymbol{x}, t, T \mid \boldsymbol{z})\|_2^2 + \lambda \|h_{\boldsymbol{\phi}}(\boldsymbol{x}, t, T) - h(\boldsymbol{x}, t, T \mid \boldsymbol{z})\|_2^2 \Big].$$

Without loss of generality, we consider the velocity field $\boldsymbol{v}$; the treatment of $h$ is analogous. The inequivalence between $\mathcal{L}$ and $\mathcal{L}_{\mathrm{c}}$ is already reflected in the cross terms related to the velocity field $\boldsymbol{v}$. All remaining terms can be treated analogously to the proof of Appendix B.1 and are therefore omitted.

We first consider the cross term appearing in the unconditional loss:

$$\mathbb{E}_{t<T, \, \boldsymbol{x} \sim \rho_t(\boldsymbol{x})} \langle \boldsymbol{v}_{\boldsymbol{\theta}}(\boldsymbol{x}, t, T), \, \boldsymbol{v}(\boldsymbol{x}, t, T) \rangle$$

$$= \mathbb{E}_{t<T, \, \boldsymbol{x} \sim \rho_t(\boldsymbol{x})} \langle \boldsymbol{v}_{\boldsymbol{\theta}}(\boldsymbol{x}, t, T), \, \frac{1}{T-t} \int_t^T \boldsymbol{u}_\tau(\boldsymbol{x}_\tau) \, d\tau \rangle$$

$$= \mathbb{E}_{t<T} \int_{\mathcal{X}} \Big\langle \boldsymbol{v}_{\boldsymbol{\theta}}(\boldsymbol{x}, t, T), \, \frac{1}{T-t} \int_t^T \boldsymbol{u}_\tau(\boldsymbol{x}_\tau) \, d\tau \rho_t(\boldsymbol{x}) \Big\rangle \, d\boldsymbol{x}$$
(24)

$$\overset{(1)}{=} \mathbb{E}_{t<T} \int_{\mathcal{X}} \Big\langle \boldsymbol{v}_{\boldsymbol{\theta}}(\boldsymbol{x}, t, T), \, \frac{1}{T-t} \int_t^T \frac{\int_{\mathcal{Z}} \boldsymbol{u}_\tau(\boldsymbol{x} \mid \boldsymbol{z}) \, \rho_\tau(\boldsymbol{x} \mid \boldsymbol{z}) \, q(\boldsymbol{z}) \, d\boldsymbol{z}}{\rho_\tau(\boldsymbol{x})} \, d\tau \rho_t(\boldsymbol{x}) \Big\rangle \, d\boldsymbol{x}$$

$$\overset{(2)}{=} \mathbb{E}_{t<T} \int_{\mathcal{Z}} \int_{\mathcal{X}} \Big\langle \boldsymbol{v}_{\boldsymbol{\theta}}(\boldsymbol{x}, t, T), \, \frac{1}{T-t} \int_t^T \frac{\boldsymbol{u}_\tau(\boldsymbol{x} \mid \boldsymbol{z}) \, \rho_\tau(\boldsymbol{x} \mid \boldsymbol{z})}{\rho_\tau(\boldsymbol{x})} \, d\tau \Big\rangle q(\boldsymbol{z}) \rho_t(\boldsymbol{x} \mid \boldsymbol{z}) \, d\boldsymbol{x} \, d\boldsymbol{z},$$

where (1) applies the definition of the marginal flow velocity

$$\boldsymbol{u}_t(\boldsymbol{x}) = \frac{\int \boldsymbol{u}_t(\boldsymbol{x} \mid \boldsymbol{z}) \, \rho_t(\boldsymbol{x} \mid \boldsymbol{z}) \, q(\boldsymbol{z}) \, d\boldsymbol{z}}{\rho_t(\boldsymbol{x})}, \quad \rho_t(\boldsymbol{x}) = \int \rho_t(\boldsymbol{x}|\boldsymbol{z}) q(\boldsymbol{z}) d\boldsymbol{z}.$$

Then we consider the corresponding cross term in the conditional objective:

$$\mathbb{E}_{t<T, \, \boldsymbol{z}, \, \boldsymbol{x} \sim \rho_t(\boldsymbol{x}|\boldsymbol{z})} \Big\langle \boldsymbol{v}_{\boldsymbol{\theta}}(\boldsymbol{x}, t, T), \, \boldsymbol{v}(\boldsymbol{x}, t, T \mid \boldsymbol{z}) \Big\rangle$$

$$= \mathbb{E}_{t<T, \, \boldsymbol{z}, \, \boldsymbol{x} \sim \rho_t(\boldsymbol{x}|\boldsymbol{z})} \Big\langle \boldsymbol{v}_{\boldsymbol{\theta}}(\boldsymbol{x}, t, T), \, \frac{1}{T-t} \int_t^T \boldsymbol{u}_\tau(\boldsymbol{x} \mid \boldsymbol{z}) \, d\tau \Big\rangle$$
(25)

$$= \mathbb{E}_{t<T} \int_{\mathcal{Z}} \int_{\mathcal{X}} \Big\langle \boldsymbol{v}_{\boldsymbol{\theta}}(\boldsymbol{x}, t, T), \, \frac{1}{T-t} \int_t^T \boldsymbol{u}_\tau(\boldsymbol{x} \mid \boldsymbol{z}) \, d\tau \Big\rangle q(\boldsymbol{z}) \, \rho_t(\boldsymbol{x} \mid \boldsymbol{z}) \, d\boldsymbol{x} \, d\boldsymbol{z}.$$

Comparing Eqs. (24) and (25), we observe that, under the time-averaged definition, the two cross terms no longer coincide in general. This establishes the failure of equivalence between $\mathcal{L}$ and $\mathcal{L}_{\mathrm{c}}$. $\square$

### B.3. Recoverability and Uniqueness of the Mean Fields

In this section, we show that the proposed stop-gradient self-referential objective admits a unique fixed point corresponding exactly to the true mean fields $v$ and $h$.

Assume that the stop-gradient objective is perfectly minimized, such that the learned velocity field satisfies the fixed-point relation

$$\boldsymbol{v_\theta}(\boldsymbol{x}, t, T) = \boldsymbol{u}_t(\boldsymbol{x}) + (T - t)\Big[\partial_t \boldsymbol{v_\theta}(\boldsymbol{x}, t, T) + (\nabla_{\boldsymbol{x}} \boldsymbol{v_\theta}(\boldsymbol{x}, t, T))\, \boldsymbol{u}_t(\boldsymbol{x})\Big].$$

Fix $(\boldsymbol{x}, T)$ and define

$$f(t) := \boldsymbol{v_\theta}(\boldsymbol{x}, t, T).$$

Then, for $t < T$, the above equation reduces to

$$f(t) = u_t + (T - t)f'(t).$$

Rearranging terms yields

$$\frac{d}{dt}\big[(T - t)f(t)\big] = -u_t.$$

Integrating both sides from $t$ to $T$, and using the terminal condition

$$(T - T)f(T) = 0,$$

we obtain

$$(T - T)f(T) - (T - t)f(t) = -\int_t^T u_\tau\, d\tau,$$

which implies

$$f(t) = \frac{1}{T - t}\int_t^T u_\tau\, d\tau.$$

Therefore,

$$\boldsymbol{v_\theta}(\boldsymbol{x}, t, T) = \frac{1}{T - t}\int_t^T u_\tau(\boldsymbol{x})\, d\tau,$$

which is exactly the definition of the true mean velocity field.

Since the associated first-order linear ODE admits a unique solution under the terminal condition above, the fixed point is unique. Consequently, perfect minimization of the stop-gradient objective necessarily recovers the true mean field. The same derivation applies analogously to the scalar growth field $h$.

## C. Adaptive Stopping Criterion

We provide an explicit adaptive stopping criterion. Let $p_0$ be the empirical distribution at $t = 0$ with $N_0$ cells and $m_0 = 1/N_0$. Let $s^{(K)} = \{(x_{1,n}^{(K)}, m_{1,n}^{(K)})\}_{n=1}^N$ be the result after $K$ steps. Define the discrepancy between iterates:

$$\Delta_k = d(s^{(k-1)}, s^{(k)}), \quad d = \frac{1}{N}\sum_{n=1}^N \Big(\|\tfrac{x^{(k)} - x^{(k-1)}}{\mathbb{E}_{p_0}[x]}\|_2^2 + |\tfrac{m^{(k)} - m^{(k-1)}}{m_0}|^2\Big).$$

This per-sample $\ell_2$ discrepancy efficiently approximates convergence, avoiding costly metrics like $\mathcal{W}_1$.

We adopt an adaptive stopping rule: iterative refinement stops once improvement falls below a tolerance. When the one-step solution is near a fixed point, $\Delta_k$ is small and further steps are unnecessary; otherwise, iterations continue until convergence. See the algorithm below.

---

**Algorithm 4** Adaptive stopping rule

---

**Input:** Maximum step count $K_{\max}$, tolerance $\tau \in (0, 1)$, patience $P \in \mathbb{N}$, distance function $d(\cdot, \cdot)$
**Output:** Selected iterate $s^{(k_{\text{best}})}$ and best step count $k_{\text{best}}$
$k_{\text{best}} \leftarrow 1, \Delta_{\text{best}} \leftarrow +\infty, c \leftarrow 0$
**for** $k = 1 \rightarrow K_{\max}$ **do**
    Compute $s^{(k)}$
    **if** $k > 1$ **then**
        $\Delta_k \leftarrow d(s^{(k-1)}, s^{(k)})$
        **if** $\Delta_k < (1 - \tau) \Delta_{\text{best}}$ **then**
            $\Delta_{\text{best}} \leftarrow \Delta_k$
            $k_{\text{best}} \leftarrow k$
            $c \leftarrow 0$
        **else**
            $c \leftarrow c + 1$
        **end if**
        **if** $c \geq P$ **then**
            **return** $s^{(k_{\text{best}})}, k_{\text{best}}$
        **end if**
    **end if**
**end for**
**return** $s^{(k_{\text{best}})}, k_{\text{best}}$

---

Here, $\tau$ controls improvement strictness (smaller $\tau \rightarrow$ more steps), and $P$ is the number of tolerated non-improving steps. On the 2D Gene dataset, $P = 2$ and $\tau = 0.15$ stop at $K = 22$, achieving near-optimal $\mathcal{W}_1$ and RME (see Fig. 1 in the main text) using only the local $\ell_2$ discrepancy.

## D. Additional Results

In this section, we present experimental details, training details, evaluation metrics, dataset descriptions, and additional results. Most of the datasets, evaluation metrics, and experimental protocols are shared with WFR-FM (Peng et al., 2026a); for completeness and self-containment, we restate the relevant information here.

### D.1. Experimental Details

All experiments were conducted on a local workstation equipped with an NVIDIA RTX 4070 Ti Super GPU and an Intel i7-12700KF CPU, except for the scalability evaluation on the 100D EB dataset, which was performed on a shared cluster with NVIDIA A100 GPUs and 128 CPU cores. The architecture of the neural networks for $\boldsymbol{v}_\theta(\boldsymbol{x}, t, T)$ and $g_\phi(\boldsymbol{x}, t, T)$ are implemented using 5-layer Multilayer Perceptrons with 256 hidden units per layer and LeakyReLU activations. These networks were optimized using Pytorch (Paszke et al., 2017). The OET problem is solved using the Python Optimal Transport (POT) package (Flamary et al., 2021).

### D.2. Training Details

**Sampling Time.** We describe the procedure used to sample time pairs $(t, T)$ that define the temporal span for evaluating the mean fields. In this work, time pairs are sampled from a uniform distribution. To ensure consistency with FM and proper boundary behavior, a fraction of samples are enforced to satisfy $t = T$, corresponding to the instantaneous-velocity case, as commonly adopted in prior work (Geng et al., 2025a). This mixture of instantaneous and interval-based samples improves training stability and generation quality. The detailed sampling procedure is summarized in Algorithm 5.

### D.3. Evaluation Metrics

We evaluate model performance using two metrics: the 1-Wasserstein distance ($\mathcal{W}_1$), which measures the similarity between predicted and true distributions, and the Relative Mass Error (RME), which assesses how well the model captures cell

---

**Algorithm 5** Sampling time pairs $(t, T)$

---

1: **Input:** batch size $B$, proportion $p_{\text{diff}}$ of samples with $t \neq T$
2: **for each sample** do
3:     Draw $z \sim \text{Bernoulli}(1 - p_{\text{diff}})$
4: **if** $z = 1$ **then**
5:     Sample $t \sim \mathcal{U}(0, 1)$
6:     $T \leftarrow t$
7: **else**
8:     Sample $t_1, t_2 \sim \mathcal{U}(0, 1)$
9:     $t \leftarrow \min(t_1, t_2), \quad T \leftarrow \max(t_1, t_2)$
10: **end if**
11: **return** $\{(t, T)\}_{i=1}^B$

---

population growth. The metrics are defined as:

$$\mathcal{W}_1(p, q) = \min_{\pi \in \Pi(p,q)} \int \|\boldsymbol{x} - \boldsymbol{y}\|_2 d\pi(\boldsymbol{x}, \boldsymbol{y}),$$

$$\text{RME}(t_k) = \frac{|\sum_i w_i(t_k) - n_k/n_0|}{n_k/n_0}.$$

Here, $p$ and $q$ denote the empirical distributions of predicted and observed cells, respectively, $w_i(t_k)$ represents the inferred mass associated with cell $i$ at time $t_k$, and $n_k$ denotes the number of observed cells at time point $k$.

For evaluation, we propagate the learned dynamics from the initial cell population, where all cells are initialized with equal weights $w_i(0) = 1/n_0$, to generate predicted cell states at later time points. When an unbalanced formulation is employed, cell weights are evolved jointly with the state dynamics; otherwise, weights remain uniform throughout. We then compute the weighted $\mathcal{W}_1$ distance and the RME by comparing the predicted distributions against the observed data at each time point. For selected datasets, we further conduct a hold-out evaluation by excluding one time point during training and reporting the $\mathcal{W}_1$ distance on the unseen snapshot. To ensure a fair comparison across methods, we reimplemented TIGON (Sha et al., 2024) to mitigate numerical instabilities observed in the original implementation. For the remaining baselines, we largely follow the default configurations reported in their respective papers when applicable; otherwise, we adjust network widths to match parameter scales and tune training epochs and learning rates for each dataset to ensure balanced comparisons.

### D.4. Results on held-out time points

We report quantitative results on held-out time points for real biological datasets, including EMT, EB, CITE, and Mouse. For each dataset, we perform an evaluation over intermediate time points by holding out one time point at a time for testing, while using all remaining time points for training. Performance is evaluated on the held-out time point using the $\mathcal{W}_1$ metric, and the final results are obtained by averaging over all intermediate time points. The mean $\mathcal{W}_1$ scores are summarized in Table 3.

*Table 3.* Mean $\mathcal{W}_1$ over held-out time points on EMT, EB, CITE, and Mouse datasets.

| METHOD | EMT (10D) | EB (50D) | CITE (50D) | MOUSE (50D) |
|---|---|---|---|---|
| MMFM | 0.323 | 11.213 | 38.521 | 8.263 |
| METRIC FM | 0.314 | 10.726 | 37.342 | 7.753 |
| SF2M | $0.308_{\pm 0.001}$ | $10.986_{\pm 0.006}$ | $38.333_{\pm 0.002}$ | $8.646_{\pm 0.004}$ |
| MIOFLOW | 0.325 | 10.960 | 39.574 | 7.779 |
| TIGON | 0.360 | 11.080 | 38.159 | 6.868 |
| DEEPRUOT | $0.323_{\pm 0.002}$ | $\mathbf{10.075}_{\pm 0.004}$ | $37.892_{\pm 0.002}$ | $6.847_{\pm 0.003}$ |
| VAR-RUOT | $0.320_{\pm 0.003}$ | $11.035_{\pm 0.017}$ | $38.393_{\pm 0.029}$ | $8.672_{\pm 0.040}$ |
| UOT-FM | 0.322 | 11.344 | 38.649 | 9.332 |
| VGFM | 0.301 | 10.370 | 37.386 | 8.496 |
| WFR-FM | **0.298** | 10.157 | 37.221 | **6.586** |
| **WFR-MFM** | 0.299 | 10.135 | **35.736** | 6.714 |

## D.5. Performance on Simulation Gene Dataset

Following the experimental setup in (Zhang et al., 2025b), we utilize a synthetic dataset representing a gene regulatory network. The temporal evolution of gene concentrations is modeled by a system of stochastic differential equations (SDEs) as follows:

$$\frac{dX_1}{dt} = \frac{\alpha_1 X_1^2 + \beta}{1 + \alpha_1 X_1^2 + \gamma_2 X_2^2 + \gamma_3 X_3^2 + \beta} - \delta_1 X_1 + \eta_1 \xi_t$$

$$\frac{dX_2}{dt} = \frac{\alpha_2 X_2^2 + \beta}{1 + \gamma_1 X_1^2 + \alpha_2 X_2^2 + \gamma_3 X_3^2 + \beta} - \delta_2 X_2 + \eta_2 \xi_t$$

$$\frac{dX_3}{dt} = \frac{\alpha_3 X_3^2}{1 + \alpha_3 X_3^2} - \delta_3 X_3 + \eta_3 \xi_t$$

In this system, $X_i(t)$ denotes the concentration level of the $i$-th gene. The network topology incorporates a mutual inhibition mechanism between genes $X_1$ and $X_2$, both of which exhibit self-activation capabilities. Furthermore, an external signal $\beta$ promotes the activation of $X_1$ and $X_2$, while gene $X_3$ acts as a repressor for both. The parameters $\alpha_i$, $\gamma_i$, and $\delta_i$ correspond to the rates of self-activation, cross-inhibition, and degradation, respectively, with $\eta_i \xi_t$ representing the stochastic noise component.

To simulate population dynamics, we incorporate a probabilistic cell division process. The instantaneous growth rate $g$ is modulated by the expression level of $X_2$, defined as $g = \alpha_g \frac{X_2^2}{1+X_2^2}$. When division occurs, the parent cell's state is passed to the daughter cells with minor random perturbations. The dataset consists of snapshots taken at discrete time intervals $t \in \{0, 8, 16, 24, 32\}$, originating from two separate initial populations: one undergoing dynamic transition and growth, and the other maintaining a steady-state equilibrium.

**Choice of WFR-penalty** $\delta$. The hyperparameter $\delta$ serves as a weighting factor in the WFR metric, balancing the trade-off between the transport cost and the unbalanced mass variation cost (birth-death). As defined in equation 1, a higher $\delta$ imposes a stronger penalty on the growth term $g$, thereby forcing the model to prioritize spatial transport over mass creation or annihilation. Table 4 (evaluated on the Simulation Gene dataset) shows that performance is highly sensitive to $\delta$, with the optimum found at $\delta = 1.5$. Increasing $\delta$ degrades results significantly; extreme values (e.g., $\delta = 20$) cause drastic divergence ($\mathcal{W}_1 > 110$). This suggests the data involves significant mass variation, and excessively penalizing the birth-death term forces the model into erroneous pure-transport solutions. We set $\delta = 1.5$ for the main experiments on this dataset.

*Table 4.* Sensitivity analysis for parameter $\delta$ on Simulation Gene Dataset ($p_{\text{diff}} = 0.6, \lambda = 0.05$).

| Parameter | t=1 | | t=2 | | t=3 | | t=4 | |
|---|---|---|---|---|---|---|---|---|
| | $\mathcal{W}_1$ | TMV | $\mathcal{W}_1$ | TMV | $\mathcal{W}_1$ | TMV | $\mathcal{W}_1$ | TMV |
| $\delta = 1$ | 0.0234 | 0.0061 | 0.0250 | 0.0086 | 0.0200 | 0.0060 | 0.0194 | 0.0057 |
| $\delta = 1.5$ | 0.0224 | 0.0006 | 0.0219 | 0.0006 | 0.0212 | 0.0054 | 0.0208 | 0.0062 |
| $\delta = 2$ | 0.0400 | 0.0005 | 0.0514 | 0.0067 | 0.0452 | 0.0133 | 0.0566 | 0.0133 |
| $\delta = 5$ | 0.0469 | 0.0002 | 0.1579 | 0.0053 | 0.1251 | 0.0283 | 0.1576 | 0.0288 |
| $\delta = 10$ | 0.0607 | 0.0001 | 0.1065 | 0.0114 | 0.1058 | 0.0218 | 0.2655 | 0.0164 |
| $\delta = 20$ | 0.1099 | 0.0001 | 0.2002 | 0.0081 | 0.7320 | 0.0289 | 110.88 | 0.1213 |

**Choice of Cross-time Sampling Proportion** $p_{\text{diff}}$. The parameter $p_{\text{diff}}$ controls the probability of sampling different time pairs ($t \neq T$). A balanced $p_{\text{diff}}$ is essential: excessively low values restrict learning to instantaneous fields, failing to capture time-averaged dynamics, while overly high values may neglect immediate time-step reconstruction. Table 5 presents the sensitivity analysis. We observe that a low $p_{\text{diff}} = 0.1$ results in suboptimal performance with higher transport errors. However, performance improves significantly as $p_{\text{diff}}$ increases, reaching an optimal range between 0.4 and 0.6 where both Wasserstein distances and TMV are minimized. Notably, setting $p_{\text{diff}}$ too high (e.g., 0.8) leads to slight degradation in long-term accuracy ($t = 3, 4$). Based on these results, we set $p_{\text{diff}} = 0.6$ for the main experiments on this dataset.

**Choice of Loss Weight** $\lambda$. The parameter $\lambda$ balances the supervision between the average velocity field and the mass-growth field. An appropriate $\lambda$ is critical: a value too low weakens growth supervision, risking incorrect population estimation, while a value too high dominates the loss, sacrificing spatial transport accuracy. As shown in Table 6, the model achieves

*Table 5.* Sensitivity analysis for parameter $p_{\text{diff}}$ on Simulation Gene Dataset ($\delta = 1.5$, $\lambda = 0.05$).

| Parameter | t=1 | | t=2 | | t=3 | | t=4 | |
|---|---|---|---|---|---|---|---|---|
| | $\mathcal{W}_1$ | TMV | $\mathcal{W}_1$ | TMV | $\mathcal{W}_1$ | TMV | $\mathcal{W}_1$ | TMV |
| $p_{\text{diff}} = 0.1$ | 0.024 | 0.0034 | 0.027 | 0.0038 | 0.029 | 0.0069 | 0.030 | 0.0135 |
| $p_{\text{diff}} = 0.2$ | 0.023 | 0.0004 | 0.027 | 0.0038 | 0.026 | 0.0086 | 0.026 | 0.0121 |
| $p_{\text{diff}} = 0.3$ | 0.023 | 0.0019 | 0.026 | 0.0051 | 0.024 | 0.0081 | 0.024 | 0.0096 |
| $p_{\text{diff}} = 0.4$ | 0.022 | 0.0007 | 0.024 | 0.0028 | 0.024 | 0.0095 | 0.023 | 0.0125 |
| $p_{\text{diff}} = 0.5$ | 0.023 | 0.0008 | 0.023 | 0.0030 | 0.022 | 0.0083 | 0.022 | 0.0092 |
| $p_{\text{diff}} = 0.6$ | 0.021 | 0.0003 | 0.022 | 0.0010 | 0.020 | 0.0052 | 0.019 | 0.0061 |
| $p_{\text{diff}} = 0.7$ | 0.023 | 0.0012 | 0.022 | 0.0011 | 0.023 | 0.0046 | 0.022 | 0.0075 |
| $p_{\text{diff}} = 0.8$ | 0.022 | 0.0015 | 0.023 | 0.0022 | 0.025 | 0.0100 | 0.024 | 0.0118 |

optimal performance at $\lambda = 0.05$, exhibiting the lowest transport errors ($\mathcal{W}_1$) and trajectory variance (TMV), particularly at later time steps ($t = 3, 4$). Notably, the algorithm demonstrates significant robustness; performance metrics remain highly stable across a wide range of magnitudes ($\lambda \in [1, 50]$), indicating that the method is insensitive to hyperparameter tuning provided $\lambda$ is not negligible. Consequently, we fix $\lambda = 0.05$ for the reported experiments.

*Table 6.* Sensitivity analysis for parameter $\lambda$ on simulation gene dataset ($\delta = 1.5$, $p_{\text{diff}} = 0.6$).

| Parameter | t=1 | | t=2 | | t=3 | | t=4 | |
|---|---|---|---|---|---|---|---|---|
| | $\mathcal{W}_1$ | TMV | $\mathcal{W}_1$ | TMV | $\mathcal{W}_1$ | TMV | $\mathcal{W}_1$ | TMV |
| $\lambda = 0.01$ | 0.023 | 0.0005 | 0.022 | 0.0001 | 0.024 | 0.0081 | 0.023 | 0.0100 |
| $\lambda = 0.05$ | 0.021 | 0.0003 | 0.022 | 0.0010 | 0.020 | 0.0052 | 0.019 | 0.0061 |
| $\lambda = 0.1$ | 0.022 | 0.0010 | 0.022 | 0.0013 | 0.023 | 0.0085 | 0.022 | 0.0109 |
| $\lambda = 1$ | 0.022 | 0.0004 | 0.023 | 0.0015 | 0.023 | 0.0076 | 0.022 | 0.0087 |
| $\lambda = 10$ | 0.022 | 0.0006 | 0.022 | 0.0003 | 0.023 | 0.0076 | 0.022 | 0.0103 |
| $\lambda = 50$ | 0.023 | 0.0012 | 0.022 | 0.0007 | 0.023 | 0.0073 | 0.022 | 0.0091 |

### D.6. Performance on Dyngen Dataset

We adopt the unbalanced bifurcation simulation previously analyzed in (Huguet et al., 2022) and (Wang et al., 2025). Produced by Dyngen (Cannoodt et al., 2021), this dataset comprises 728 cells with dimensionality reduced to 5 via PHATE (Moon et al., 2019). The complexity of this data arises from two main factors: significant fluctuations in total mass over time, and a pronounced structural asymmetry, characterized by a much larger number of cells populating the lower branch compared to the upper one. We fix WFR penalty to $\delta = 1.5$, cross-time sampling proportion to $p_{\text{diff}} = 0.5$ and growth penalty to $\lambda = 5$. As shown in Figure 5, WFR-FM accurately captures the underlying dynamics.

Detailed quantitative comparisons are provided in Table 7. Our method yields highly competitive results compared to WFR-FM. Notably, we achieve the lowest $\mathcal{W}_1$ error at $t = 3$ and consistently maintain the second-lowest errors across other time points for both $\mathcal{W}_1$ and RME.

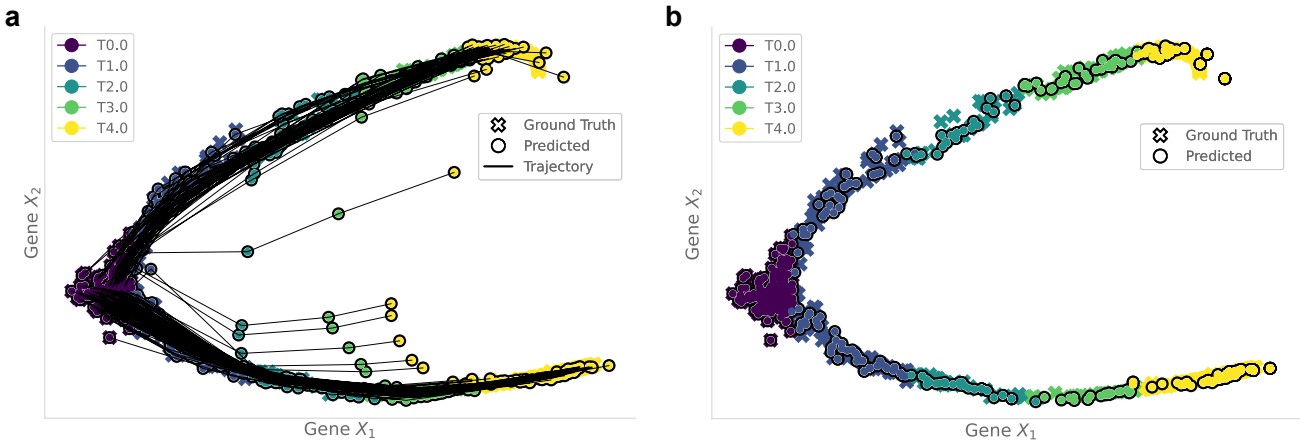

*Figure 5.* **Learned dynamics on the Dyngen dataset.** (a) Trajectories of individual cells with a fixed cell count, where only cell weights evolve over time. (b) Population-level dynamics obtained by normalizing weights into probabilities and resampling cells, revealing changes in cell abundance.

*Table 7.* Comparison of method performance over time on the Dyngen dataset.

| Method | t=1 | | t=2 | | t=3 | | t=4 | |
|---|---|---|---|---|---|---|---|---|
| | $\mathcal{W}_1$ | RME | $\mathcal{W}_1$ | RME | $\mathcal{W}_1$ | RME | $\mathcal{W}_1$ | RME |
| MMFM | 0.574 | — | 1.704 | — | 1.499 | — | 1.706 | — |
| Metric FM | 0.892 | — | 2.347 | — | 2.030 | — | 1.799 | — |
| SF2M | 0.637 | — | 1.266 | — | 1.415 | — | 1.790 | — |
| MIOFlow | 0.420 | — | 0.640 | — | 1.537 | — | 1.263 | — |
| TIGON | 0.446 | 0.033 | 0.584 | 0.060 | 0.415 | 0.023 | 0.603 | 0.071 |
| DeepRUOT | 0.454 | 0.011 | 0.481 | 0.070 | 0.870 | 0.104 | 0.688 | 0.074 |
| Var-RUOT | 0.315 | 0.128 | 0.548 | 0.336 | 0.630 | 0.222 | 0.593 | 0.023 |
| UOT-FM | 0.652 | 0.008 | 0.780 | 0.077 | 1.252 | 0.090 | 2.130 | 0.213 |
| VGFM | 0.335 | **0.001** | 0.312 | 0.073 | 1.109 | 0.041 | 0.634 | 0.033 |
| WFR-FM | **0.110** | 0.003 | **0.098** | **0.007** | 0.211 | **0.008** | **0.121** | **0.002** |
| WFR-MFM | 0.130 | 0.007 | 0.163 | 0.009 | **0.202** | 0.017 | 0.197 | 0.111 |

## D.7. Performance on Gaussian Mixture Dataset

We evaluate our method on the 1000-dimensional Gaussian Mixture dataset adopted from (Wang et al., 2025). Following their experimental setup, we generate an initial distribution of 500 samples (100 from an upper Gaussian component and 400 from a lower Gaussian component) and a target distribution of 1,400 samples (1,000 from the upper component and 200 from each of the two lower Gaussians). This configuration is designed to simulate unbalanced population dynamics, specifically modeling cell proliferation in the upper region, which serves as a robust benchmark for evaluating transport algorithms in high-dimensional settings.

We apply WFR-MFM to this task with hyperparameters set to $\delta = 1.4$, $p_{\text{diff}} = 0.05$ and $\lambda = 1$. As illustrated in Figure 6, our method successfully captures the underlying dynamics.

## D.8. Performance on Epithelial–Mesenchymal Transition Dataset

We utilize the single-cell dataset capturing the epithelial-mesenchymal transition (EMT) in A549 lung cancer cells, originally collected by (Cook & Vanderhyden, 2020). This dataset comprises samples taken at four distinct time points during the transition process. Consistent with the preprocessing steps outlined in (Sha et al., 2024), the data dimensionality was reduced to 10 using an AutoEncoder.

We evaluate WFR-MFM on this dataset with hyperparameters set to $\delta = 2$, $p_{\text{diff}} = 0.05$ and $\lambda = 20$. The quantitative results

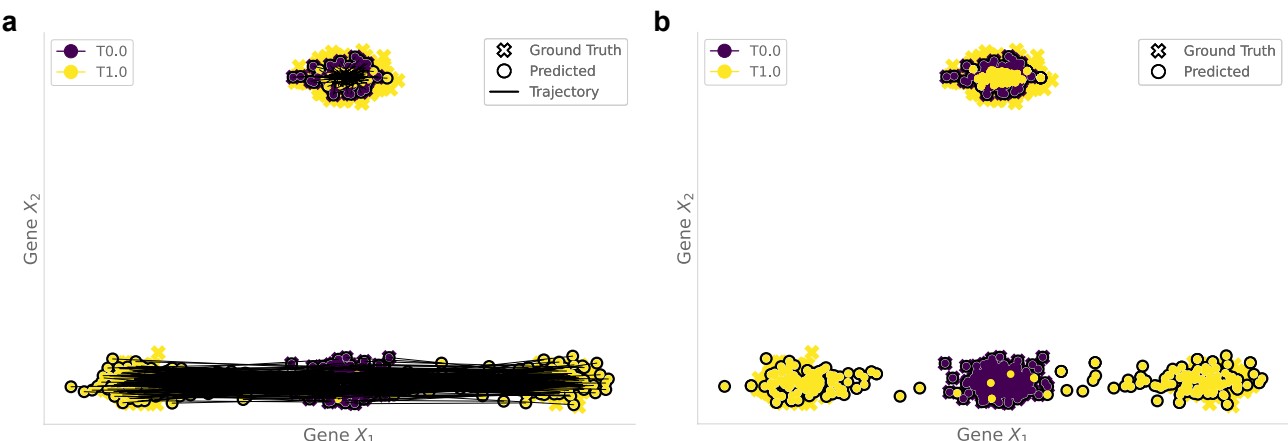

*Figure 6.* **Learned dynamics on the Gaussian 1000D dataset.** (a) Trajectories of individual cells with a fixed cell count, where only cell weights evolve over time. (b) Population-level dynamics obtained by normalizing weights into probabilities and resampling cells, revealing changes in cell abundance.

*Table 8.* Comparison of method performance over time on the 10D EMT dataset.

| Method | t=1 | | t=2 | | t=3 | |
|---|---|---|---|---|---|---|
| | $\mathcal{W}_1$ | RME | $\mathcal{W}_1$ | RME | $\mathcal{W}_1$ | RME |
| MMFM | 0.2576 | — | 0.2874 | — | 0.3102 | — |
| Metric FM | 0.2605 | — | 0.2971 | — | 0.3050 | — |
| SF2M | 0.2566 | — | 0.2811 | — | 0.2900 | — |
| MIOFlow | 0.2439 | — | 0.2665 | — | 0.2841 | — |
| TIGON | 0.2433 | 0.002 | 0.2661 | 0.003 | 0.2847 | **0.001** |
| DeepRUOT | 0.2902 | **0.001** | 0.3193 | 0.011 | 0.3291 | 0.002 |
| Var-RUOT | 0.2540 | 0.075 | 0.2670 | 0.014 | 0.2683 | 0.041 |
| UOT-FM | 0.2538 | 0.002 | 0.2696 | 0.013 | 0.2771 | 0.010 |
| VGFM | 0.2350 | 0.016 | 0.2420 | 0.011 | 0.2450 | 0.018 |
| WFR-FM | **0.2099** | **0.001** | **0.2272** | **0.002** | **0.2346** | **0.001** |
| WFR-MFM | 0.2250 | 0.005 | 0.2376 | 0.003 | 0.2440 | 0.003 |

are summarized in Table 8. WFR-MFM demonstrates highly competitive performance across all time intervals. In the distribution matching task (measured by $\mathcal{W}_1$), our method consistently ranks as the second-best approach, outperforming other unbalanced transport baselines such as VGFM and UOT-FM, and trailing only the WFR-FM benchmark. Furthermore, WFR-MFM maintains a low RME, indicating its robustness in modeling the mass variation inherent in the EMT process where cells exhibit enhanced stemness and proliferation.

### D.9. Performance on Embryoid Bodies Dataset

Our study employs the human embryoid body (EB) differentiation dataset from (Moon et al., 2019), which captures 16,819 cells sampled at five intervals over a 27-day period to model early development. To prepare the data for trajectory inference, we adopt the dimensionality reduction strategy used in (Wang et al., 2025), compressing the original gene expression space using Principal Component Analysis (PCA). This preprocessed, lower-dimensional representation serves as the direct input for our WFR-MFM algorithm.

**Large-scale Experiment.** Figure 7 provides complementary views of the efficiency–accuracy trade-offs on the 100D EB dataset, illustrating the relationships between predictive accuracy ($\mathcal{W}_1$ distance) and training time, as well as inference time, with color indicating GPU memory usage. Tmehe corresponding quantitative results are summarized in Table 9.

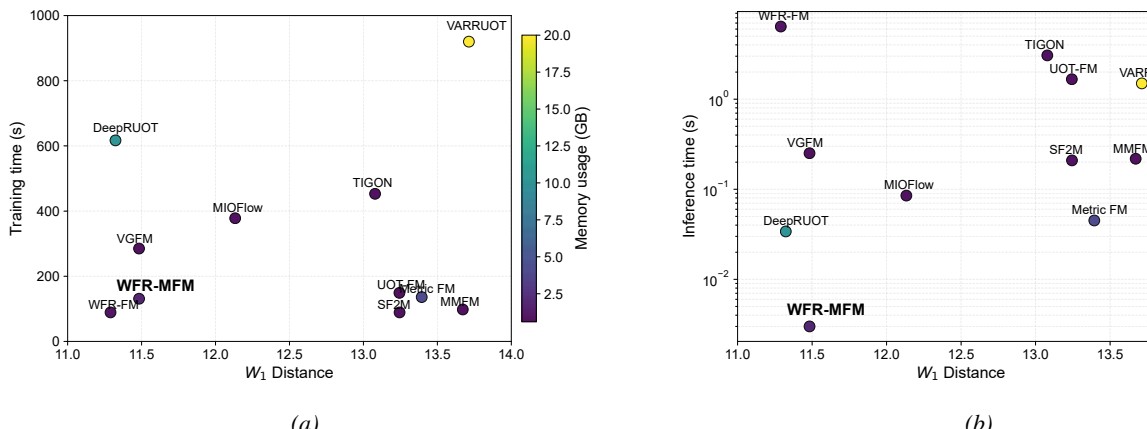

*Figure 7.* **Efficiency on the 100D EB dataset.** Methods are compared in terms of predictive accuracy ($\mathcal{W}_1$ distance) and computational efficiency, including training cost and inference latency, with color indicating GPU memory usage.

*Table 10.* Sensitivity analysis for batch size of mini-batch WFR-MFM on the 100D EB dataset. The time reported represents the total training time.

| Batch Size | t=1 | | t=2 | | t=3 | | t=4 | | Average | | Train Time (s) |
|---|---|---|---|---|---|---|---|---|---|---|---|
| | $\mathcal{W}_1$ | RME | $\mathcal{W}_1$ | RME | $\mathcal{W}_1$ | RME | $\mathcal{W}_1$ | RME | $\mathcal{W}_1$ | RME | |
| 500 | 10.072 | 0.007 | 11.248 | 0.005 | 11.748 | 0.004 | 12.815 | 0.009 | 11.471 | 0.006 | 136 |
| 1000 | 10.025 | 0.008 | 11.294 | 0.011 | 11.803 | 0.007 | 12.831 | 0.018 | 11.488 | 0.011 | 131 |
| 1500 | 10.020 | 0.007 | 11.274 | 0.007 | 11.779 | 0.006 | 12.859 | 0.015 | 11.483 | 0.009 | 131 |
| 2000 | 10.050 | 0.008 | 11.289 | 0.012 | 11.858 | 0.011 | 12.926 | 0.018 | 11.531 | 0.012 | 135 |
| 3000 | 9.999 | 0.008 | 11.283 | 0.009 | 11.792 | 0.012 | 12.892 | 0.022 | 11.492 | 0.013 | 151 |
| w/o mini-batch | 9.995 | 0.009 | 11.222 | 0.012 | 11.720 | 0.010 | 12.782 | 0.017 | 11.430 | 0.012 | 295 |

*Table 9.* Efficiency and overall performance comparison.

| Method | $W_1$ | Training Time | Inference Time |
|---|---|---|---|
| MMFM | 13.672 | 1m38s | 0.218s |
| Metric FM | 13.394 | 2m16s | 0.045s |
| SF2M | 13.245 | 1m29s | 0.209s |
| MIOFlow | 12.133 | 6m18s | 0.085s |
| TIGON | 13.079 | 7m33s | 3.06s |
| DeepRUOT | 11.324 | 10m17s | 0.034s |
| VGFM | 11.483 | 4m45s | 0.251s |
| WFR-FM | **11.290** | 2m08s | 6.40s |
| VARRUOT | 13.713 | 15m20s | 1.496s |
| **WFR-MFM** | 11.483 | **2m11s** | **0.003s** |

**Efficiency of Mini-batch WFR-OET.** We utilized a mini-batch strategy to improve training efficiency. Table 10 shows that this method significantly cuts down training time with negligible impact on accuracy. The model demonstrates robustness across batch sizes, with the 1500-batch setting achieving an even better RME (0.009) than the full-batch baseline (0.012). Therefore, we selected 1500 as the optimal batch size, providing the best combination of low training cost, high mass conservation, and reliable $\mathcal{W}_1$ performance.

**Scalability of WFR-MFM across Dimensions.** Utilizing the optimal mini-batch size of 1500 established previously, we evaluated the scalability of WFR-MFM by comparing it against baseline methods across varying data dimensions: 5 (Table 11), 50 (Table 12), and 100 (Table 13). In these experiments, we fixed $\delta = 25$ and $p_{\text{diff}} = 0.6$, with $\lambda$ adjusted to

50, 100, and 1, respectively. The results indicate that WFR-MFM consistently outperforms competing approaches at the majority of time points across all datasets.

*Table 11.* Comparison of method performance over time on the 5D EB dataset.

| Method | t=1 | | t=2 | | t=3 | | t=4 | |
|---|---|---|---|---|---|---|---|---|
| | $\mathcal{W}_1$ | RME | $\mathcal{W}_1$ | RME | $\mathcal{W}_1$ | RME | $\mathcal{W}_1$ | RME |
| MMFM | 0.477 | — | 0.554 | — | 0.781 | — | 0.872 | — |
| Metric FM | 0.449 | — | 0.552 | — | 0.583 | — | 0.597 | — |
| SF2M | 0.556 | — | 0.715 | — | 0.750 | — | 0.650 | — |
| MIOFlow | 0.442 | — | 0.585 | — | 0.651 | — | 0.670 | — |
| TIGON | 0.386 | 0.002 | 0.502 | 0.015 | 0.602 | 0.021 | 0.600 | 0.027 |
| DeepRUOT | 0.386 | 0.005 | 0.497 | 0.017 | 0.591 | 0.021 | 0.585 | 0.030 |
| Var-RUOT | 0.416 | 0.111 | 0.486 | 0.144 | 0.509 | 0.054 | 0.511 | 0.022 |
| UOT-FM | 0.544 | 0.032 | 0.670 | 0.029 | 0.729 | 0.016 | 0.852 | 0.041 |
| VGFM | 0.402 | 0.046 | 0.494 | 0.018 | 0.525 | 0.035 | 0.573 | 0.021 |
| WFR-FM | **0.324** | 0.003 | **0.401** | **0.001** | **0.431** | 0.005 | 0.510 | 0.005 |
| WFR-MFM | 0.356 | **0.0001** | 0.438 | 0.010 | 0.477 | **0.0002** | 0.502 | **0.001** |

*Table 12.* Comparison of method performance over time on the 50D EB dataset.

| Method | t=1 | | t=2 | | t=3 | | t=4 | |
|---|---|---|---|---|---|---|---|---|
| | $\mathcal{W}_1$ | RME | $\mathcal{W}_1$ | RME | $\mathcal{W}_1$ | RME | $\mathcal{W}_1$ | RME |
| MMFM | 9.124 | — | 10.474 | — | 11.022 | — | 11.480 | — |
| Metric FM | 8.506 | — | 9.795 | — | 10.621 | — | 12.042 | — |
| SF2M | 9.247 | — | 10.882 | — | 11.650 | — | 12.154 | — |
| MIOFlow | 8.447 | — | 9.229 | — | 9.436 | — | 10.123 | — |
| TIGON | 8.433 | 0.067 | 9.275 | 0.022 | 9.802 | 0.179 | 10.148 | 0.101 |
| DeepRUOT | 8.169 | 0.003 | 9.049 | 0.038 | 9.378 | 0.088 | 9.733 | 0.004 |
| Var-RUOT | 9.442 | 0.128 | 9.709 | 0.081 | 10.482 | 0.031 | 10.735 | 0.030 |
| UOT-FM | 8.717 | 0.063 | 10.858 | 0.009 | 11.813 | 0.022 | 12.733 | 0.018 |
| VGFM | 7.951 | 0.089 | 8.747 | 0.042 | 9.244 | 0.019 | 9.620 | 0.044 |
| WFR-FM | 7.664 | 0.008 | 8.659 | 0.006 | 9.182 | **0.004** | 9.914 | 0.004 |
| WFR-MFM | **5.236** | 0.002 | **5.904** | 0.005 | **6.190** | 0.006 | **6.647** | **0.001** |

### D.10. Performance on CITE-seq Dataset

We further evaluated our method on the CITE-seq dataset (Lance et al., 2022), comprising 31,240 cells collected over four time points. Following the preprocessing steps in (Wang et al., 2025), we utilized the gene expression matrix reduced to 50 dimensions via PCA. The experiments were conducted with a batch size of 1,500, setting the hyperparameters to $\delta = 30$, $p_{\text{diff}} = 0.3$, and $\lambda = 1$. As presented in Table 14, although the one-step implementation (WFR-MFM) yields suboptimal performance, increasing the number of inference steps leads to significant improvements. By adjusting the inference to 10 steps, our method achieves state-of-the-art results at $t = 1$ for both $\mathcal{W}_1$ and RME, and secures the best mass estimation accuracy at $t = 2$. This demonstrates that while a coarse integration may be insufficient for complex dynamics, a multi-step scheme effectively unlocks the method's potential, yielding satisfactory distribution matching and population growth modeling.

*Table 13.* Comparison of method performance over time on the 100D EB dataset.

| Method | t=1 | | t=2 | | t=3 | | t=4 | |
|--------|-----------|-------|-----------|-------|-----------|-------|-----------|-------|
| | $\mathcal{W}_1$ | RME | $\mathcal{W}_1$ | RME | $\mathcal{W}_1$ | RME | $\mathcal{W}_1$ | RME |
| MMFM | 11.460 | — | 13.879 | — | 14.441 | — | 14.907 | — |
| Metric FM | 10.806 | — | 12.348 | — | 13.622 | — | 16.801 | — |
| SF2M | 11.333 | — | 12.982 | — | 13.718 | — | 14.945 | — |
| MIOFlow | 11.387 | — | 12.331 | — | 11.905 | — | 12.908 | — |
| TIGON | 10.547 | 0.014 | 12.926 | 0.052 | 13.897 | 0.107 | 14.945 | 0.096 |
| DeepRUOT | 10.256 | **0.002** | 11.103 | 0.074 | 11.529 | 0.136 | **12.406** | 0.047 |
| Var-RUOT | 11.746 | 0.091 | 12.237 | 0.024 | 12.957 | 0.150 | 13.335 | 0.074 |
| UOT-FM | 10.757 | 0.056 | 12.799 | 0.037 | 13.761 | 0.044 | 15.657 | 0.022 |
| VGFM | 10.313 | 0.048 | 11.278 | 0.035 | 11.703 | 0.028 | 12.637 | 0.066 |
| WFR-FM | **9.941** | 0.009 | **11.040** | **0.006** | **11.516** | 0.008 | 12.664 | **0.005** |
| WFR-MFM | 10.020 | 0.007 | 11.274 | 0.007 | 11.779 | **0.006** | 12.859 | 0.015 |

*Table 14.* Comparison of method performance over time on the 50D CITE dataset.

| Method | t=1 | | t=2 | | t=3 | |
|--------|-----------|-------|-----------|-------|-----------|-------|
| | $\mathcal{W}_1$ | RME | $\mathcal{W}_1$ | RME | $\mathcal{W}_1$ | RME |
| MMFM | 33.971 | — | 36.854 | — | 43.721 | — |
| Metric FM | 28.314 | — | 28.617 | — | 33.212 | — |
| SF2M | 29.543 | — | 32.655 | — | 36.265 | — |
| MIOFlow | 28.290 | — | 28.524 | — | **32.230** | — |
| TIGON | 28.196 | 0.186 | 27.921 | 0.545 | 32.846 | 0.653 |
| DeepRUOT | 28.245 | 0.168 | 27.908 | 0.525 | 32.950 | 0.634 |
| Var-RUOT | 30.219 | 0.331 | 32.702 | 0.325 | 40.613 | 0.486 |
| UOT-FM | 33.531 | 0.009 | 32.795 | 0.046 | 49.751 | 0.097 |
| VGFM | 29.449 | 0.020 | 29.722 | 0.057 | 33.752 | **0.001** |
| WFR-FM | 27.831 | 0.043 | **27.478** | 0.045 | 34.784 | 0.022 |
| WFR-MFM | 30.841 | 0.150 | 29.390 | 0.187 | 40.471 | 0.427 |
| WFR-MFM(10 steps) | **27.509** | **0.006** | 28.255 | **0.033** | 34.055 | 0.062 |

In addition to latent-space evaluation, we further report several representative experiments directly in the original 2000-dimensional gene expression space to address the reviewer's concern regarding high-dimensional biological datasets. In single-cell analysis, modeling dynamics in a low-dimensional latent space (e.g., PCA, autoencoders, or VAEs) is standard practice due to the substantial sparsity, redundancy, and technical noise in raw gene expression measurements (Lopez et al., 2018; Lotfollahi et al., 2019; Klein et al., 2025). Accordingly, our primary experiments are conducted in latent space for more stable and computationally tractable comparisons. Nevertheless, we additionally decoded representative predictions back to gene expression space and evaluated $\mathcal{W}_1$. As shown in Table 15, WFR-MFM consistently outperforms competing baselines, while increasing the number of inference steps further improves performance, demonstrating the effectiveness of the proposed iterative refinement strategy even in high-dimensional settings.

*Table 15.* Representative results on the 2000D CITE dataset in gene expression space.

| Method | $\mathcal{W}_1$ (t=1) | $\mathcal{W}_1$ (t=2) | $\mathcal{W}_1$ (t=3) |
|---|---|---|---|
| MMFM | 42.314 | 46.996 | 72.802 |
| SF2M | 42.568 | 50.009 | 78.588 |
| Var-RUOT | 41.230 | 37.645 | 54.266 |
| VGFM | 41.058 | 37.591 | 53.354 |
| WFR-FM | 41.084 | 39.342 | 58.024 |
| WFR-MFM | 40.513 | 37.319 | 51.587 |
| WFR-MFM(10 steps) | **39.837** | **36.327** | **49.357** |

### D.11. Performance on Mouse Hematopoiesis Dataset

We further validated our method on the mouse blood hematopoiesis dataset (Weinreb et al., 2020), comprising 49,302 lineage-traced cells at three time points. Following PCA reduction to 50 dimensions, we applied our method with a batch size of 1,500 and fixed hyperparameters ($\delta = 15, p_{\text{diff}} = 0.6, \lambda = 50$). As detailed in Table 16, WFR-MFM yields highly competitive results: it secures the top performance for RME at $t = 1$ and $\mathcal{W}_1$ at $t = 2$, and remains the runner-up in other scenarios. These results highlight the method's reliability in capturing the dynamics of hematopoiesis.

*Table 16.* Comparison of method performance over time on the 50D Mouse dataset.

| Method | t=1 | | t=2 | |
|---|---|---|---|---|
| | $\mathcal{W}_1$ | RME | $\mathcal{W}_1$ | RME |
| MMFM | 7.647 | — | 10.156 | — |
| Metric FM | 7.788 | — | 11.449 | — |
| SF2M | 8.217 | — | 11.086 | — |
| MIOFlow | 6.313 | — | 6.746 | — |
| TIGON | 6.140 | 0.382 | 6.973 | 0.326 |
| DeepRUOT | 6.052 | 0.062 | 6.757 | 0.041 |
| Var-RUOT | 7.951 | 0.131 | 10.862 | 0.154 |
| UOT-FM | 8.114 | 0.035 | 9.170 | **0.011** |
| VGFM | 6.274 | 0.076 | 6.796 | 0.070 |
| WFR-FM | **5.486** | 0.012 | 6.211 | **0.011** |
| WFR-MFM | 5.925 | **0.009** | **5.548** | 0.023 |

In addition, we analyze the computational cost and scaling behavior of the mini-batch OET coupling step. While standard OT solvers typically scale as $\mathcal{O}(N^2)$ or worse, our mini-batch strategy computes couplings within batches of fixed size $B$, yielding a per-batch complexity of $\mathcal{O}(B^2)$. Since the number of batches scales as $N/B$, the total per-epoch complexity becomes

$$\mathcal{O}\left(\frac{N}{B} \cdot B^2\right) = \mathcal{O}(NB).$$

With fixed batch size $B$, the OET coupling step therefore scales linearly with the number of cells, i.e., $\mathcal{O}(N)$.

*Table 17.* Scaling behavior of the mini-batch OET coupling step on the Mouse dataset.

| Cells | Time (s) | ms/cell |
|---|---|---|
| 9.8k | 2.48 | 0.253 |
| 19.6k | 4.92 | 0.251 |
| 29.4k | 8.39 | 0.285 |
| 39.2k | 11.11 | 0.283 |
| 49.0k | 13.73 | 0.280 |

### D.12. Performance on the MEF Reprogramming Dataset

To further evaluate the scalability of our method, we conducted additional experiments on the large-scale mouse embryonic fibroblast (MEF) reprogramming dataset (Schiebinger et al., 2019a), which contains 251,203 cells collected densely across 39 time points over an 18-day reprogramming process. After removing cells without time-point annotations, the gene expression data were reduced to 100 dimensions using PCA.

As shown in Table 18, our method maintains exceptional inference efficiency even on this large-scale biological dataset. In particular, WFR-MFM achieves inference in only 0.035 seconds, substantially outperforming all competing methods. Compared with WFR-MFM, existing approaches exhibit slowdowns ranging from $108\times$ to over $1800\times$. These results demonstrate that the proposed one-step mean-field formulation provides significant computational advantages for large-scale trajectory modeling.

We acknowledge that benchmarking on even larger datasets and more complex biological systems would further strengthen the scalability analysis. We plan to investigate such large-scale applications in future work.

*Table 18.* Inference time on the MEF dataset. Slowdown is measured relative to WFR-MFM.

| Method | Inference Time (s) | Slowdown ($\times$) |
|---|---|---|
| MMFM | 4.013 | 114.65 |
| SF2M | 3.798 | 108.51 |
| VGFM | 8.047 | 229.91 |
| WFR-FM | 64.337 | 1838.20 |
| WFR-MFM | 0.035 | 1.00 |

### D.13. Performance on Simulation Perturbation Dataset

In this section, we simulate a synthetic gene regulatory network for a perturbation experiment. The dynamics of the system are governed by the following set of SDEs:

$$\frac{dX_1}{dt} = \frac{\rho_{1,c} + \alpha_1 X_1^2}{1 + \alpha_1 X_1^2 + \gamma_2 X_2^2 + \gamma_3 X_3^2} - \delta_1 X_1 + \eta_1 \xi_t,$$

$$\frac{dX_2}{dt} = \frac{\rho_{2,c} + \alpha_2 X_2^2}{1 + \gamma_1 X_1^2 + \alpha_2 X_2^2 + \gamma_3 X_3^2} - \delta_2 X_2 + \eta_2 \xi_t,$$

$$\frac{dX_3}{dt} = \frac{\rho_3 + \alpha_3 X_3^2}{1 + \alpha_3 X_3^2} - \delta_3 X_3 + \eta_3 \xi_t.$$

Here, $X_i(t)$ represents the concentration of gene $i$. The model features a toggle switch between $X_1$ and $X_2$ (mutual inhibition and self-activation), where both are further regulated by $X_3$ and an external signal $\beta$. The equations are parameterized by rates for basic transcription ($\rho_{i,.}$), self-activation ($\alpha_i$), inhibition ($\gamma_i$), and degradation ($\delta_i$), with an additional stochastic noise term ($\eta_i \xi_t$). In addition, we incorporate probabilistic cell death, which is dependent on the expression of $X_3$. The instantaneous death rate $g$ is defined as

$$g = \alpha_{g,c} \frac{X_3^2}{1 + X_3^2}.$$

which depends on condition-specific parameter $\alpha_{g,c}$. Each perturbation condition $c$ corresponds to a triplet of parameters $(\rho_{1,c}, \rho_{2,c}, \alpha_{g,c})$, randomly sampled from predefined ranges. From the full parameter space, we generate 5100 perturbation conditions. A subset of 100 conditions is selected to form the training set, and the remaining 5000 are reserved for evaluation. This construction emulates realistic experimental settings where only a limited set of perturbations is experimentally measured, and the goal is to infer system behavior under a much larger number of unseen perturbations.

In this dataset, for each perturbation condition $c$, we set the WFR mass variation penalty $\delta_c = N_c/N_{\text{ctrl}}$, where $N_c$ and $N_{\text{ctrl}}$ denote the numbers of cells under condition $c$ and the unperturbed condition, respectively. The proportion of cross-time sampling is fixed to $p_{\text{diff}} = 0.5$, and the mass-growth regularization coefficient is set to $\lambda = 0.1$.

