# OpenReview forum: "WFR-MFM: One-Step Inference for Dynamic Unbalanced Optimal Transport"
_ICML.cc/2026/Conference — ICML 2026 regular_

### Official Review · Reviewer_pSZQ · 2026-03-05

**Soundness:** 3
**Presentation:** 2
**Significance:** 2
**Originality:** 2
**Overall Recommendation:** 4
**Confidence:** 3

**Summary:**

This paper addresses the inference bottleneck in dynamic unbalanced optimal transport (OT) for single-cell biology applications. Existing methods for modeling coupled transport and mass variation — such as WFR-FM — require iterative ODE simulation at inference time, which becomes prohibitively expensive at scale. The authors propose a mean-flow framework for unbalanced flow matching that introduces time-averaged mean velocity and mean mass-growth fields, enabling direct one-step state-to-state updates without trajectory simulation. Specializing this framework to the Wasserstein–Fisher–Rao (WFR) geometry yields WFR-MFM, which learns mean-flow fields along WFR geodesics via a conditional regression objective with stop-gradient targets. The training objective is shown to be equivalent (up to a constant) to the unconditional objective (Theorem 4.1), and a key remark explains why a naive time-averaging formulation of the conditional targets would break this equivalence. The method is validated across synthetic and real scRNA-seq datasets, demonstrating 2–3 orders-of-magnitude inference speedup over ODE-based baselines while maintaining competitive predictive accuracy. A conditional extension is applied to a large-scale synthetic perturbation benchmark with 5000 unseen conditions.

**Compliance With Llm Reviewing Policy:**

Affirmed.

**Key Questions For Authors:**

1. **When does one-step inference suffice?** Can you provide theoretical or empirical guidelines for predicting when one-step inference will be adequate versus when multi-step is needed? A principled criterion (e.g., based on the degree of mass variation or transport distance between marginals) would significantly strengthen the practical utility and could raise my assessment of significance.

2. **How does WFR-MFM perform on real perturbation data?** The perturbation experiment is the most compelling application story, but uses only synthetic data. Have you evaluated on any real perturbation dataset (e.g., sci-Plex, Dixit et al.)? Even a small-scale real experiment would substantially change my assessment of practical impact.

3. **What is the computational cost of the OET coupling step at scale?** For the largest real dataset (Mouse, ~49K cells), what fraction of total training time does OET consume? How does OET scale with cell count, and does the mini-batch strategy maintain coupling quality for datasets with >100K cells? This bears directly on whether the claimed scalability advantages hold end-to-end.

**Limitations:**

Yes — the authors discuss key limitations in the conclusion, including the absence of recent mean-flow refinements, the restriction to synthetic perturbation data, and the limitation to WFR geometry. The discussion could be strengthened by explicitly acknowledging (a) the accuracy degradation observed on certain datasets under one-step inference, and (b) the potential scaling limitations of the OET preprocessing step for very large datasets.

**Strengths And Weaknesses:**

### Strengths:

S1. **Clear problem motivation and practical impact (Significance).** The inference bottleneck in unbalanced OT is a genuine and important limitation for scalable single-cell applications, particularly perturbation prediction where the number of conditions grows combinatorially. The demonstrated speedups (78–2965x over adaptive RK solvers, ~100x over Euler; Table 1) are compelling and hard to dismiss.

S2. **Sound theoretical framework (Soundness/Originality).** The mean-flow formulation for unbalanced dynamics is mathematically principled. Theorem 4.1 showing equivalence of unconditional and conditional objectives is correctly derived (best-effort verification). Remark 4.2 and Proposition B.1 — showing that naive time-averaged conditional targets break this equivalence — add genuine insight and justify the specific construction.

S3. **Comprehensive experimental evaluation (Soundness).** The paper compares against 10 baselines across 7 datasets spanning synthetic and real biological data. Extensive ablation studies on hyperparameters delta, p_diff, and lambda (Tables 4–6) demonstrate robustness properties. The multi-step speed–accuracy trade-off analysis is informative.

S4. **Honest reporting of limitations (Presentation).** The paper transparently reports cases where WFR-MFM underperforms WFR-FM (e.g., CITE-seq in Table 13 where one-step inference is substantially worse, improving only with 10 steps). This builds trust in the reported results.

### Weaknesses:

W1. **Incremental novelty over prior work (Originality).** The contribution is essentially a combination of MeanFlow (Geng et al., 2025a), which provides the mean-flow framework for mass-preserving FM, and WFR-FM (Peng et al., 2026a), which provides the WFR-geometric coupling. The paper itself acknowledges (Appendix A.1) that "most constructions in this subsection follow WFR-FM; the mean-flow training objective is the main departure." The proof of Theorem 4.1 mirrors the MeanFlow equivalence proof adapted to include the growth term. Given that both MeanFlow and WFR-FM originate from overlapping research groups, the question of whether the combination constitutes a sufficiently distinct contribution is relevant.

W2. **Accuracy trade-off is non-negligible on several datasets (Soundness).** While the paper frames WFR-MFM as maintaining "high predictive accuracy," the evidence is more nuanced. On CITE-seq (Table 13), one-step WFR-MFM yields W1 = 30.841 versus WFR-FM's 27.831 — a ~11% degradation requiring 10 steps to recover. On Dyngen at t=4 (Table 7), WFR-MFM has RME = 0.111 versus WFR-FM's 0.002, a 63x worse mass estimation. On EMT (Table 8), WFR-MFM consistently trails by 7–8% in W1. These gaps deserve more explicit discussion rather than being subsumed under a general claim of "maintained accuracy."

W3. **Perturbation experiment limited to synthetic data (Significance).** The large-scale perturbation prediction experiment (Section 5, Q4) — arguably the most compelling application — uses only a simplified synthetic gene regulatory network with 3 genes. The gap between this and real perturbation datasets (thousands of genes, complex regulatory networks, substantial biological noise) is large. This significantly weakens the practical impact claim for the perturbation use case.

W4. **No formal characterization of approximation quality (Soundness).** The paper provides no theoretical analysis of how one-step approximation error relates to the complexity of the underlying dynamics (e.g., curvature of WFR geodesics, degree of mass variation). The empirical results (CITE-seq, Dyngen) show that one step can be insufficient, but there are no guarantees or guidelines on when one-step inference is adequate versus when multi-step is necessary.

---

> ### Author Rebuttal · Authors · 2026-03-31
>
> **W1. Limited novelty.**
> We thank the reviewer for this question. Our key focus in this paper is the sampling efficiency, which is crucial in perturbation-response prediction. The proposed WFR-MFM achieves this goal and paves the way for further challenging  applications. This is a big success, as we consider.
>
> **Q1**
> We provide an explicit adaptive stopping criterion.
>
> Let $p_0$ be the empirical distribution at $t=0$ with $N_0$ cells and $m_0 = 1/N_0$. Let
> $s^{(K)} = \{(x_{1,n}^{(K)}, m_{1,n}^{(K)})\}_{n=1}^N$  be the result after $K$ steps.
>
> Define the discrepancy between iterates:
> $$
> \Delta_k = d(s^{(k-1)}, s^{(k)}),
> $$
>
> $$
> d = \frac{1}{N} \sum_{n=1}^N
> \left[
> \frac{\|x^{(k)} - x^{(k-1)}\|^2}{\|\mathbb{E}_{p_0}[x]\|^2}
> +
> \frac{|m^{(k)} - m^{(k-1)}|^2}{m_0^2}
> \right]
> $$
>
> This per-sample $\ell_2$ discrepancy efficiently approximates convergence, avoiding costly metrics such as $\mathcal{W}_1$.
>
> The specific adaptive stopping rule is as follows.
>
> **Adaptive stopping rule**
> - **Input:** maximum step count $K_{\max}$, tolerance $\tau \in (0,1)$, patience $P \in \mathbb{N}$, distance function $d(\cdot,\cdot)$
> - **Output:** selected iterate $s^{(k_{\mathrm{best}})}$ and best step count $k_{\mathrm{best}}$
>
> 1. Initialize
>    $k_{\mathrm{best}} \gets 1$, $\Delta_{\mathrm{best}} \gets+\infty$, $c \gets 0$
>
> 2. For $k = 1,2,\dots,K_{\max}$:
>    - Compute $s^{(k)}$
>    - If $k > 1$:
>      - $\Delta_k \gets d(s^{(k-1)}, s^{(k)})$
>      - If $\Delta_k < (1-\tau)\,\Delta_{\mathrm{best}}$:
>        - $\Delta_{\mathrm{best}} \gets \Delta_k$
>        - $k_{\mathrm{best}} \gets k$
>        - $c \gets 0$
>      - Else:
>        - $c \gets c + 1$
>      - If $c \ge P$:
>        - Return $s^{(k_{\mathrm{best}})}, k_{\mathrm{best}}$
> 3. Return $s^{(k_{\mathrm{best}})}, k_{\mathrm{best}}$
>
> Here, $\tau$ controls improvement strictness (smaller $\tau$ → more steps), and $P$ is the number of tolerated non-improving steps.
>
> On the 2D Gene dataset, $P=2$ and $\tau=0.15$ stop at $K=22$, achieving near-optimal $\mathcal{W}_1$ and RME (Fig.1) using  the $\ell_2$ discrepancy.
>
> **Q2**
>  Extending WFR-MFM to such datasets is ongoing work that we are preparing as a separate study.
> To address the reviewer’s interest, we provide a preliminary evaluation on a PBMC perturbation dataset [1] with $\sim 10$M cells from 12 donors and 90 cytokine perturbations.
> We hold out 20% of perturbation conditions for testing, use 5,412 HVGs with 100-d PCA, and encode conditions into 4,096-d embeddings.  As this analysis is preliminary, we do not yet include baseline comparisons. These will be systematically evaluated in follow-up work. Results averaged across conditions are shown below:
>
> **Results on PBMC**
> | Metric         | Train  | Test   |
> |----------------|--------|--------|
> | Gene MSE       | 7.4e-4 | 2.2e-3 |
> | Gene MAE       | 0.0145 | 0.0194 |
> | Gene $R^2$     | 0.987  | 0.962  |
> | DEG50 $R^2$    | 0.920  | 0.625  |
> | $W_1$          | 0.140  | 0.142  |
> | KL             | 6.64   | 6.83   |
>
> WFR-MFM shows consistent train/test performance, suggesting generalization to real perturbations. These results are preliminary and provided in response to the reviewer. A full study will be reported separately.
>
> [1] Parse Biosciences. *10 Million Human PBMCs in a Single Experiment*. 2024.
>
> **Q3**
> We address the cost, scaling, and quality of our mini-batch OET strategy below.
>
> **Linear Scaling with Cell Count:**
> While standard OT algorithms scale at $\mathcal{O}(N^2)$ or worse, our mini-batch OET strategy with a fixed batch size $B$ computes a coupling per batch in $\mathcal{O}(B^2)$. With $N/B$ batches per epoch, the total per-epoch cost is $\mathcal{O}(B^2 \times \frac{N}{B}) = \mathcal{O}(B N)$. Since $B$ is constant, our OET coupling step scales *linearly*, $\mathcal{O}(N)$.
>
> We subsample the Mouse dataset; the table below shows linear runtime scaling and nearly constant per-cell cost.
>
> | Cells  | Time (s) | ms/cell |
> |--------|----------|---------|
> | 9.8k   | 2.48     | 0.253   |
> | 19.6k  | 4.92     | 0.251   |
> | 29.4k  | 8.39     | 0.285   |
> | 39.2k  | 11.11    | 0.283   |
> | 49.0k  | 13.73    | 0.280   |
>
> **Fraction of Training Time:**
> For the Mouse dataset, the OET step accounts for approximately 51% of the total training time under the full-batch setting.
>
> **Coupling Quality and Mini-Batch Dynamics:**
> While mini-batch OT convergence to the true global coupling in unbalanced settings remains an open theoretical problem [1], batch size serves as a practical trade-off between computational efficiency and strict global fidelity. Importantly, mini-batch approximations do not prevent WFR-MFM from successfully matching the marginals to the data distributions. Despite potential slight deviations from the global optimal plan, the approach remains valid and practical for learning the underlying dynamics.
>
> [1] Sommerfeld M, et al. Optimal transport: Fast probabilistic approximation with exact solvers. *Journal of Machine Learning Research*, 2019.

---

> > ### Author Rebuttal · Reviewer_pSZQ · 2026-03-31
> >
> > I thank the authors for the substantive responses. Two of the three questions are well addressed:
> >
> > - **Q1 (adaptive stopping criterion):** This is a genuine and useful contribution that directly addresses W4. The criterion is simple, practical, and validated on the Gene dataset. I appreciate this addition.
> > - **Q3 (OET scaling):** The linear scaling analysis with empirical data is convincing. The 51% training time figure is informative — it is non-negligible but acceptable given the inference speedup.
> >
> > Two points require follow-up:
> >
> > 1. **Q2 (real perturbation data):** The preliminary PBMC results are encouraging (Gene R² = 0.962 on test), but without any baseline comparison, these numbers are difficult to interpret. What is the Gene R² for a simple baseline (e.g., zero-change or linear shift) on this dataset? Even one number would help contextualize whether 0.962 is trivially achievable or genuinely strong. This would meaningfully affect my assessment of W3 (significance of the perturbation use case).
> >
> > 2. **W2 (accuracy degradation):** The response does not address the accuracy trade-off on CITE-seq (11% W1 degradation) and Dyngen (63x worse RME). The adaptive stopping criterion helps for *choosing* when to use more steps, but does not change the underlying accuracy gap when one step is used. Could the authors clarify whether the revised framing acknowledges this trade-off more explicitly?
> >
> > I maintain my score of **4** pending these clarifications.

---

> > > ### Author Response · Authors · 2026-04-08
> > >
> > > **Q2**
> > >
> > > We thank the reviewer for this insightful suggestion, which helped us better understand the PBMC dataset.
> > > We therefore evaluated a zero-change baseline, which achieves Gene $R^2 = 0.967$ on test, slightly higher than WFR-MFM ($R^2 = 0.962$).
> > >
> > > **Interpretation**
> > >
> > > On the PBMC dataset, most genes exhibit only mild expression changes under perturbations, and the use of $5,412$ HVGs further includes many weakly responsive genes. In this regime, predictions close to the control state can already achieve high Gene $R^2$, as they capture near-invariant signals. This phenomenon is also observed in [1] (Fig. S1), where the identity (zero-change) baseline performs well for subtle perturbations but fails to capture stronger effects $R^2$. Therefore, the observed performance gap does not indicate a limitation of WFR-MFM, but rather reflects the interaction between dataset characteristics and the evaluation metric.
> > >
> > > **Further results**
> > >
> > > Restricting to $2{,}000$ HVGs (commonly used in PBMC studies [1,2]) yields Gene $R^2 = 0.949$ for WFR-MFM and $0.939$ for zero-change. Compared to the $5{,}412$-HVG setting, both scores decrease, but zero-change drops more, making WFR-MFM superior. This is consistent with our interpretation: the larger HVG set contains many weakly responsive genes that inflate $R^2$ for near-identity predictions, while focusing on more perturbation-relevant genes reduces this effect and reveals the advantage of WFR-MFM.
> > >
> > >
> > > To further demonstrate the capability of WFR-MFM, we evaluate on the Norman dataset [3], which comprises 91,205 cells and 5,045 genes. We train on all genes without HVG selection, use 2560-dimensional condition embeddings, and project the data into a 100-dimensional PCA space. The dataset is split into 139 training conditions and 107 test conditions. On the test set, WFR-MFM achieves higher performance ($R^2 = 0.9704$) than the zero-change baseline ($R^2 = 0.9584$).
> > >
> > > [1] Klein D, Fleck J S, Bobrovskiy D, et al. CellFlow enables generative single-cell phenotype modeling with flow matching. bioRxiv, 2025.
> > >
> > > [2] Yuan, Xinyu, et al. "Perturbdiff: Functional diffusion for single-cell perturbation modeling." arXiv preprint arXiv:2602.19685 (2026).
> > >
> > > [3] Norman, Thomas M., et al. Exploring genetic interaction manifolds constructed from rich single-cell phenotypes. Science, 2019.
> > >
> > >
> > > **W2**
> > >
> > >  We thank the reviewer for pointing out the accuracy degradation in one-step inference on datasets such as CITE-seq and Dyngen, which we agree is an important aspect that deserves clearer discussion.
> > >
> > > We agree that there exists an inherent accuracy--efficiency trade-off in one-step inference, and we now make this point explicit in our revised framing. First, we provide a brief convergence analysis demonstrating that the learned velocity and mass-growth fields converge to their respective true mean fields. Due to space limitations, we refer the reviewer to our initial response to Reviewer GUbx (Q2) for a more detailed discussion.
> > >
> > > Notice that the error dynamics are:
> > >
> > > $$
> > > e^{(n+1)}(x, t, T)
> > > =(T - t)E_{z \mid x, t, T}
> > > \left[
> > > \frac{d}{dt} e^{(n)}(x, t, T)
> > > \right].
> > > $$
> > >
> > > This implies that when $(t,T)$ are close, the error readily converges to 0; conversely, when $(t,T)$ are far apart, convergence becomes more difficult. Consequently, the learned mean fields $v_{\theta}(x, t, T)$ and $h_{\phi}(x, t, T)$ are more accurate over smaller intervals of $(t,T)$.
> > >
> > > In the one-step setting, we use $(t,T)=(0,1)$ and directly apply
> > > $v_{\theta}(x, 0, 1)$ and $h_{\phi}(x, 0, 1)$ for inference. In this regime, the approximation error can be larger, potentially leading to lower accuracy compared to WFR-FM.
> > >
> > > In contrast, a multi-step strategy with $N$ steps and interval $\Delta t \triangleq \frac{1}{N}$ uses
> > >
> > > $$
> > > \\{v_{\theta}(x, i\Delta t, (i+1)\Delta t),
> > > h_{\phi}(x, i\Delta t, (i+1)\Delta t) \\}_{i=0}^{N-1}
> > > $$
> > >
> > > which improves accuracy. This explains the observed degradation on datasets such as CITE-seq and Dyngen.
> > >
> > > We will revise the paper to explicitly highlight this trade-off: one-step inference prioritizes efficiency, while accuracy may degrade when the temporal gap is large.

---

### Official Review · Reviewer_cq4S · 2026-03-11

**Soundness:** 2
**Presentation:** 3
**Significance:** 3
**Originality:** 2
**Overall Recommendation:** 4
**Confidence:** 3

**Summary:**

The authors propose WFR-MFM, a mean-flow matching framework for solving dynamic unbalanced optimal transport under the Wasserstein–Fisher–Rao (WFR) geometry. The mean-flow framework allows for fast one-step inference. Further, the author uses a growth function that models the cell growth and death.

**Compliance With Llm Reviewing Policy:**

Affirmed.

**Final Justification:**

Most of my concerns and questions are addressed. The authors gave a robust response by including experiments with high dimensional data. Also, the authors pointed me to more ablation studies that I previously missed.

**Key Questions For Authors:**

1. Can you show more ablation studies in real datasets that shows the contribution of the growth function?

2. Could you evaluate on real high dimensional biological datasets (at least 1000 dims).

3. What are some additional values of modeling the growth function if we could directly tackle if the OT coupling is under Katorovich relaxation?


I am happy to increase score if questions are addressed.

**Limitations:**

yes

**Strengths And Weaknesses:**

**Strength:**

1. The paper tackles the slow inference time for flow matching problems, which is to some extent meaningful. Many flow matching methods indeed suffer from slow inference time when the control batch size is large.

2. Authors uses growth function to solve the unbalanced OT problem in Biology.

3. The paper is technically sound in methodology.

4. The paper is clearly and well presented.

**Weakness:**

1. The real biological data used for validation are all very low dimensional. `To show scalability, it is advisable to show strong results in high dimensional biological datasets.

---

> ### Author Rebuttal · Authors · 2026-03-31
>
> **Q1**
> We thank the reviewer for this question on the contribution of the growth function. We conduct ablation studies on real datasets (EMT, EB, Mouse), summarized below, which exhibit varying levels of population change. We compare:
> (i) **OT-CFM (w/o growth)**: mass-conserving baseline;
> (ii) **WFR-MFM ($\delta \to \infty$)**: mass-conserving limit reducing to OT and a one-step mean-flow scheme;
> (iii) **WFR-MFM ($g=0$)**: UOT coupling without growth, isolating transport;
> (iv) **WFR-MFM**: UOT with learned growth.
>
> As shown below, WFR-MFM consistently outperforms all mass-conserving or no-growth variants. Improvements are more pronounced on datasets with stronger mass variation, where mass-conserving methods incur large errors in both $\mathcal{W}_1$ and RME (e.g., Mouse). In the Mouse dataset, cell counts change drastically across time points (4638 $\to$ 14985 $\to$ 29679). This shows that (i) mass-conserving assumptions lead to systematic mismatches in population size, and (ii) modeling growth improves accuracy.
>
> We further ablate $g \equiv 0$, reducing the model to mass-conserving transport. In this case, RME matches mass-conserving baselines (e.g., OT-CFM), while the velocity field remains unchanged, isolating the effect of growth.
>
> Overall, modeling growth consistently improves performance, especially in settings with significant proliferation or death.
>
> **Ablation study on real datasets (EMT, EB, Mouse) evaluated on held-out timepoints**
>
> | Method | EMT W1 | EMT RME | EB W1 | EB RME | Mouse W1 | Mouse RME |
> |--------|--------|---------|--------|---------|-----------|------------|
> | OT-CFM | 0.3039 | 0.3079 | 10.6228 | 0.3507 | 7.0967 | 0.6219 |
> | WFR-MFM ($\delta \to \infty$) | 0.3755 | 0.1762 | 12.5584 | 0.2288 | 10.6822 | 0.5670 |
> | WFR-MFM ($g=0$) | 0.2994 | 0.3079 | 10.3176 | 0.3507 | 6.7646 | 0.6219 |
> | WFR-MFM | **0.2990** | **0.1750** | **10.1350** | **0.2144** | **6.7140** | **0.0655** |
>
> **Q2**
> We thank the reviewer for raising this important question regarding evaluation on high-dimensional biological data. In single-cell analysis, it is a standard strategy to model dynamics in a low-dimensional latent space (e.g., PCA, autoencoders, VAEs) due to noise, sparsity, and redundancy in raw gene expression [1,2,3]. Following this practice, our original manuscript performs modeling and evaluation in latent space for more stable comparisons.
>
> To further address this, we evaluate on the 2000D CITE dataset by decoding predictions to gene expression space and computing $\mathcal{W}_1$ and RME. The results below show that WFR-MFM consistently outperforms baselines.
>
> **Results on the 2000D CITE dataset**
>
> | Method | $\mathcal{W}_1$ (t=1) | $\mathcal{W}_1$ (t=2) | $\mathcal{W}_1$ (t=3) |
> |--------|------------------------|------------------------|------------------------|
> | MMFM | 42.314 | 46.996 | 72.802 |
> | SF2M | 42.568 | 50.009 | 78.588 |
> | Var-RUOT | 41.230 | 37.645 | 54.266 |
> | VGFM | 41.058 | 37.591 | 53.354 |
> | WFR-FM | 41.084 | 39.342 | 58.024 |
> | WFR-MFM | 40.513 | 37.319 | 51.587 |
> | WFR-MFM (10 steps) | **39.837** | **36.327** | **49.357** |
>
> [1] Lopez R, Regier J, Cole M B, et al. *Deep generative modeling for single-cell transcriptomics*. Nature Methods, 2018.
> [2] Lotfollahi M, Wolf F A, Theis F J. *scGen predicts single-cell perturbation responses*. Nature Methods, 2019.
> [3] Klein D, Fleck J S, Bobrovskiy D, et al. *CellFlow enables generative single-cell phenotype modeling with flow matching*. bioRxiv, 2025.
>
> **Q3**
> We thank the reviewer for this question on the additional value of modeling the growth function beyond OT couplings under Kantorovich relaxation. While unbalanced OT handles mass variations via static mappings, explicitly modeling continuous dynamics (vector field + growth function) offers critical advantages unattainable by OT couplings:
>
> - **Predictive Generalization:** OT is restricted to mapping observed distributions. Our learned dynamical model captures underlying biological rules, enabling reliable prediction of forward trajectories for completely unseen cells.
>
> - **Continuous Interpolation:** Unlike OT's discrete point-to-point correspondence, our continuous WFR-MFM framework captures non-equilibrium flows, allowing accurate simulation of biologically plausible intermediate states at any arbitrary time.
>
> - **Biological Insights:** Explicit, differentiable velocity and growth functions enable deeper mechanistic analyses (e.g., evaluating state-space growth rates or Jacobians) to identify key driver genes, yielding actionable insights beyond mere geometric mapping [1].
>
> [1] Qiu X, et al. *Mapping transcriptomic vector fields of single cells*. Cell, 2022.

---

> > ### Author Rebuttal · Reviewer_cq4S · 2026-04-01
> >
> > I thank the author for the response. Most of my concerns are addressed. Hence, now I give a score of 4.

---

### Official Review · Reviewer_GUbx · 2026-03-12

**Soundness:** 3
**Presentation:** 3
**Significance:** 2
**Originality:** 2
**Overall Recommendation:** 5
**Confidence:** 3

**Summary:**

This work introduces WFR-mean flow matching (WFR-MFM), an extension of the WFR-flow matching framework (Peng et al., 2026) designed to address the computational bottlenecks of trajectory-based inference in unbalanced optimal transport problems, WFR formulation. Borrowing from mean flow matching, the authors replace the instantaneous velocity field and growth rate with their time-averaged counterparts, which enables rapid, one-step inference without requiring iterative ODE simulation.  The proposed method is evaluated across diverse benchmarks, including synthetic gene expression datasets, high-dimensional Gaussian mixtures, and real-world data such as scRNA-seq and CITE-seq datasets. WFR-MFM demonstrates inference speedups of up to two orders of magnitude compared with WFR-FM while maintaining high predictive accuracy and precisely modeling mass variation.

**Compliance With Llm Reviewing Policy:**

Affirmed.

**Final Justification:**

The authors have provided a detailed rebuttal that addresses my questions and resolves most of my concerns. I will increase my score.

**Key Questions For Authors:**

- When performing multi-step inference ($T > 1$), are the mean velocity and mean mass-growth fields calculated as a single average across the entire time span of the experiment, or are they conditioned on specific start ($t$) and end ($T$) time pairs to allow for time-varying dynamics?

- How does the use of stop-gradient operators in the training loss for the derivative identities prevent the model from collapsing into a trivial solution during the regression of mean fields?

- When comparing WFR-MFM to standard WFR-FM, does the one-step mean-flow approach miss any fine-grained local transitions or branching events in complex sub-populations that a multi-step ODE solver might capture? How does the model prevent over-smoothing of trajectories?

- While WFR-MFM is orders of magnitude faster at inference, does it require a longer training time to converge the mean fields compared to the instantaneous fields used in WFR-FM?

**Limitations:**

The current formulation of WFR-MFM is defined based on the existence of well-defined Dirac-to-Dirac (point-to-point) flows within the transport geometry. While this assumption holds for the WFR framework, extending it with the mean-flow approach to other complex unbalanced transport formulations may not be feasible if these systems do not admit similar closed-form geodesic descriptions.

**Strengths And Weaknesses:**

**Strengths:**
- The paper effectively integrates the Wasserstein–Fisher–Rao (WFR) geometry with the mean Flow framework originally developed for mass-preserving systems. Building on the WFR-FM framework, the authors extend simulation-free flow matching to the unbalanced setting by introducing time-averaged mean velocity and growth fields.

- The mean-flow formulation substantially improves the computational efficiency of the unbalanced WFR framework while maintaining strong predictive performance.

- The model allows the number of inference steps to be adjusted, enabling finer approximations of the underlying continuous dynamics when needed.

- The method is evaluated on eight diverse datasets, including both synthetic and real-world benchmarks.

- The authors present an extensive ablation study, including a sensitivity analysis of key hyperparameters. The results show that the model’s performance remains stable across a wide range of values for the loss weight ($\lambda$) and the cross-time sampling proportion ($p_{diff}$), while highlighting the importance of properly tuning ($\delta$) to balance spatial transport and mass variation.

- The paper is well structured and generally easy to follow.



**Weakness:**
- *Limited novelty:* While the combination of WFR-MF and mean flow is well executed, the novelty appears limited, as most derivations and findings rely heavily on the WFR-MF framework.

- *Benchmarking on small datasets:* Many of the datasets used for primary evaluation are relatively small and much smaller than typical scRNA-seq datasets, which often contain hundreds of thousands of cells. The method’s scalability to large datasets remains unclear.

- *Sensitivity to $\delta$:* As noted by the authors, performance is highly sensitive to the choice of $\delta$, which is treated as a fixed constant across the entire trajectory. In real biological systems, proliferation and apoptosis rates are time-dependent—for example, a differentiation process may exhibit a growth burst at $t=1$ and almost none at $t=2$. A constant $\delta$ risks misrepresenting growth dynamics at different stages of the same process.

- *Temporal resolution gap:* By replacing continuous trajectories with direct state-to-state updates, it is unclear how the mean-flow assumption handles cells undergoing rapid, non-linear changes within a single time point.

---

> ### Author Rebuttal · Authors · 2026-03-31
>
> **W1. Limited novelty.**
> We thank the reviewer for this question. Our key focus in this paper is the sampling efficiency, which is crucial in perturbation-response prediction. The proposed WFR-MFM achieves this goal and paves the way for further challenging  applications. This is a big success, as we consider.
>
> **Q1**
> We thank the reviewer for this question. The mean velocity and mean mass-growth fields are conditioned on specific start ($t$) and end ($t+1$) time pairs, allowing the model to capture time-varying dynamics across the trajectory.
>
> **Q2**
> We thank the reviewer for this question on how the use of stop-gradient operators prevents collapse to trivial solutions during the regression of mean fields.
> We only discuss the velocity field $v$, since the same argument applies to $h$.
> Consider the loss
>
> $$
> L_c(\theta) = E_{t<T, z, x} [ ||v_\theta(x,t,T) - sg(v(x,t,T|z))||^2 ]
> $$
>
> where
>
> $$
> v(x,t,T|z) = u_t(x) + (T-t)\frac{d}{dt}v(x,t,T)
> $$
>
> $$
> = u_t(x|z) + (T-t)\Big[\partial_t v_\theta(x,t,T)+ (\nabla_x v_\theta(x,t,T))u_t(x|z)
> \Big]
> $$
>
> The stop-gradient operator treats the target $\mathrm{sg}(v(x,t,T \mid z))$ as a constant per iteration. Thus, minimizing $\mathcal{L}_{\mathrm c}$ reduces to regression against a fixed target:
>
> $$
> v_{\theta}(x,t,T)=\mathbb{E}_{z \mid x, t, T}
> \left[\mathrm{sg}\big(v(x,t,T \mid z)\big)\right].
> $$
>
> Consider the corresponding iterative scheme:
>
> $$v^{(n+1)}(x,t,T)=\mathbb{E}_{z \mid x,t,T}\left[u_t(x \mid z)+(T-t)\,\frac{d}{dt}v^{(n)}(x,t,T)
> \right].
> $$
>
> Let $v^\star$ be the true target field satisfying:
>
> $$
> v^\star(x, t, T)=
> \mathbb{E}_{z|x,t,T}[
> u_t(x|z)+(T-t)\frac{d}{dt}v^\star(x,t,T)]
> $$
>
>
> Defining the error $e^{(n)} \triangleq v^{(n)} - v^\star$, the error dynamics are:
>
> $$
> e^{(n+1)}(x, t, T)=(T-t)\,\mathbb{E}_{z \mid x, t,T}
> \left[\frac{d}{dt}e^{(n)}(x,t,T)\right].
> $$
>
> Note that in general one can not bound $\Vert\frac{d}{dt}{e}^{(n)}\Vert$ by $\Vert{e}^{(n)}\Vert$. However, upon assuming
>
> $$
> \left\Vert \frac{d}{dt} {e}^{(n)} \right\Vert_{L^\infty}\le
> L \left\Vert {e}^{(n)} \right\Vert_{L^\infty}
> $$
> for $L > 0$, e.g., for a band-limited function, we get
>
> $$
> \Vert e^{(n+1)}(x, t, T) \Vert_{L^\infty} \le L \vert T-t \vert \cdot \Vert e^{(n)} \Vert_{L^\infty}.
> $$
>
> Taking the expectation over the training time-step distribution $p_{\mathrm{data}}(t, T)$ yields the expected error convergence:
>
> $$
> E_{t, T \sim p_{\mathrm{data}}}
> \left[\Vert e^{(n+1)} \Vert_{L^{\infty}}\right]\le E_{t, T \sim p_{\mathrm{data}}}\big[L \vert T-t \vert\big]\cdot
> \Vert e^{(n)} \Vert_{L^{\infty}}.
> $$
>
> If
>
> $$
> E_{t, T} [L \vert T-t \vert] \le L_0 < 1,
> $$
>
> the scheme becomes a contraction mapping:
>
> $$
> E \Vert e^{(n+1)} \Vert_{L^\infty}\le L_0 \cdot \mathbb{E} \Vert e^{(n)} \Vert_{L^\infty},
> $$
>
> which guarantees convergence to $v^\star$.  Notably, $v^\star \equiv 0$ is not a fixed point unless $u_t \equiv 0$.
>
> Empirically, we set $p_{\mathrm{diff}} \in [0.05,0.3]$. Thus, $T-t=0$ for most samples, and $t,T \sim U[0,1]$ otherwise.
> This makes $\mathbb{E}[|T-t|]$small, which might ensure $\mathbb{E}[L|T-t|] < 1$ in the above analysis, aligning with our contraction theory and observed stability.
> However, we remark that a fully rigorous analysis of the MFM is beyond the scope of this paper.
>
> **Q3**
>  We thank the reviewer for this insightful question. For accurate final-state prediction, the one-step mean-flow formulation is sufficient and offers substantial efficiency gains. For scenarios where finer local transitions or branching structures are of interest, the framework can be naturally extended to a multi-step procedure, allowing more detailed trajectory refinement. Thus, WFR-MFM does not inherently over-smooth, but provides a flexible trade-off between efficiency and resolution.
>
>  **Q4**
> We thank the reviewer for the question on training efficiency. As shown in Fig. 3 and Fig. 7, WFR-MFM introduces only a modest training overhead compared to WFR-FM (2m11s vs. 1m28s on the 100D EB dataset), while yielding substantial inference speedups. This overhead is negligible given the orders-of-magnitude acceleration at inference.

---

> > ### Author Rebuttal · Reviewer_GUbx · 2026-04-04
> >
> > I thank the authors for their detailed response.
> > Thanks for the derivation provided for Q2. The proof characterizing the training objective as a contraction mapping is helpful and clarifies why the sg(.) operator ensures convergence to a non-trivial fixed point.
> > Also thanks for the empirical details regarding training overhead and the flexibility of the multi-step procedure.
> >
> > Regarding temporal resolution, the authors state that multi-step procedures can be used for finer local transitions. Does it imply that samples undergoing faster changes require more intermediate sampling? I think this could be problematic whene there are subgroup of samples with different kinetic profiles (growth rates).
> >
> > Still some of my primary concerns remain unaddressed in the rebuttal:
> >
> > Scalability and Dataset Size: My concern regarding the benchmarking on relatively small datasets (small N) was not addressed. If the main success and contribution is computational efficeienty, should not be show cased for large-scale datasets?
> >
> > Sensitivity to $\gamma$: The issue regarding a fixed $\gamma$ (growth/death rate) across the entire trajectory remains.

---

> > > ### Author Response · Authors · 2026-04-08
> > >
> > > We thank the reviewer for these insightful comments.
> > >
> > > **W1**
> > >
> > > We remark that as the multi-step strategy presents more accurate results, it is not a bottleneck for resolving fast kinetic process. A simple explanation is to consider the numerical methods for stiff ODEs, in which one does not need to resolve the transient details even they could change very fast. Here, the mean flow does not need to resolve the transient details, either, as it provides a solution operator instead of a traditional temporal discretization. In short, the temporal resolution is not necessary for fast kinetic process.
> > >
> > > **Q1**
> > >
> > > We perform repeated evaluations to ensure stability. On the largest dataset in Table 1 (Mouse hematopoiesis, $\sim$49K cells), results are averaged over 100 runs, with total inference time of 0.21s for WFR-MFM versus 344s for WFR-FM (RK).
> > >
> > > On a simulated perturbation dataset, WFR-MFM completes inference for 5,000 unseen conditions on 10,000 cells in 6.55s (i.e., 5,000 repeated evaluations), compared to an estimated $\sim$2,000s for WFR-FM. This setting reflects virtual perturbation tasks requiring large-scale repeated generation, where WFR-MFM is specifically designed and shows clear advantages.
> > >
> > > To further assess scalability, we evaluate on a large-scale MEF reprogramming dataset [1] with 251,203 cells across 39 time points (100D PCA after filtering). Results are:
> > >
> > > | Method              | Inference Time (s) | Slowdown (×) |
> > > |--------------------|-------------------|--------------|
> > > | MMFM               | 4.013             | 114.65       |
> > > | SF2M               | 3.798             | 108.51       |
> > > | VGFM               | 8.047             | 229.91       |
> > > | WFR-FM (RK)        | 64.337            | 1838.20      |
> > > | WFR-FM (Euler)     | 4.357             | 124.49       |
> > > | WFR-MFM            | 0.035             | 1.00         |
> > >
> > > WFR-MFM remains orders of magnitude faster. We acknowledge that evaluation on even larger datasets would further strengthen this point and will be explored in future work.
> > >
> > > [1] Schiebinger, G., Shu, J., Tabaka, M., et al. Optimal-Transport Analysis of Single-Cell Gene Expression Identifies Developmental Trajectories in Reprogramming. Cell, 2019.
> > >
> > > **Q2**
> > >
> > > We address this concern through the following points.
> > >
> > > **Standard formulation.**
> > > A balance parameter between transport and growth is intrinsic to UOT formulations. In the standard WFR formulation [1–3], this is given by a constant hyperparameter $\delta$, which defines the trade-off between transport and mass variation (Eq. 1 in the original formulation).
> > >
> > > **Non-constant growth.**
> > > Importantly, a constant $\delta$ does not imply a constant growth rate. The instantaneous growth is modeled by the learned field $g(x,t)$, which can vary across time and space. For example, even in the closed-form WFR geodesic between two Dirac measures, the mass evolves as
> > > $$
> > > m(t)=At^2-2Bt+m_0,
> > > $$
> > > (Appendix A, Eq. 5), yielding a time-dependent growth rate
> > > $$
> > > g(t)=\ln m(t),
> > > $$
> > > which is clearly non-constant. In practice, $g(x,t)$ is jointly determined by $\delta$ and the data (e.g., relative cell counts and densities across time points). The role of $\delta$ is to define a consistent global trade-off, rather than to enforce a static biological growth rate.
> > >
> > > Such constant trade-off parameters are common across many physical models. For example, in molecular dynamics, force field parameters are used to balance kinetic and potential energy in constructing the Lagrangian in classical mechanics; in phase-field modeling, the balance between bulk and interfacial energies must be prescribed for different materials. Here, the balance between kinetic energy and growth energy is analogous. In the absence of a clear first-principles characterization, such parameters generally require tuning. In fact, this issue is common across OT/loss-based models in the literature, for both static and dynamic formulations. It is therefore not straightforward to derive $\delta$ from first principles.
> > >
> > > The reviewer’s suggestion of a time-dependent penalty $\delta(t)$ is theoretically intriguing and could provide a more flexible description of biological processes. If explicit biological priors are available (e.g., known cell cycle bursts), designing a time-dependent $\delta$ could be a valuable extension to guide inference. This represents an interesting direction for future work, which would require additional theoretical and computational development.
> > >
> > >
> > > [1] Chizat, L., Peyré, G., Schmitzer, B., et al. *An interpolating distance between optimal transport and Fisher-Rao metrics.* Foundations of Computational Mathematics, 2018.
> > >
> > > [2] Chizat, L., Peyré, G., Schmitzer, B., et al. *Unbalanced optimal transport: Dynamic and Kantorovich formulations.* Journal of Functional Analysis, 2018.
> > >
> > > [3] Liero, M., Mielke, A., Savaré, G. *Optimal entropy-transport problems and a new Hellinger-Kantorovich distance between positive measures.* Inventiones mathematicae, 2018.

---

### Official Review · Reviewer_zYq3 · 2026-03-13

**Soundness:** 3
**Presentation:** 3
**Significance:** 4
**Originality:** 3
**Overall Recommendation:** 5
**Confidence:** 3

**Summary:**

This paper proposes WFR-MFM, a mean-flow matching method for dynamic unbalanced optimal transport under the Wasserstein–Fisher–Rao geometry. The central idea is to replace instantaneous transport/growth fields with time-averaged mean fields. Instantaneous velocity/growth fields are replaced with interval-level mean velocity and mean mass-growth fields, then learn these fields via a self-referential mean-flow matching objective. Empirically, the paper reports very large inference speedups over WFR-FM inference while keeping similar predictive quality.

**Compliance With Llm Reviewing Policy:**

Affirmed.

**Final Justification:**

Solid contribution to the field, the rebuttal answered all my questions thus I am increasing my score to a 5. I think this paper is above the acceptance threshold.

**Key Questions For Authors:**

* Can the authors better quantify when one-step inference is sufficient versus when a few-step refinement is needed? A more explicit rule or diagnostic would make the speed–accuracy trade-off more actionable.
* The perturbation experiment is synthetic. Do the authors have preliminary evidence on a real perturbation dataset, even a small-scale one, for which the same conditioning strategy transfers?
* Theorem 4.1 shows conditional/unconditional objective equivalence, but can the authors provide an argument that minimizing your stop-gradient self-referential loss recovers the true mean fields v,h, or that the target fixed point is unique/stable?

**Limitations:**

Yes.

**Strengths And Weaknesses:**

## Strengths

* The paper addresses a real bottleneck. Inference cost is a serious issue for dynamic unbalanced OT.
* The theory is rigorous. Theorem 4.1 gives a useful justification for optimizing the conditional objective rather than the unconditional one. The proposed extension from mean-flow ideas to the unbalanced/WFR setting is nontrivial because it must account for both transport and mass growth, not just mass-preserving motion.
* The evaluation is fairly broad in terms of conducted experiments: synthetic gene, Dyngen, high-dimensional Gaussian, and several real scRNA-seq datasets, plus a conditional perturbation benchmark. Additionally, the empirical runtime gains against WFR-FM are strong and appear to be the main practical contribution.

---

## Weaknesses

*  The novelty is meaningful but somewhat limited. Essentially, it is a specialization of mean-flow / one-step FM ideas to the WFR unbalanced OT setting, rather than a fundamentally new transport formulation.
* The theory does not fully justify the central learning objective. Theorem 4.1 shows equivalence between conditional and unconditional objectives, but it does not establish that minimizing the self-referential stop-gradient loss actually recovers the true mean fields, nor that the fixed point is unique or stable.
* The inference-speed results are convincing against the paper’s closest ODE-based baseline, WFR-FM, and Table 1 clearly shows large gains there. However, the wall-clock runtime comparison is much narrower than the broader accuracy comparison
* The title and main framing emphasize one-step inference, but the appendix shows at least one important case (CITE-seq) where one-step performance is clearly suboptimal, and 10 steps are needed to become competitive.

---

> ### Author Rebuttal · Authors · 2026-03-31
>
> **W1. Limited novelty.**
> We thank the reviewer for this question. Our key focus in this paper is the sampling efficiency, which is crucial in perturbation-response prediction. The proposed WFR-MFM achieves this goal and paves the way for further challenging  applications. This is a big success, as we consider.
>
> **W3. Runtime comparison is narrow.**
> We include broader results in Fig. 3 (3D) and Fig. 7 (2D, appendix). The table below reports inference times on the 100D EB dataset, showing WFR-MFM is substantially faster than all baselines.
>
> **Inference time on 100D EB (Slowdown relative to WFR-MFM)**
> | Method | Time (s) | Slowdown |  |  | |
> |---|---|---|---|---|---|
> | MMFM | 0.218 | 72.7× | UOT-FM | 1.664 | 554.7× |
> | Metric FM | 0.045 | 15.0× | VGFM | 0.251 | 83.7× |
> | SF2M | 0.209 | 69.7× | WFR-FM (RK) | 6.400 | 2133× |
> | MIOFlow | 0.085 | 28.3× | WFR-FM (Euler)| 0.169 | 56.3× |
> | TIGON | 3.060 | 1020× | VARRUOT | 1.496 | 498.7× |
> | DeepRUOT | 0.034 | 11.3× | **WFR-MFM** | **0.003** | **1.0×** |
>
> **Q1**
> We thank the reviewer for raising this. We provide an explicit adaptive stopping criterion.
>
> Let $p_0$ be the empirical distribution at $t=0$ with $N_0$ cells and $m_0 = 1/N_0$. Let  $s^{(K)} = \{(x_{1,n}^{(K)}, m_{1,n}^{(K)})\}_{n=1}^N$  be the result after $K$ steps.
>
> Define the discrepancy between iterates:
> $$
> \Delta_k = d(s^{(k-1)}, s^{(k)}),
> $$
>
> $$
> d = \frac{1}{N} \sum_{n=1}^N
> \left[
> \frac{\|x^{(k)} - x^{(k-1)}\|^2}{\|\mathbb{E}_{p_0}[x]\|^2}
> +
> \frac{|m^{(k)} - m^{(k-1)}|^2}{m_0^2}
> \right]
> $$
>
>
> This per-sample $\ell_2$ discrepancy efficiently approximates convergence, avoiding costly metrics such as $\mathcal{W}_1$.
>
> We adopt an adaptive stopping rule: iterative refinement stops once improvement falls below a tolerance. When the one-step solution is near a fixed point, $\Delta_k$ is small and further steps are unnecessary; otherwise, iterations continue until convergence.
>
> **Adaptive stopping rule**
> - **Input:** maximum step count $K_{\max}$, tolerance $\tau \in (0,1)$, patience $P \in \mathbb{N}$, distance function $d(\cdot,\cdot)$
> - **Output:** selected iterate $s^{(k_{\mathrm{best}})}$ and best step count $k_{\mathrm{best}}$
>
> 1. Initialize
>    $k_{\mathrm{best}} \gets 1$, $\Delta_{\mathrm{best}} \gets+\infty$, $c \gets 0$
>
> 2. For $k = 1,2,\dots,K_{\max}$:
>    - Compute $s^{(k)}$
>    - If $k > 1$:
>      - $\Delta_k \gets d(s^{(k-1)}, s^{(k)})$
>      - If $\Delta_k < (1-\tau)\,\Delta_{\mathrm{best}}$:
>        - $\Delta_{\mathrm{best}} \gets \Delta_k$
>        - $k_{\mathrm{best}} \gets k$
>        - $c \gets 0$
>      - Else:
>        - $c \gets c + 1$
>      - If $c \ge P$:
>        - Return $s^{(k_{\mathrm{best}})}, k_{\mathrm{best}}$
> 3. Return $s^{(k_{\mathrm{best}})}, k_{\mathrm{best}}$
>
> Here, $\tau$ controls improvement strictness (smaller $\tau$ → more steps), and $P$ is the number of tolerated non-improving steps.
>
> On the 2D Gene dataset, $P=2$ and $\tau=0.15$ stop at $K=22$, achieving near-optimal $\mathcal{W}_1$ and RME (Fig.1) using only the local $\ell_2$ discrepancy.
>
> **Q2**
> Extending WFR-MFM to such datasets is ongoing work that we are preparing as a separate study.
> To address the reviewer’s interest, we provide a preliminary evaluation on a PBMC perturbation dataset [1] with $\sim 10$M cells from 12 donors and 90 cytokine perturbations.
> We hold out 20% of perturbation conditions for testing, use 5,412 HVGs with 100-d PCA, and encode conditions into 4,096-d embeddings.  As this analysis is preliminary, we do not yet include baseline comparisons. These will be systematically evaluated in follow-up work. Results averaged across conditions are shown below:
>
> **Results on PBMC**
> | Metric         | Train  | Test   |
> |----------------|--------|--------|
> | Gene MSE       | 7.4e-4 | 2.2e-3 |
> | Gene MAE       | 0.0145 | 0.0194 |
> | Gene $R^2$     | 0.987  | 0.962  |
> | DEG50 $R^2$    | 0.920  | 0.625  |
> | $W_1$          | 0.140  | 0.142  |
> | KL             | 6.64   | 6.83   |
>
> WFR-MFM shows consistent train/test performance, suggesting generalization to real perturbations. These results are preliminary and provided in response to the reviewer. A full evaluation with baseline comparisons and additional datasets will be reported separately.
>
> [1] Parse Biosciences. *10 Million Human PBMCs in a Single Experiment*. 2024.
>
> **Q3**
> We thank the reviewer for the question on the uniqueness and recoverability of $v$ and $h$.  We show that the stop-gradient objective admits a unique fixed point that recovers the true mean fields. Suppose the stop-gradient loss is perfectly minimized, reaching a fixed point where
> $$
> v_{\theta}(x, t, T)=u_t(x)+(T - t)[\partial_t v_{\theta}(x, t, T)+(\nabla_x v_{\theta}(x, t, T))\,u_t(x)]
> $$
> Fix $x, T$ and define $f(t) := v_\theta(x, t, T)$. Then for $t < T$,
> $$
> f(t) = u_t + (T - t) f'(t).
> $$
> Integrating gives
> $$
> f(t) = \frac{1}{T - t} \int_t^T u_\tau \, d\tau.
> $$
> This proves that the unique solution to this ODE is exactly the true mean velocity field. The same argument applies to $h$.

---

> > ### Author Rebuttal · Reviewer_zYq3 · 2026-04-02
> >
> > I would like to thank the authors for their detailed and thoughtful rebuttal. I am especially impressed by the runtime results, and I also appreciate the inclusion of the additional PBMC perturbation results. While I understand that these perturbation experiments are intended for a separate study, I believe the current manuscript would also be strengthened by including them. Overall, the paper’s strong empirical performance, together with the authors’ careful and detailed responses, has addressed my main concerns and prompts me to raise my score to 5.

---

### Decision · Program_Chairs · 2026-04-30

**Decision:**

Accept (regular)

**Comment:**

**Summary**

In this paper, the authors extend existing flow map methodology to unbalanced transport to reach fast one-step inference in unbalanced settings. More precisely, the authors propose to extend the mean flow paradigm to the unbalanced framework. To do so, they need to define a consistency objective for the mass variation. This is based on the Wasserstein–Fisher–Rao (WFR) Mean Flow Matching framework which computes the WFR Optimal Transport between two Dirac masses. The authors then evaluate their method on a few different benchmarks, including synthetic gene expression datasets, high-dimensional Gaussian mixtures, and real-world data such as scRNA-seq and CITE-seq datasets. They report genuine speed-up while preserving the quality.

**Reviewer concerns**

Reviewer concerns focused mostly on the limited novelty and the scalability of the methods. Most of those concerns were resolved during the discussion period with the authors.